DOI: 10.1038/s41467-018-04921-2　　**OPEN**

# Mouse MRI shows brain areas relatively larger in males emerge before those larger in females

Lily R. Qiu[1,2,3], Darren J. Fernandes[2,4], Kamila U. Szulc-Lerch[2,3], Jun Dazai[2], Brian J. Nieman[2,4], Daniel H. Turnbull[5,6], Jane A. Foster [7], Mark R. Palmert[1,8,9] & Jason P. Lerch[2,3,4]

Sex differences exist in behaviors, disease and neuropsychiatric disorders. Sexual dimorphisms however, have yet to be studied across the whole brain and across a comprehensive time course of postnatal development. Here, we use manganese-enhanced MRI (MEMRI) to longitudinally image male and female C57BL/6J mice across 9 time points, beginning at postnatal day 3. We recapitulate findings on canonically dimorphic areas, demonstrating MEMRI's ability to study neuroanatomical sex differences. We discover, upon whole-brain volume correction, that neuroanatomical regions larger in males develop earlier than those larger in females. Groups of areas with shared sexually dimorphic developmental trajectories reflect behavioral and functional networks, and expression of genes involved with sex processes. Also, post-pubertal neuroanatomy is highly individualized, and individualization occurs earlier in males. Our results demonstrate the ability of MEMRI to reveal comprehensive developmental differences between male and female brains, which will improve our understanding of sex-specific predispositions to various neuropsychiatric disorders.

[1] Institute of Medical Science, University of Toronto, Toronto, ON M5S 1A8, Canada. [2] Mouse Imaging Centre, The Hospital for Sick Children, Toronto, ON M5T 3H7, Canada. [3] Department of Neurosciences and Mental Health, The Hospital for Sick Children, Toronto, ON M5G 1X8, Canada. [4] Department of Medical Biophysics, University of Toronto, Toronto, ON M5G 1L7, Canada. [5] Skirball Institute of Biomolecular Medicine, New York University School of Medicine, New York, NY 10016, USA. [6] Department of Radiology, New York University School of Medicine, New York, NY 10016, USA. [7] Department of Psychiatry & Behavioural Neurosciences, McMaster University, Hamilton, ON L8S 4L8, Canada. [8] Division of Endocrinology, The Hospital for Sick Children, Toronto, ON M5G 1X8, Canada. [9] Departments of Paediatrics and Physiology, The University of Toronto, Toronto, ON M5S 1A8, Canada. These authors contributed equally: Lily R. Qiu, Darren J. Fernandes. These authors jointly supervised this work: Mark R. Palmert, Jason P. Lerch. Correspondence and requests for materials should be addressed to L.R.Q. (email: lily.qiu@sickkids.ca)

Sex differences in the brain are pervasive as demonstrated by differences in a wide range of processes including pain[1], learning and memory[2], and language[3]. Notably, there are robust sex differences in the prevalence, age of onset, and course of various psychiatric disorders. Males tend to have a predisposition for disorders that have earlier onset, many of which emerge during childhood, including autism spectrum disorders, attention deficit disorders, and Tourette syndrome. Females tend to have a predisposition for disorders that have later onset, during adolescence and early adulthood, which include major depressive disorders, anxiety disorders and eating disorders[4]. Key to understanding these sex-specific vulnerabilities and predispositions is a better understanding of the normal development of sex differences in the brain.

Noninvasive image acquisition using MRI enables repeated scanning of the same individual to study longitudinal development. This is important because of the temporal nature of sex differences in the brain[5], behaviors, and risk factors for psychiatric disorders. Mesoscopic anatomy is also translatable between model organisms, such as mice, and humans due to the structural homology of their brains.

Anatomical sex differences have been studied in both rodents and humans. Histological studies in rodents have revealed some of the most well-known sex differences in the brain, such as the bed nucleus of the stria terminalis (BNST), the medial nucleus of the amygdala (MeA)[6], and the medial preoptic nucleus of the hypothalamus (MPON)[7]. Due to the rodent's accelerated lifespan, the sexually dimorphic development of these areas has been well-characterized beginning from neonatal life[8,9]. Using MRI, several anatomical sex differences have been found in the human brain as well. Whole brain size, the putamen, globus pallidus, basal ganglia, amygdala and hypothalamus are larger in males, while the caudate, thalamus and hippocampus are larger in females[10–13].

Many of these sex differences arise during a critical period of sexual differentiation that occurs during neonatal life. The presence or absence of hormones in this developmental window organize the structure of the brain[14]. These sex differences are also subject to change throughout life as activational effects of hormones present during puberty and onwards can modulate neuroanatomy in males and females[15]. In addition, sex chromosomes themselves influence sex differences in brain anatomy[16].

Due to the nature of sex differences in the brain and how they develop, there are shortcomings in the methods for investigating neuroanatomy in both rodents and humans. The accelerated life spans of rodents and their availability for histology renders them useful for investigating the development of specific nuclei; however, small nuclei are often examined in isolation, therefore neglecting others. In humans, MRI allows for whole-brain investigation, but lacks the ability to detect smaller sexually dimorphic nuclei, even though they do exist. Furthermore, the long lifespan of humans renders studying development over a comprehensive period difficult. Most studies that investigate sex differences in the human brain are either cross-sectional, or examine sex differences on a relatively short developmental timescale. A recent meta-analysis examining sex differences in human brain structure across life determined that very few, if any, studies investigate sex differences during infancy and early childhood (0–6 years of age)[12]. Of those that do, however, there is an indication that some aspects of neuroanatomy, such as cortical gyrification, are larger in males in infancy compared to females[17], pointing to the existence of early life sex differences, and further emphasizing the need to examine sex differences within neonatal development. Because there are opportunities for activational sex

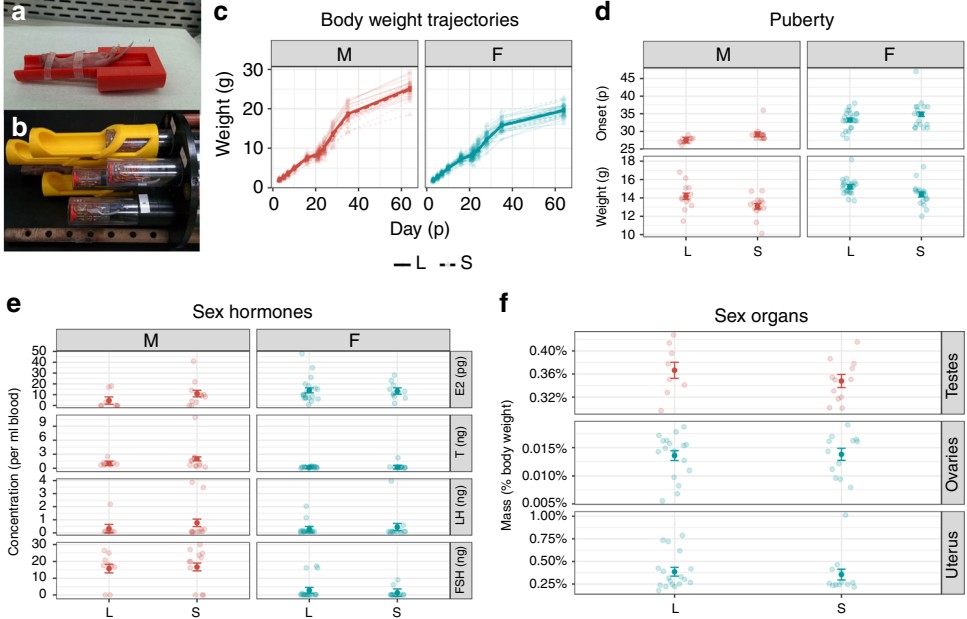

**Fig. 1** Scanning apparatus, and scanned mice (S) vs. non-scanned littermates (L) data. **a** Custom 3D printed holders for neonatal mice. **b** Up to seven mice at a time were scanned using a saddle coil array. **c** Body weight was not significantly different between scanned and unscanned mice ($\chi_2^2 = 0.13$, $P = 0.94$). There was also no interaction effect of scanning and sex ($\chi_2^2 = 0.097$, $P = 0.75$). Trendlines and bars for standard error were calculated using centered linear mixed-effect models. **d** Repeated scanning did not have a significant effect on puberty onset ($F_{3,64} = 24.67$, $t_{64} = 1.56$, $P = 0.12$) but did have a significant effect on weight at puberty ($F_{3,64} = 10.67$, $t_{64} = -2.43$, $P = 0.02$). However, neither measure had a sex-scanning interaction ($t_{64} = -0.045$, $P = 0.96$ and $t_{64} = 0.451$, $P = 0.65$). **e** Scanning did not have a significant effect on the levels of sex hormones estradiol ($F_{3,48} = 1.11$, $t_{48} = 0.16$, $P = 0.87$), testosterone ($F_{3,48} = 4.17$, $t_{48} = 0.32$, $P = 0.15$), LH ($F_{3,48} = 0.68$, $t_{48} = 1.03$, $P = 0.31$), FSH ($F_{3,48} = 14.19$, $t_{48} = 0.28$, $P = 0.78$). **f** Organ weight of testes ($F_{1,18} = 1.05$, $t_{18} = -1.0$, $P = 0.32$), ovaries ($F_{1,28} = 0.03$, $t_{28} = 0.17$, $P = 0.87$), and uteri ($F_{1,28} = 0.17$, $t_{28} = -0.41$, $P = 0.69$) were not affected by repeated scanning. Means and bars for standard error for **d**, **e**, and **f** were estimated using linear models

differences to emerge, a comprehensive timeline is needed to capture changes across the rest of postnatal life as well. Furthermore, there is a need for a bridge that connects in-depth cellular and molecular work about sex differences arising from animal studies, to macroscopic imaging information about sex differences in the human brain.

Longitudinal in vivo manganese-enhanced magnetic resonance imaging (MEMRI) in mice addresses many of these shortcomings and is ideally suited for the comprehensive study of neuroanatomical sex differences. This technique builds upon ex vivo mouse MRI studies that have been used to capture dimorphisms in the whole brain: from large structures such as the cortex, cerebellum and thalamus; to smaller nuclei in the hypothalamus[18]. In MEMRI, systemic administration of manganese chloride, $MnCl_2$, results in visualization of brain architecture, particularly in neonatal neuroanatomy where large-scale changes of cellular composition may pose challenges for optimizing MRI contrast[19]. MEMRI has been used to acquire in vivo images of the brain of postnatal rodents as young as 1 day old, and is well tolerated by neonatal rodents through repeated rounds of imaging[20]. Since MEMRI allows for repeated in vivo imaging of the same animal, neurodevelopmental sexual dimorphisms can be studied beginning in early life and into adulthood.

Here, we use MEMRI to investigate the development of structural sex differences in the mouse brain beginning from early neonatal life. Male and female C57BL/6J mice were scanned longitudinally with MEMRI across 9 postnatal time points (p), at days 3, 5, 7, 10, 17, 23, 29, 36, and 65. First, known sex differences in the BNST, MeA, and MPON were investigated to affirm that MEMRI is a robust technique for detecting sex differences in the brain. Second, linear mixed-effects models were used to identify sexually dimorphic areas across the whole brain and characterize their development. Third, k-means clustering was used to find areas that share patterns of sexually dimorphic development, and spatial gene expression patterns in sexually dimorphic areas were uncovered. Finally, we investigated emergence of brain individualization, and whether individualization showed any sex differences.

## Results

**MEMRI captures neurodevelopment without adverse effects.** The methodology and image analysis techniques employed here are an extension of previous work[20], expanding both the throughput and the imaging window to include adult brain development, and applied to both sexes. Imaging over such a

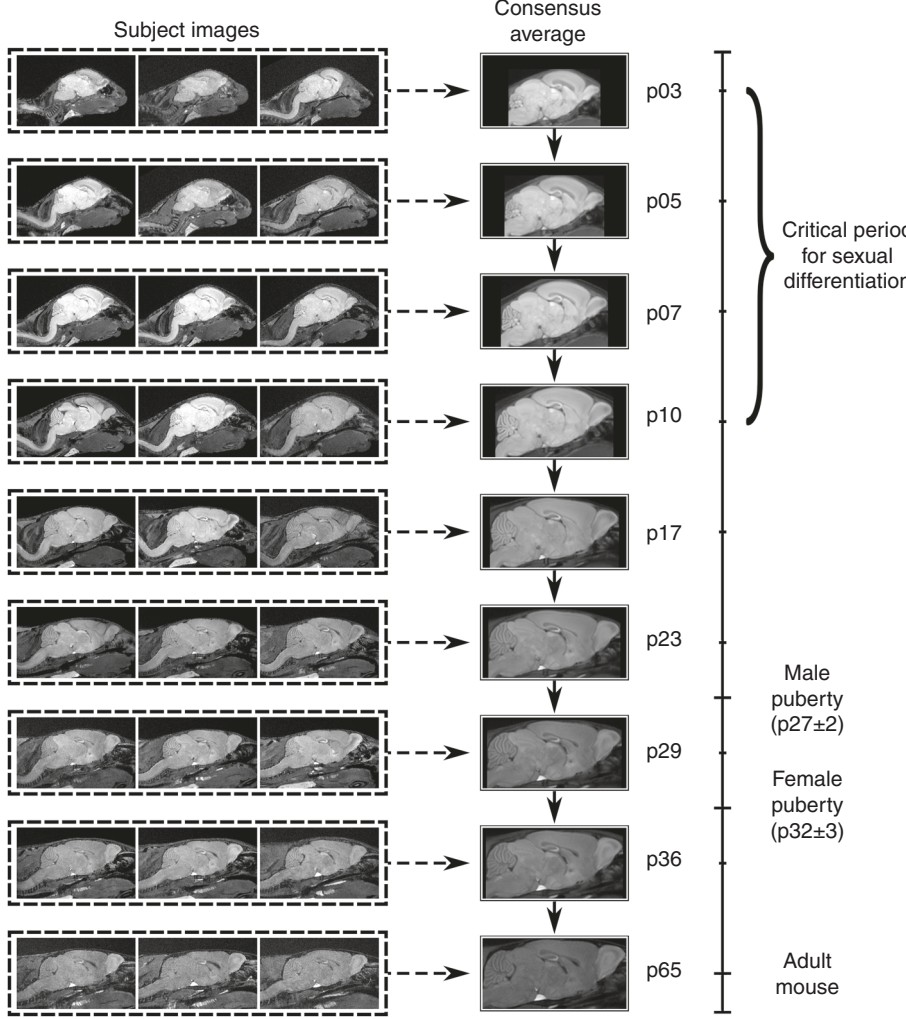

**Fig. 2** MR images were registered using a two-level approach. In Level 1 (dashed lines), images from each time point were registered together to create a consensus average for each age. In Level 2 (solid lines), the consensus averages for each age were registered to the following adjacent age. The transformations created from concatenating Level 1 (dashed arrows) and Level 2 (solid arrows) allow us to map points in the p65 consensus average to all images

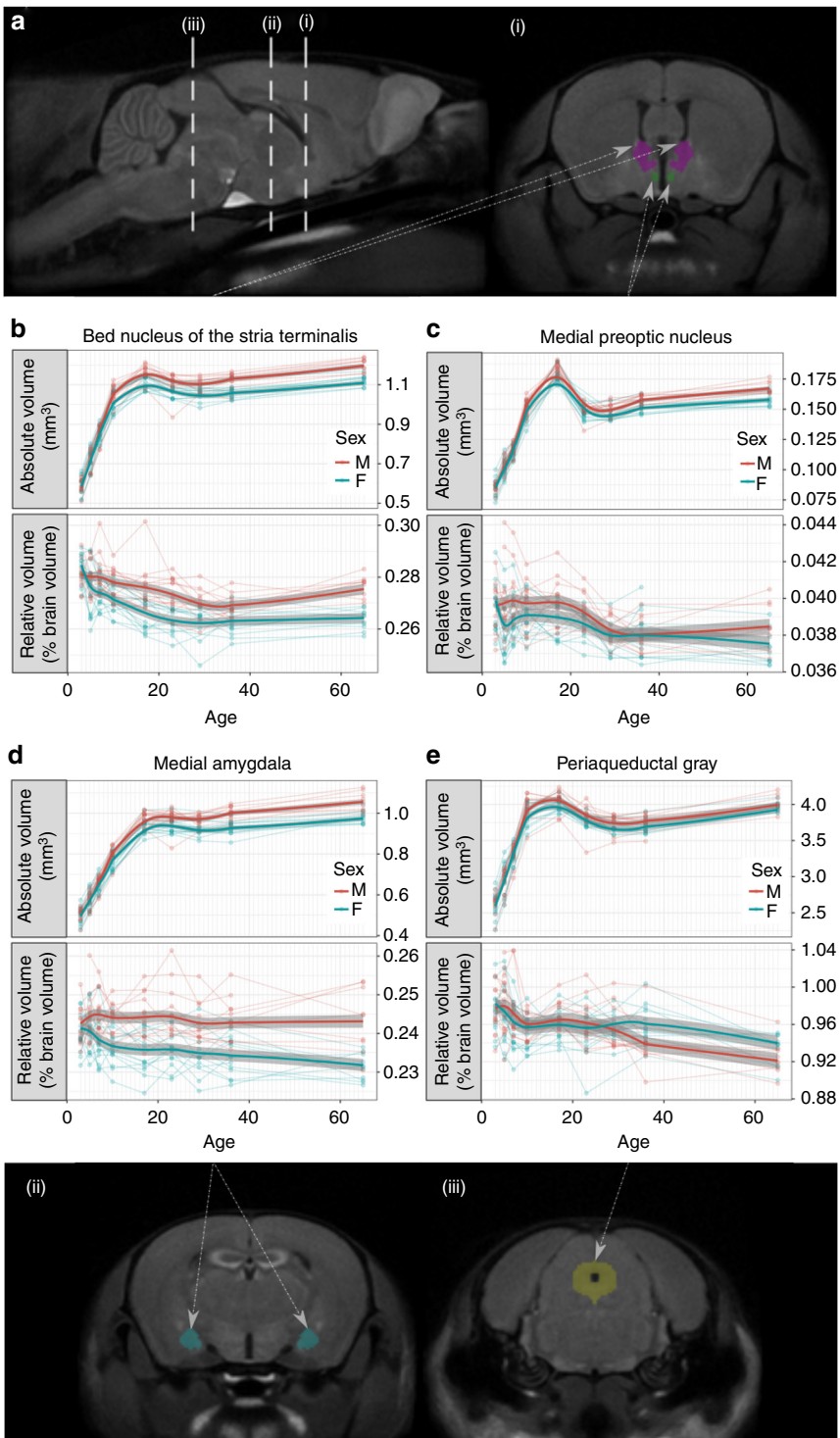

**Fig. 3** MEMRI captures sex differences in brain structure sizes. **a** Sagittal and coronal slices of the average p65 brain showing segmentations of mouse brain structures: bed nucleus of the stria terminalis (BNST), medial preoptic nucleus (MPON), medial nucleus of the amygdala (MeA), and periaqueductal gray (PAG). **b**–**e** the absolute and relative volumes of these structures over time. Points in these plots represent measurement at a time point for individual mice and lines connect the measurements for the same mouse over time. Shaded regions represent standard error estimated using linear mixed-effects models. Relative volume corrects for whole-brain size differences between subjects and is expressed as a percent difference from the average volume of the brain structure. Using linear mixed-effects models, we recapitulate known canonical sex differences in absolute volumes of the MeA ($\chi^2_9 = 66$, $P<10^{-10}$), BNST ($\chi^2_9 = 77$, $P<10^{-12}$), and MPON ($\chi^2_9 = 28$, $P<10^{-3}$). Sex differences in these structures emerge pre-puberty, at around p10. Using relative volumes to correct for whole brain size, we see that sex differences in these structures are preserved but differences emerge earlier in development around p5. PAG relative volume also shows a significant effect of interaction between sex and age ($\chi^2_8 = 21$, $P<10^{-2}$), which is not found in the absolute volumes ($\chi^2_8 = 10$, $P = 0.2$). Taken together, these results indicate that relative volumes obtained by MEMRI are a sensitive marker for detecting canonical and novel sexual dimorphisms in neuroanatomy

comprehensive time window allows us to study neuroanatomical change that occurs across the human equivalent of prenatal life (p3–10), birth (p10), childhood (p10–p29), puberty and adolescence (p23–p36), and adulthood (p65)[21].

To accommodate the small but rapidly changing body size of neonatal mice, custom 3D-printed holders were created (Fig. 1a). Up to 7 mice were scanned in individual saddle coils simultaneously (Fig. 1b).

Mice undergoing repeated MEMRI scanning experienced increased handling, anesthesia exposure, and $MnCl_2$ injections. To determine if repeated scanning had any effects on development, several measurements were collected from scanned mice and their non-scanned littermates. Weight at puberty (Fig. 1d) was significantly affected in scanned mice; however, we did not find a significant effect of scanning on any other measurements collected (Fig. 1c, e, f), nor did we find evidence for an interaction between scanning and sex. Weight could impact circulating steroid levels as adipose tissue has aromatase activity[22]. Our scanned mice weighed less at puberty, so it is possible that they had lower levels of circulating estradiol. However, there were no differences in pubertal timing, which suggests that there were no functional effects of potentially differing levels of estradiol between groups of mice. Neonatal anesthesia exposure can cause cell death in the brain, and can affect males more severely[23]; thus, we assessed several neonatal neurodevelopmental outcomes (righting reflex, eye opening and open field) on additional groups of mice (Supplementary Methods). We found no significant effects of group or a group–sex interaction on any of the neonatal metrics collected (Supplementary Fig. 1). We conclude that the effect of MEMRI scanning on neurodevelopment was small and not sexually biased.

Image registration is necessary for identifying homologous features in MR images, thereby allowing statistical comparison. To accommodate the rapid changes in brain shape during early development (Supplementary Movie 1), we modified an existing image registration pipeline[24] into a two-level registration pipeline (Fig. 2, see Methods for details). In the first level, images from different mice collected at the same time point were registered together, creating a consensus average brain for each age. In the second level, the age-specific consensus average brains were registered to one another serially to map them all to the p65 consensus average space. By concatenating the transformations from the first and second levels, we achieved point-correspondences between all images and generated transformations mapping all images—from every mouse and time point—to the p65 consensus average space. Computation of the determinants of these transformations allows measurement of the volumetric differences between all images at individual points in the brain or across defined structures.

**MEMRI can detect canonical and novel sexual dimorphisms**. To compare structure volumes between sexes, an atlas that segments 182 structures in the adult mouse brain[25] was overlaid onto the p65 consensus average brain. Figure 3a (i, ii) shows segmentations of the MeA, BNST, and MPON, which are canonical sexually dimorphic areas. Extensive literature has shown that these areas are larger in the male brain. MEMRI captures these sexual dimorphisms (top-panels, Fig. 3b–d) and shows that these dimorphisms emerge between p10 and p17. Males tend to have bigger brains than females[18]; as MEMRI allows for whole-brain imaging, we were also able to identify dimorphisms in brain structure relative volumes—that is, subjects' structure volumes divided by their whole-brain volume. Similar to absolute volumes, males had larger relative volumes of BNST, MeA, and MPON. However, relative volumetric sex differences emerged earlier than the absolute volume differences, around p5. Moreover, several structures, such as the periaqueductal gray (PAG) (Fig. 3e), exhibited no sex-dependent growth differences in absolute volumes, but did exhibit significant differences in relative volumes. Compared with male-enlarged structures, the PAG became relatively larger in females at a later developmental time, around p29 (summary structure data in Table 1 and Supplementary table 1). Correction for whole-brain size using relative volume measurements from MEMRI data reveal time courses of well-established and novel sexual dimorphisms, highlighting the strength of whole-brain MRI.

**Table 1 Mean volume of sexually dimorphic structures in males and females**

| Age | Bed nucleus of the stria terminalis | | Medial preoptic nucleus | | Medial amygdala | | Periaqueductal gray | |
|---|---|---|---|---|---|---|---|---|
| | M | F | M | F | M | F | M | F |
| Absolute volume ($mm^3$) | | | | | | | | |
| 3 | 0.589 | 0.583 | 0.0856 | 0.0844 | 0.506 | 0.490 | 2.630 | 2.580 |
| 5 | 0.719 | 0.724 | 0.103 | 0.102 | 0.573 | 0.578 | 2.950 | 2.980 |
| 7 | 0.857 | 0.850 | 0.117 | 0.115 | 0.660 | 0.653 | 3.310 | 3.310 |
| 10 | 1.060 | 1.010 | 0.152 | 0.147 | 0.800 | 0.764 | 3.960 | 3.860 |
| 17 | 1.160 | 1.110 | 0.184 | 0.178 | 0.973 | 0.933 | 4.070 | 3.990 |
| 23 | 1.110 | 1.060 | 0.149 | 0.146 | 0.976 | 0.938 | 3.820 | 3.770 |
| 29 | 1.100 | 1.050 | 0.150 | 0.146 | 0.966 | 0.917 | 3.740 | 3.670 |
| 36 | 1.130 | 1.060 | 0.158 | 0.152 | 1.000 | 0.930 | 3.760 | 3.690 |
| 65 | 1.200 | 1.120 | 0.168 | 0.158 | 1.060 | 0.983 | 4.010 | 3.970 |
| Relative volume (% Brain) | | | | | | | | |
| 3 | 0.2810 | 0.2850 | 0.0394 | 0.0400 | 0.2420 | 0.2410 | 0.9740 | 0.9840 |
| 5 | 0.2800 | 0.2750 | 0.0397 | 0.0382 | 0.2440 | 0.2400 | 0.9830 | 0.9700 |
| 7 | 0.2810 | 0.2740 | 0.0400 | 0.0389 | 0.2450 | 0.2380 | 0.9750 | 0.9580 |
| 10 | 0.2780 | 0.2710 | 0.0396 | 0.0390 | 0.2440 | 0.2360 | 0.9590 | 0.9510 |
| 17 | 0.2760 | 0.2650 | 0.0399 | 0.0388 | 0.2440 | 0.2350 | 0.9700 | 0.9580 |
| 23 | 0.2740 | 0.2630 | 0.0393 | 0.0386 | 0.2450 | 0.2350 | 0.9620 | 0.9490 |
| 29 | 0.2700 | 0.2620 | 0.0383 | 0.0378 | 0.2430 | 0.2350 | 0.9580 | 0.9560 |
| 36 | 0.2690 | 0.2630 | 0.0380 | 0.0380 | 0.2430 | 0.2340 | 0.9400 | 0.9590 |
| 65 | 0.2770 | 0.2640 | 0.0386 | 0.0373 | 0.2430 | 0.2310 | 0.9240 | 0.9340 |

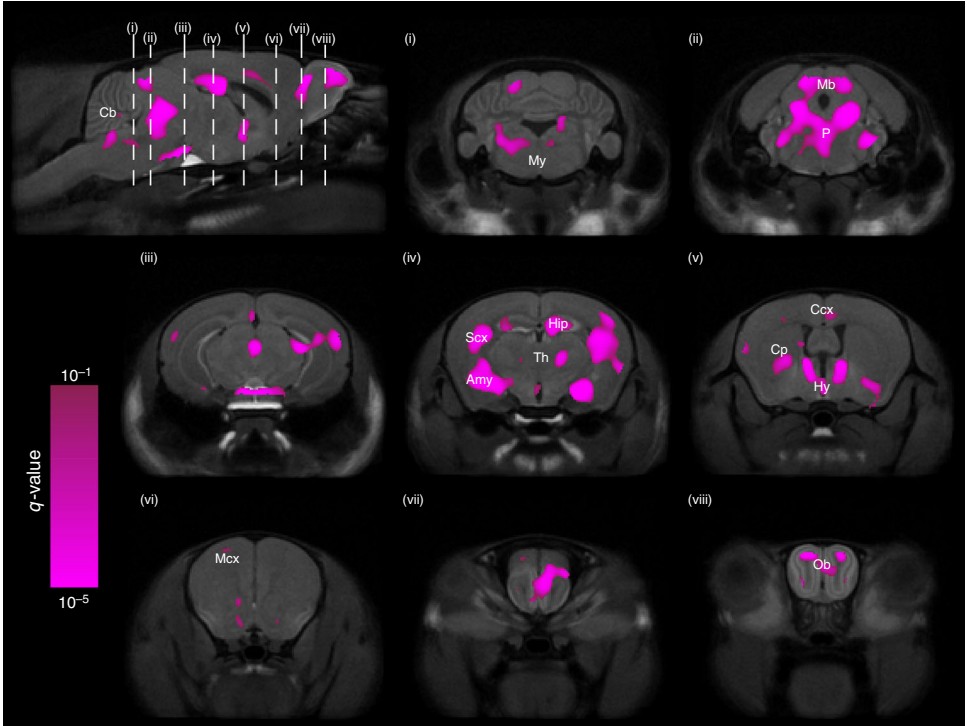

**Fig. 4** Sexually dimorphic neuroanatomy. Two linear mixed-effects models were fit to the data: Model 1 had sex as a predictor and Model 2 did not. Every voxel was assessed on whether Model 1 had a significantly better fit than Model 2. The resulting q-values (thresholded to q<0.1) were overlaid on p65 average brain cross-sections, identifying several regions in the brain where volumes are significantly dependent on sex. These sexually dimorphic regions include parts of the cerebellum (Cb), medulla (My), midbrain (Mb), pons (P), PAG, thalamus (Th), hippocampus (Hip), amygdala (Amy), sensory cortex (SCx), hypothalamus (Hy), cingulate cortex (Ccx), caudoputamen (Cp), BNST, motor cortex (Mcx), and olfactory bulbs (Ob)

**Regions larger in males emerge before those larger in females**. To investigate the presence of novel sexually dimorphic areas across the whole brain, we examined anatomical differences at the voxel level. The registration pipeline generates a Jacobian determinant field for every image. At every voxel, the value of this field measures the volumetric growth or shrinkage compared to the average brain at p65. Statistics were conducted on the logarithm of the relative Jacobian determinants.

To test the effect of sex on region volume, we ran two linear mixed-effects models on a per-voxel basis: one with fixed effects of sex, age, their interaction and random effect of mouse ID; and a similar one without the effect of sex (and therefore also without sex–age interaction). A log-likelihood test was performed between these two models to assess the significance of sex. Our analysis revealed that sex has widespread influence on neuroanatomical development (Fig. 4); including regions such as the cerebellum, midbrain, pons, medulla, PAG, thalamus, hypothalamus, hippocampus, amygdala (defined in our atlas as all amygdalar nuclei except the MeA), caudoputamen, BNST and olfactory bulbs (OB).

To visualize the effect that sex has on brain development at particular time points, we fit a model with both age and sex as predictors, but translated the age term such that it was zero at the time point of interest. Nine such models were fit, each centered to one imaging time point. The t-statistic field associated with the sex term from each age-centered model was overlaid on the consensus average image from that time point (Fig. 5, Supplementary Movies 2–4). Areas shown in red are relatively larger in males, and areas shown in blue are relatively larger in females. It is clear that areas larger in males or larger in females differed in the timing of their emergence. Regions that were relatively larger in males predominated the brain in neonatal, pre-pubertal life and encompassed areas in the MeA, hypothalamus (MPON), BNST, anterior cingulate cortex, hippocampus, and OB. In

contrast, areas that were relatively larger in females predominated the brain later in life, post-puberty, and encompassed broad, diffuse networks of areas that included parts of the cerebellum, pons, midbrain, PAG, caudoputamen, thalamus and cortex.

**Distinct trajectories of sexually dimorphic development**. We sought to identify groups of sexually dimorphic regions that had similar developmental trajectories. For all sexually dimorphic voxels identified from our prior analysis (Fig. 4), we computed the effect size of sex on relative determinants for each time point. Effect size is positive for regions larger in males and negative for regions larger in females. Using k-means, we then clustered the voxels by their effect size time-series into 4 developmental trajectories (Fig. 6). To study differences in growth rate of sexually dimorphic regions, we first fit splines at every voxel for every individual, and differentiated the result to estimate growth rate.

Cluster 1 describes areas in the brain that were relatively larger in males throughout development, and this dimorphism emerged early in development. Voxels from this cluster reside in the BNST, MeA, MPON, OB, hippocampus, and cingulate cortex. This cluster had a higher growth rate in males, which stabilized in later life. Cluster 2 corresponds to regions relatively larger in males that emerged later in development. This cluster shows growth rate that is biased towards males in early life, and includes the pallidum and OB. Clusters 3 and 4 describe areas that both became relatively larger in females by adulthood, but differed in early development. Similar to Cluster 1, voxels in Cluster 3 began as relatively larger in males. However, during peripubertal development, they transitioned to becoming relatively larger in females. Voxels in this cluster fall in the PAG, inferior colliculus, cortex, amygdala, and caudoputamen. Cluster 4 showed little sex difference in early life, but across puberty a sex difference

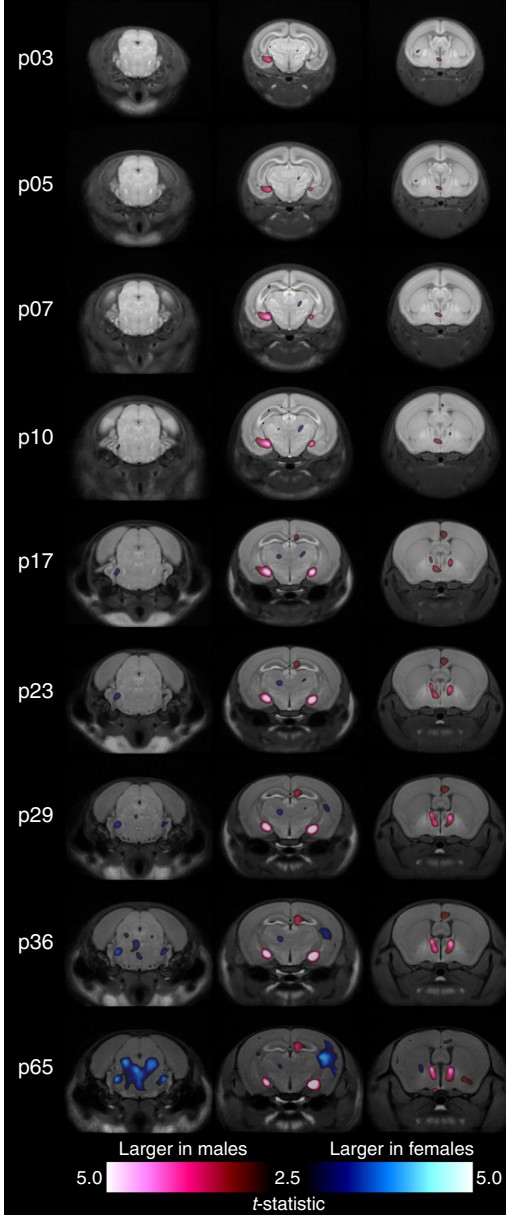

Larger in males Larger in females

5.0 2.5 5.0

$t$-statistic

**Fig. 5** Expansion of neuroanatomical structures in males and females over time. Each column follows a coronal cross-section of the developing brain through the nine experimental time points with red regions indicating areas relatively larger in males and blue regions indicating areas relatively larger in females. Nine age-centered linear mixed-effects models were fit to the data, one for each experimental time point. Each model had identical predictors of sex and age, however the age terms were translated such that they were 0 at the time point of interest. For each time point and corresponding age-centered model, the average brain at the time point was overlaid with the $t$-statistics map associated with the main effect of sex. Statistics are thresholded to 10% FDR in the model centered at p65. Regions relatively larger in males predominate the brain in pre-puberty life (as shown by the second and third image columns). Post-puberty however, regions larger in adult females begin showing dimorphisms

emerged as these areas became relatively larger in females. Areas in Cluster 4 reside in the cerebellum, pons, medulla, midbrain, PAG, thalamus, hippocampus, caudoputamen, nucleus acumbens, sensory cortices, and parts of the OB. The growth rate of both Clusters 3 and 4 show strong bias towards females peaking around puberty. This pattern of relatively larger areas in males

emerging in early life, and larger areas in females emerging in post-pubertal life holds true for absolute volumes and cortical thickness measures as well (Supplementary Fig. 2–4).

To investigate possible mechanistic drivers, we also compared regions of sexually dimorphic development with spatial gene expression data from the Allen Brain Institute[26], which has genome-wide gene expression maps in the adult male mouse brain. Genes that showed preferential spatial expression in sexually dimorphic regions include *Esr2* (Fig. 7a), *Esr1* (Supplementary Fig. 5a), and *Slc6a4* (Fig. 7b), which encode estrogen receptors beta and alpha, and the serotonin transporter, respectively (full list of genes in Supplementary Table 1). Compared to a background set of genes from all chromosomes, we also found that sex chromosome genes have a significantly higher likelihood of preferential spatial expression in regions of sexually dimorphic development (Fig. 7c).

**Individualization of neuroanatomy emerges earlier in males.** Genetic and environmental variability is limited with the usage of standardized laboratory mice; however, neuroanatomy is still remarkably individualized by adulthood. Most studies typically treat this individualization as variability that is either accounted for as residuals or random effects in a statistical model in order to study other factors influencing variability, such as sex. We instead hypothesized that some of the variability in structure volumes of mature brains are inherently individualized. We sought to identify when this neuroanatomical individuality emerged across development, and whether it differed across sexes.

We used a set of linear mixed-effects regression models to predict structure volume at a specified time from the structure's volume at earlier times. Similar to validation methods followed by Tavor et al.[27], we used a leave-one-out approach to evaluate our model: the model for predicting volume of a structure $s$ from a particular subject $i$ at a time $t$ was not trained on data containing information about any structures of subject $i$ at time $t$. Figure 8a demonstrates the model's sensitivity by plotting the predicted and observed volume of three representative structures for every individual at p36. We quantitatively assessed the specificity of the model (Fig. 8b) and found that predicted structure volumes for subject $i$ generally matched observations for subject $i$ closer than observations for other subjects. This is despite the fact that when trying to predict any subject at a time point, the model was trained on everything but the data from the subject at that time point. Yet, the prediction made is closer to the unseen subject data than the seen data from other subjects.

The model accurately captures neuroanatomical individualization in the mature mouse brain. To investigate when this individuality emerged, we withheld more information regarding the subject to be predicted. When only considering data from p3 to predict p36 structure volumes, model specificity and accuracy is quite poor (56% probability predicted volumes match predicted subject better than other subjects; 0.13 mm³ root-mean-square-difference (RMSD)). However, when considering data from p10 and younger, specificity and accuracy improves (70%; 0.12 mm³) and is quite high for data from p17 and younger (85%; 0.096 mm³) (Fig. 8c). Thus, only the first 10–17 days of brain development is sufficient to predict individualization of mature mouse brain anatomy.

We plotted how the prediction accuracy (RMSD) at p36 changed for all subjects as we included more data closer to p36 (Fig. 8d). As expected, accuracy increased as more data was included. Accuracy of predicting male neuroanatomy was not significantly different from predicting female neuroanatomy at any time point. However, male accuracy improved earlier than female accuracy (permutation test, $P = 0.025$), needing only data

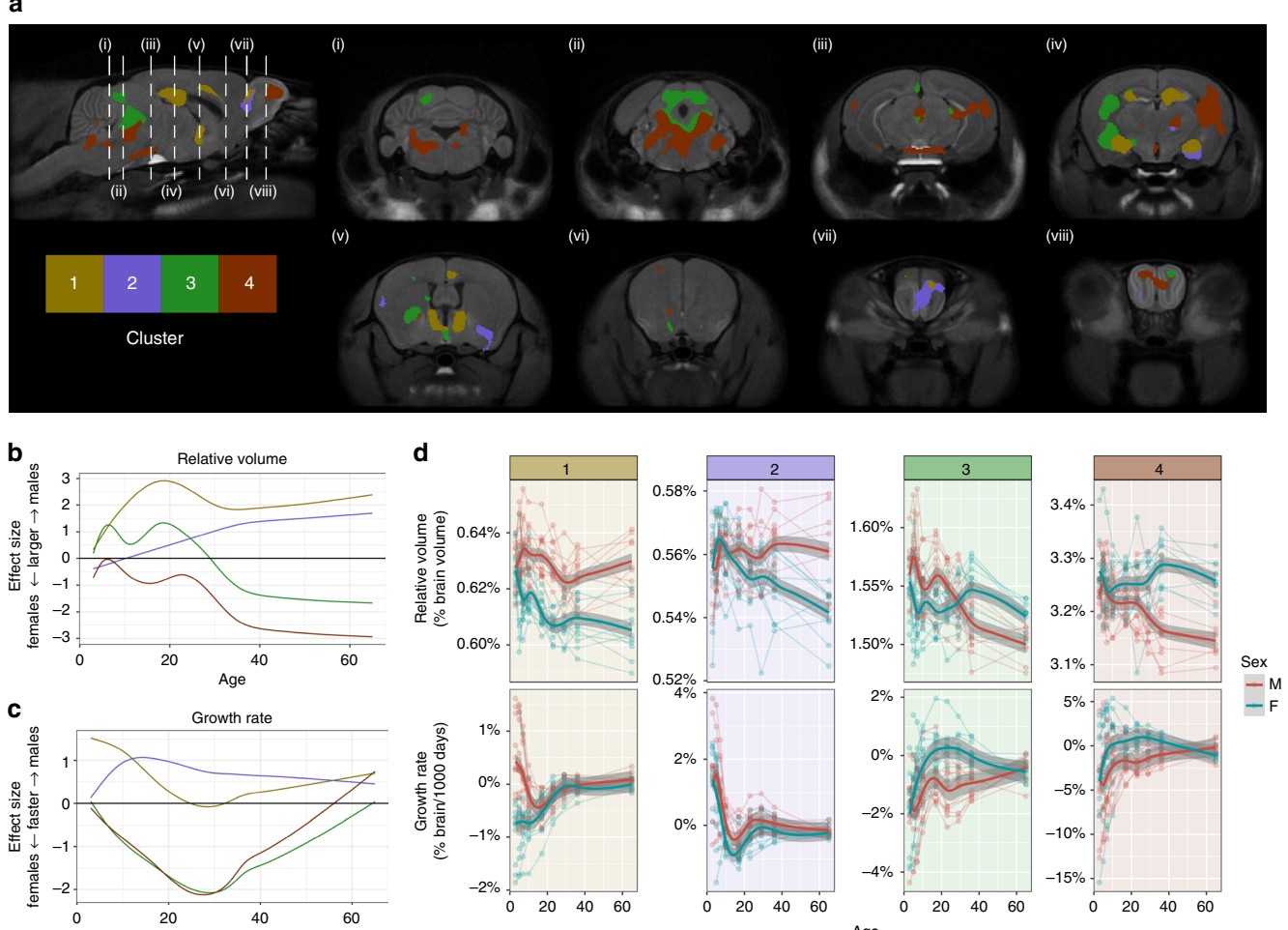

**Fig. 6** Coordinated growth of sexually dimorphic functional networks. Sexually-dimorphic voxels with similar effect sizes through time were clustered into 4 groups using $k$-means. **a** Results of the clustering analysis on sexually dimorphic voxels. Effect sizes (positive is bigger in males, and negative is bigger in females) of **b** Relative volumes and **c** Growth rate for the different clusters. **d** Average volume and growth rate in each cluster for each individual. Cluster 1 corresponds to regions larger in males and this dimorphism emerges early in development. Regions involved in the vomeronasal system, which processes pheromonal information, are found in this cluster. Cluster 2 also contains regions larger in males but the onset is more delayed. However, this cluster in early life shows strong bias in growth rate towards males. Parts of this cluster include the olfactory bulb and pallidum. Cluster 3 voxels trajectory switches from being larger in males in early life, to larger in females post-puberty. Parts of the sensory cortex and PAG belong to this cluster. Cluster 4 contains regions that are not sexually dimorphic in early life but become relatively larger in females over the course of development. This cluster includes association related areas such as parts of the central thalamus and temporal association cortex; and motor related areas such as parts of the hindbrain, cerebellum, caudoputamen, and motor cortex. Both Cluster 3 and 4 show growth rate bias towards females peaking around puberty

from the first 7 days of life, while females required data from the first 17 days of life to improve accuracy significantly. Thus, male neuroanatomy individualizes earlier than female neuroanatomy. This was also true ($P = 0.034$) when we computed root-mean-square-percent-difference (RMSPD) which is less biased against smaller brain structures (Supplementary Fig. 6). We found a similar pattern when predicting p29 ($P = 0.025$) and p65 ($P = 0.057$; Supplementary Fig. 7) time points, but it was no longer significant for p65. Furthermore, we also improved our model by using a random forest ($P = 0.030$) and introducing a covariate for whole-brain volume ($P = 0.036$) and found similar results.

## Discussion

Longitudinal MEMRI captures the emergence and development of sex differences in brain anatomy. We recapitulated known sex differences in the brain and identified new regions where development trajectories are influenced by sex. We discovered that

differences in neuroanatomical size emerge at different developmental times across sexes: relatively larger areas in the male brain emerge early in development, and relatively larger areas in females emerge peri- and post-puberty. Clustering regions based on shared sexually dimorphic development revealed networks of areas that are functionally connected. Examining spatial gene expression shows that these brain regions preferentially express genes on sex chromosomes and genes involved with sexual differentiation of the brain. Furthermore, individualization of the male and female brain occurs at different times in development, with male individualization occurring earlier.

Examining canonical sexually dimorphic areas with MEMRI largely recapitulates what is known about their development from rodent histology studies. In the BNST and MPON, significant differences in cell number and brain volume emerge by postnatal day 10, following an increase in rates of apoptosis in the female BNST and MPON in neonatal life[8]. Part of the MeA sex difference depends on differences in synaptic organization in its middle

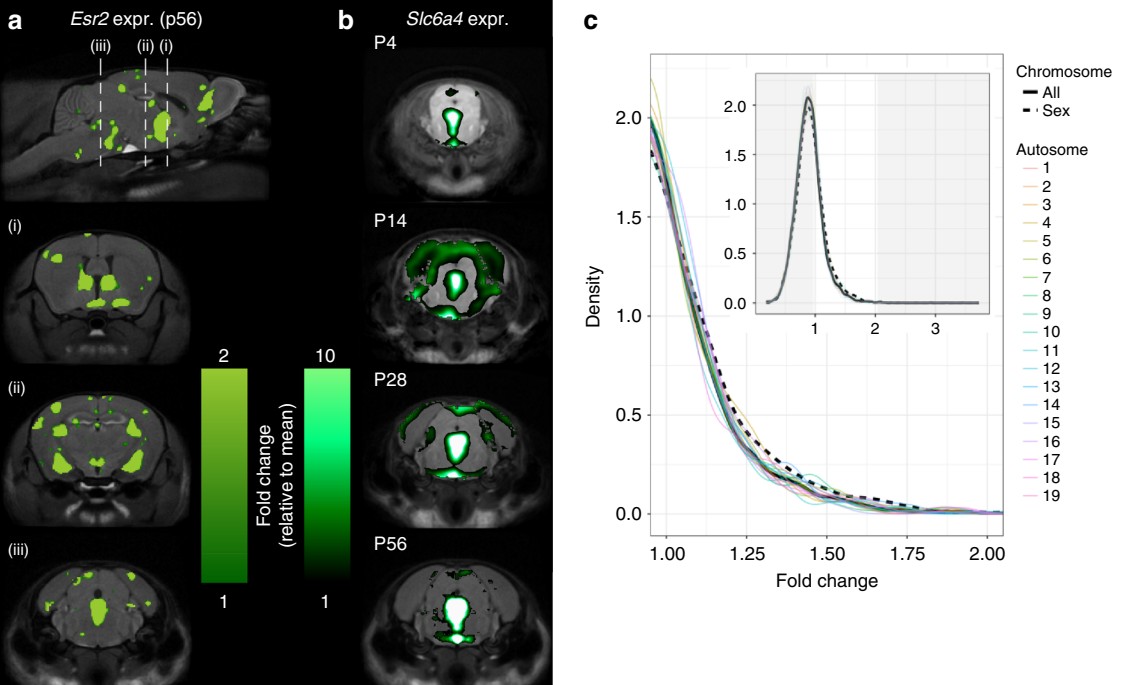

**Fig. 7** Gene expression patterns overlaid onto MRI results. **a** Spatial Gene Expression of Estrogen Receptor 2 (*Esr2*) in the adult mouse brain (ABI Dataset ID: 71670737) overlaid onto the MRI average in our study. *Esr2* and *Esr1* genes (Supplementary Fig. 5) were found to be preferentially expressed in regions with sexually dimorphic development. To measure preferential expression, we used a fold-change measure dividing the mean expression signal in regions with sexually dimorphic development by the mean expression in the brain. *Esr2* had fold-change of 2.9 and *Esr1* had a fold-change of 2.1. This was primarily driven by expression in Cluster 1, where the genes had fold-changes of 9.9 and 6.1, respectively. **b** *Slc6a4* expression throughout development. *Slc6a4* had the highest preferential expression (fold-change of 3.7) in sexually dimorphic regions primarily driven by expression in Cluster 3 (7.0 fold-change). Time course of Slc6a4 gene expression reveals peak expression in mid-brain and hind-brain between p4 and p28. **c** Genes on sex chromosomes have a higher likelihood of preferential spatial expression in regions with sexually dimorphic development. We computed the fold-change for all the genes in the Allen Brain Atlas and created the density plot marked by the solid line. The dashed line indicates the density plot associated with only the genes on the sex chromosomes and the colored lines represent genes on different chromosomes. In both the overall density plot (inset), and on the zoomed density plot, we see that genes on sex chromosomes have significant preferential expression bias (One-sample Kolmogorov–Smirnov test: $D^- = 0.052, P = 0.02, n = 730$)

layer around postnatal day 21[9,28]. This corresponds with our observations that differences emerge in pre-pubertal life. Slight discrepancies in timing of these sexual dimorphisms can likely be attributed to methodological differences, namely the histological study of specific nuclei, layers or areas versus whole structure volume measurements by MRI.

Repeated MEMRI scanning allows for longitudinal observation of the same individual and whole-brain volume correction, thereby increasing our sensitivity to detect subtle sex differences across time, and thus enabling us to identify major characteristics about the development of sex differences across the brain. First, developmental periods of relative change in male and female brains are different. Relatively larger areas in males predominate the brain in early, pre-pubertal life and relatively larger areas in females predominate the brain in later, post-pubertal life. These patterns mirror what is known about sex differences in age of onset for many psychiatric disorders: males are more likely to be diagnosed with disorders that are developmental in nature, and have an onset during childhood, while females are disproportionately diagnosed with disorders that have an emotional nature and emerge in adolescence and young adulthood[4]. Although, it should be noted that males are more prone to addiction and schizophrenia in adolescence, prior to when onset occurs in females[4,29]. MRI-detectable change in brain structure represents underlying cellular or molecular processes. Our results show that periods of relative neuroanatomical change differ across development for males and females, and thus may reflect a difference in cellular or molecular processes between males and females across developmental times. Periods of relative change in the brain can be considered, then, both as windows of brain development and windows of vulnerability when development goes awry[30]. The differences in timing of relative change in the brain across males and females serve as sex-specific opportunities where predispositions to certain stimuli, insults or processes can shift the likelihood of a particular behavior or psychiatric outcome to one sex or the other.

In neonatal life, males have high levels of circulating testosterone, which becomes aromatized to estradiol; this estradiol plays a crucial role in the sexual differentiation of the brain and modulates many cellular processes[14]. A vulnerability for neurodevelopmental disorders is conferred to males in early life if aberrant estradiol-related action occurs; for example, higher rates of autism-like behavior is linked to fetal testosterone levels[31]. Later in life, the presence of hormones affects both the male and female brain[32,33]; however, in females specifically, this change is linked with estradiol levels[34], since ovarian hormones further feminize the brain[35]. During adolescence, sex differences in many psychiatric disorders emerge[36], with female-biased psychiatric disorders increasing in prevalence. Furthermore, fluctuation of hormone levels during the menstrual cycle, pregnancy and parturition can affect mood and risk of depression[37]. The presence of these hormonal transitional periods may confer a unique vulnerability for mood and anxiety disorders to women[38]. The ability of MEMRI to detect sex differences in the timing of relative

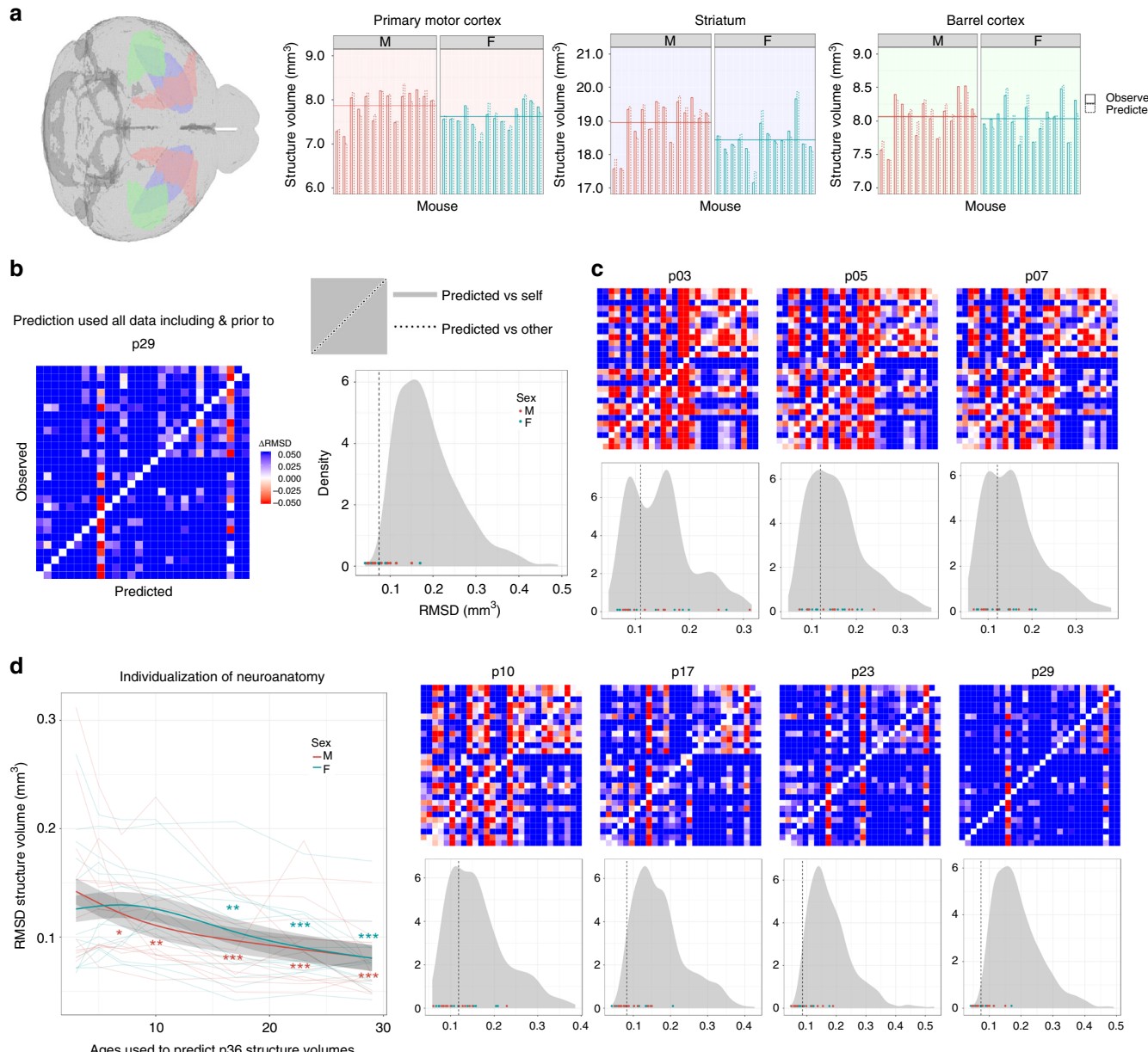

**Fig. 8** Prediction and individualization of neuroanatomical structure volumes. **a** Predicted and Observed p36 structure volumes for all subjects and three representative structures: primary motor cortex, striatum, and the barrel cortex. When predicting any subject at p36, the model was trained on all data excluding the predicted subject at p36 and all subjects at p65. Despite this exclusion, the model uses neuroanatomical information from earlier time points to predict structure volumes' variation from the average (horizontal line). **b** Matrix plotting the root-mean-square-difference (RMSD) between the observed structure volumes (rows) and the predicted structure volumes (columns) for each subject at p36. Prediction used all data prior to and including p29. Each column (Subject X prediction) was centered so that the diagonal RMSD (observation vs prediction for Subject X) is 0. Blue or red off-diagonal cells (observation Subject Y vs prediction Subject X) indicate RMSD greater or less than diagonal entry. Density plot shows that diagonal RMSD elements (values as points and median as vertical line) tend to be less than off-diagonal RMSD (grey distribution), indicating high prediction specificity (One-sample Kolmogorov–Smirnov test: $D^+ = 0.79894, P < 10^{-15}, n = 28$). **c** Models were trained on data closer to the time point of prediction (p36). For example, models corresponding to 'p10' were trained on p10 data and all earlier data (p3, p5, p7). As more data is included, model accuracy and specificity improved with significant specificity at p10 and above (One-sample Kolmogorov–Smirnov test: $D^+ = 0.34392, P = 0.001, n = 28$). **d** Prediction accuracy (RMSD) at p36 for each subject tends to improve over the course of neurodevelopment. When x-axis = X, subject data from ages ≤ X were used in the prediction. Trendlines denote patterns seen by sex and there is no significant difference in prediction accuracy between the sexes at any time (with shaded regions denoting standard error). Improvement in prediction accuracy (*$P < 0.05$, **$P < 0.01$, ***$P < 0.001$) indicates neuroanatomy individualization, which occurs significantly earlier in males (permutation test, $P = 0.025$)

neuroanatomical change provides insight on developmental windows that may be particularly important for understanding disorders that show sex bias.

The clustering analysis shows 4 groups of areas, each cluster characterized by a unique trajectory of sexually dimorphic development over time. Clustering by relative volume effect size allows us to provide insight on areas that cluster together as networks of connected structures that may share function[39]. Parts of the BNST, MeA, MPON and OB are featured prominently in Cluster 1. These areas are well known to be sexually dimorphic

and larger in males, and these sex differences depend on the presence of neonatal hormones[6,7,9,40–42]. These areas are part of a functionally and structurally connected network related to the vomeronasal system, which processes pheromonal information to mediate a wide range of social behavior in both sexes, and is particularly important for sexual behavior and aggression in males[43]. Many structures from the clustering analysis are structures involved in pain and analgesia processes[44,45]. There is robust clinical and laboratory evidence that indicates there are sex differences in pain and analgesia[1]. Although sex differences in pain sensitivity can vary with rodent species and even with strain[46], the underlying structures implicated in pain are highly conserved across mammalian species. Interestingly, these structures come from all 4 clusters, belonging to clusters that show relatively larger areas in males in early life, and relatively larger areas in females in later life. This may be a reflection of their sensitivity to hormones during both neonatal and adult life. Indeed, there is evidence suggesting that pain processes are sensitive to, and can be modulated by both neonatal gonadal hormones in males[47], later-life activational hormones in females[48,49], as well as both[50]. Cluster 4 describes areas of the brain that are slightly larger in females in early life, which then become enhanced post-pubertally. Parts of the sensory cortices and temporal association areas are featured prominently amongst this cluster. In humans, females have greater cortical thickness in many parts of the cortex, particularly in temporal and parietal areas[51]. These results correspond with our cortical thickness analysis that shows a prominent cluster of cortical areas that is larger in females and emerges later in life (Supplementary Fig. 4).

Our study, although thorough in neuroanatomical characterization of sex differences across development, does not directly address the functional relevance of these dimorphisms. However, there are robust examples of neuroanatomical sex differences which directly relate to behavioral sex differences, that also change in a corresponding fashion upon hormonal manipulation; for example, the MPON and male- and female-typical sexual behavior[52]. Furthermore, recent research points to the utility of information about typically developing male and female brain anatomy in predicting the presence of psychiatric disorders that show sex differences, such as autism[53,54]. Our findings can inform future work that seeks to further elucidate sexually dimorphic structure-function relationships of the brain.

Overlaying gene expression maps onto our images provides insight into the underlying causes that drive our MRI results. Genes *Esr1* and *Esr2*, which encode for estrogen receptors, as well as *Slc6a4*, which encodes the serotonin transporter, were amongst the most preferentially expressed genes from our analysis. *Esr1* and *Esr2* are known to be involved in sexual differentiation of the brain and behavior[14], while variants of *Slc6a4* have been implicated in disorders that show sex bias in type and in age of onset[55]. Expression of these genes also changes across development and are sensitive to hormones[56]. Such comprehensive gene expression datasets currently only exist for males and not females[26]. However, since these genes were preferentially expressed in sexually dimorphic areas, further investigation of how these candidate genes are expressed in females, and how they drive sexual dimorphisms is warranted. Additionally, more direct investigations — such as in-depth histology — of the cellular underpinnings of our mesoscopic neuroanatomical changes would be useful. Our results provide indications of candidate genes and areas to explore in further detail.

Non-invasive longitudinal data over the course of development provides strong leverage to answer questions of individualization of the brain. This individualization has been explored in the context of variations in task fMRI activation maps in adult humans[27], and exploratory behavior in enriched environments in

adult female mice[57]. We found that individualization of the brain occurred earlier in males than in females, meaning that males achieved their mature neuroanatomical phenotype earlier than females. This method benefits from a greater degree of genetic and environmental control available for mouse studies versus humans. However, with the advent of ever-bigger high-quality human datasets spanning neurodevelopment, this method could be useful in characterizing normal human neuroanatomy individualization, and perhaps earlier detection or prediction of neuroanatomical pathologies associated with disorders — especially those that show sex differences.

There are several limitations concerning cross-species differences in development, endocrinology and brain anatomy to acknowledge. The rodent brain is less mature than the human brain at birth[21]; thus our neonatal findings shed light on the human prenatal brain. Investigating this period of development is important though, as sexual differentiation of the human brain begins in the latter half of pregnancy[58]. Furthermore, there is evidence that some processes that underlie certain psychiatric disorders in postnatal life occur during prenatal development, such as autism[59]. Unlike mice, humans and some upper-level primates undergo adrenarche prior to puberty. Studies that examine the impact of adrenarche on the brain are both limited and show conflicting results, although there is some evidence suggesting that dysregulation of adrenarche timing is related to future mental health symptoms, and may affect males and females differently[60]. We cannot overcome this in mouse studies, but it is worth considering this endocrine event which may have additional activational effects in humans. The human cortex is highly folded and has a more developed prefrontal cortex that contains areas such as the dorsolateral prefrontal cortex (DLPFC) which is virtually nonexistent in the mouse. Prefrontal cortical areas are important due to their roles in complex behaviors and symptoms of many psychiatric disorders. This inter-species difference in cortical structure and function is difficult to reconcile, although it has been demonstrated that certain rat cortical areas contain features that resemble the primate DLPFC[61]. However, the cortex does not operate in isolation and is connected to subcortical structures, whose interconnections and functions have been highly conserved across mammalian species.

In summary, we have used longitudinal MEMRI to study anatomical sex differences across the whole mouse brain to characterize both known and novel male-female differences throughout postnatal development. We have shown that MEMRI is a robust method for detecting neuroanatomical sex differences and their time courses. We found distinct periods of relative developmental change in males and females, where increased male growth predominates in early life and increased female growth predominates in post-pubertal life. By clustering areas in the brain based on shared sexually dimorphic development, we have revealed networks of areas that are functionally connected and mediate sexually dimorphic processes. Furthermore, we have found that as the brain individualizes across development, the male brain does so at an earlier time compared to females. These findings demonstrate the power of whole-brain in vivo MEMRI for examining the development of sex differences, and the importance of studying sex differences across the whole brain across a comprehensive temporal context that begins in neonatal life.

## Methods

**Animals and non-imaging procedures.** Male and female C57BL/6J mice were scanned longitudinally across 9 postnatal day (p) time points: p3, p5, p7, p10, p17, p23, p29, p36 and p65. Number of mice at each time point are as follows: p3 (n = 13 males, n = 15 females), p5 (n = 15 males, n = 14 females), p7 (n = 14 males, n = 14 females), p10 (n = 14 males, n = 14 females), p17 (n = 14 males,

n = 14 females), p23 (n = 14 males, n = 12 females), p29 (n = 11 males, n = 11 females), p36 (n = 14 males, n = 14 females), p65 (n = 9 males, n = 11 females). Each individual mouse was scanned at all time points; discrepancies in mouse number at some time points was due to occasional scanner issues that caused certain scans at time points to be excluded. There was a minimum of 9 scans per sex per age. In a cross-sectional statistical analysis, this minimum is enough to recover 3% volumetric group differences at a significance level of 5% and a power of 80%[62]. By comparison, we saw volumetric differences of 7% in the BNST and 6% in the MPON at p65. Number of mouse pups in each litter was reduced to 6 to ensure equal manganese intake by pups through maternal milk. Because manganese is administered to neonatal mice through maternal milk, non-scanned littermates were also exposed to manganese for the first 10 days of life. The ratio of male to female mice in each litter was kept equal. To differentiate neonatal pups, mice received black ink tattoos on their paws at p2 (AIMS Lab Animal Tattoo Kit, AT-3 General Rodent Tattoo System).

Two pups from each cage (one male and one female) were used for longitudinal scanning. The pups were randomly selected by the experimenter with the only restriction being that they were of dissimilar sex—however, no formal randomization procedure was used. Experimenter was not blind to sex. Both scanned and non-scanned littermates were weighed either on the day of scanning, or the day prior to scanning as a measure of overall growth throughout the experiment. Mice were weaned at p21 and separated into cages by sex. Post-weaning, mice were assessed for puberty daily. First occurrence of preputial separation after weaning was used as an indicator of puberty for male mice[63], and first occurrence of vaginal opening after weaning was used as an indicator of puberty for females[64]. Weight at puberty was also recorded. Mice were housed in cages with up to 4 mice, and maintained on a 12-hour light/dark cycle, with ad libitum access to food and water.

At postnatal day 66, blood was collected for hormone level measurements and organs (gonads and uteri) were dissected out of scanned and non-scanned mice, to be weighed. Mice were anaesthetized with 1-4% isoflurane in air. While under anesthesia, blood for plasma was collected via cardiac perfusion by opening the thoracic cavity and drawing blood from the left ventricle. Mice then underwent cervical dislocation, and ovaries and uteri were dissected from female mice, while testes were dissected from male mice. Dissected tissues were placed in a dish with phosphate-buffered saline and excess fat was removed from the tissues under a light microscope. Before weighing, tissues were blotted on a Kimwipe to remove excess liquid. Tissues were weighed on an analytical scale accurate to 0.1 mg. Blood samples were sent to The Endocrine Technologies Support Core at the Oregon National Primate Research Center (Beaverton, OR). Barring samples of insufficient size, all samples were analyzed for estradiol, testosterone, follicle-stimulating hormone (FSH) and luteinizing hormone (LH). Estradiol and testosterone levels were measured with extraction-chromatography RIA, with an intra-assay CV of 14.7% and 3.3%, respectively. Assay sensitivity was 5 pg/ml for estradiol, and 0.2 ng/ml for testosterone. LH and FSH were analyzed with RIA, with an intra-assay CV of 9.9% and 3.0%, respectively. All experiments were approved by The Centre for Phenogenomics Animal Care Committee.

Growth was compared between scanned and non-scanned animals by running two linear mixed-effects models: both models had fixed effects of sex, age (approximated as a quintic spline), and their interaction, with a random effect of growth for each mouse. One model had an additional fixed effect of type (scanned or non-scanned) and type-sex interaction. The two models were then compared with a likelihood ratio test to assess whether scanning affected growth of mice or if the effect had significant sex-bias. A quintic spline was chosen as the optimal model for weight versus age effects by minimizing Bayesian Information Criterion[65], although results were similar when we chose cubic and quadratic models as well. Differences between scanned and non-scanned mice in weight and puberty onset, hormone levels, and organ weights were analyzed using linear models.

We did not correct data comparing scanned and non-scanned mice for multiple corrections, as the only significant differences were for weight at puberty. Upon correcting for multiple comparisons using FDR[66], these differences disappear, further illustrating the similarity between scanned and non-scanned mice.

**In vivo imaging**. Up to 7 mice of the same age were scanned simultaneously in vivo. 24 hours prior to the scan, mice received a 0.4 mmol/kg dose of 30 mM manganese chloride ($MnCl_2$) solution. For mice 10 days and younger (neonates), $MnCl_2$ was provided through maternal milk by injecting mothers 24 hours prior to the scan. Mice 17 days and older received intraperitoneal injections directly 24 hours prior to the scan. Throughout the scan, bore temperature was maintained at 29 °C, and a steady stream of 1-2% isoflurane was used to keep the mice anaesthetized. Respiration was monitored throughout the scan. Respiratory pillows were used for mice 17 days and older; self-gated signals from a modified 3D gradient-echo sequence[67] provided respiratory motion information for neonatal mice which were too small for respiratory pillow use.

A multi-channel, 7.0 Tesla, 40 cm diameter bore magnet MRI scanner (Varian Inc. Palo Alto, CA) was used to acquire images of mouse brains. Parameters of the scan are as follows: T1-weighted, 3D gradient echo sequence, TR = 26 ms, TE = 5.37 ms, flip angle = 37°, field-of-view = 77 × 20 × 20 mm, matrix size = 854 × 224 × 224, number of averages = 5, total acquisition time = 1 hour and 40 minutes, isotropic resolution = 90 $\mu$m. Post-scanning, mice were transferred to a heated cage

for 5–10 minutes in order to recover from anesthesia, and then returned to their home cages.

**Longitudinal registration**. Image registration allows quantification of anatomical differences between images. For a group of images, this procedure results in a transformation that maps every point in one image to corresponding points in the other images. Thus, the differences between the images are captured by this transformation. Our procedure for image registration is composed of an affine registration, followed by a series of non-affine registrations. The affine registration applies global translation, rotation, scaling, and shearing to align images. Information regarding global deformations (i.e. the overall brain sizes) are stored in these transformation models. The non-affine registration creates a vector field that maps every point in one image to another and provides information about localized deformation. Illustrations of these deformations are available in Supplementary Methods and Supplementary Fig. 11.

The Pydpiper toolkit[24] extends the processes described above to group-wise registrations (described in detail by[62]). Pydpiper takes multiple images as inputs, and outputs a consensus average, as well as linear and non-linear transforms that map the consensus average to all input images.

We modified the registration process to accommodate longitudinal data using a two-level approach. In Level 1, group-wise registration was performed on each age. For example, all the p3 brain images were registered together to create a p3 brain average, all the p5 brain images were registered together to create a p5 average, etc. The results of this level are consensus averages of each age and their appropriate transforms to the input images; however, the results do not capture deformations across time. Time-dependent deformations are captured by Level 2 of the registration, where the consensus average from each time point is registered to the average from the following time point (p3 average registered to p5 average, p5 average to p7 average, etc). The final step in the registration is to concatenate the transforms from both levels so all images can be mapped to the p65 consensus average brain in a single interpolation step. For example, to align the image of a p29 subject brain to the p65 average brain, the following transformations are concatenated: p29 subject to p29 average, p29 average to p36 average, p36 average to p65 average, where the first transformation is obtained in Level 1 and the remaining from Level 2. The concatenated transform can be used to resample the image of a p29 subject brain to the p65 average space.

The two-level registration procedure creates transformations that map the p65 consensus average to every image. As described earlier, each transform contains a global transformation (derived from the affine registration) and local transformations (derived from the non-affine registration). We used deformation-based morphometry to analyze these transformations. First, the transformation vector field is converted into a Jacobian determinant scalar field. Each point in the consensus average has a scalar value associated with it, characterizing the degree to which volume elements (voxels) had to grow or shrink to map to the individual images. Thus, the volumetric differences between images are captured by the Jacobian determinants. Determinants of the total transformations (global + local) are called absolute Jacobian determinants as they characterize the true volumetric differences between the images and the p65 consensus average. Relative Jacobian determinants are the determinants of only the local transformations and characterize volumetric differences with the overall effect of brain size removed. The advantage of relative Jacobians is that they can eliminate variability due to overall size, and can reveal relative neuroanatomical differences otherwise difficult to detect. Illustrations of absolute and relative Jacobian determinants are available in Supplementary Methods and Supplementary Fig. 12.

We took the logarithm of absolute and relative Jacobian determinants prior to statistical analysis. Regions with negative log determinants suggest that the region is smaller than the consensus average, while regions with positive log determinants suggest that the region is larger.

To perform volume analysis on structures, we registered an MRI-atlas[25] onto the p65 average. Since subject images from all ages were registered to the p65 average, aligning the MRI-atlas to the p65 average enabled automated quantification of structure volumes over time. We obtained PAG and BNST segmentations from the MRI-atlas[25]; MPON and MeA segmentations were obtained from a modified atlas in which these two structures were manually segmented. We also assessed for biases arising from choosing the p65 average as the registration consensus average (Supplementary Methods and Supplementary Fig. 9,10,13,14) and found them to be minimal and indiscriminate of individuals and sex.

**Statistical Analysis**. Statistical analysis was performed using linear mixed-effects models using the lme4 package[65]. By incorporating fixed and random effects, these models are appropriate for data from the same subject over time and enable more powerful analysis of longitudinal studies. The model formula is given below:

$$y_{ij} = \sum_{p=1}^{P} \alpha_p X_{pij} + \sum_{r=0}^{R} \beta_{ri} Z_{rij} + \varepsilon_{ij} \qquad (1)$$

In (1), for a particular mouse $i$ measured at a particular time point $j$, $y_{ij}$ is the response variable we want to model, $P$ is the total number of fixed effects, the matrix $X$ represents our fixed effects with $X_{pij}$ and $\alpha_p$ being the value of the $p$th

fixed effect and its coefficient, $R$ is the total number of random effects, the matrix $Z$ represents our random effects with $Z_{rij}$ and $\beta_{ri}$ representing the value of the $r$th random effect for the $i$th mouse and its coefficient, and $\varepsilon_{ij}$ represent the residuals assumed to be independent and normally distributed. Wherever possible, we qualitatively checked if the residuals of the linear mixed-effects models were normally distributed.

The response variable $y_{ij}$ can represent any volumetric measurement, and the predictors are flexible enough to handle the various analyses we performed. When analyzing structures, $y_{ij}$ represents the structure volume; and when analyzing voxels, $y_{ij}$ represents the relative log determinant at that voxel.

To perform significance testing for a set of $q$ effects (i.e. $q = \{p_1, p_2, ..., p_Q\}$), we fit the data with both the full model (1) and a similar model without the particular effects:

$$y_{ij} = \sum_{p=1, p \neq q}^{P} \alpha_p X_{pij} + \sum_{r=0}^{R} \beta_{ri} Z_{rij} + \varepsilon_{ij} \tag{2}$$

We used the standard likelihood-ratio test to assess whether the full model (1) fits the data significantly better than the partial model (2). Given data, the test statistic $D$ can be computed from the maximum likelihood of the full model $L_f$ and the partial model $L_p$:

$$D = -2 \ln \frac{L_p}{L_f} \tag{3}$$

According to Wilks' theorem[65], Equation (3) follows the $\chi^2$ distribution with degrees of freedom being equal to $Q$—the difference between the number of parameters in the full model and the partial model. We can thus compute p-values to measure the significance of the $q$ effects. Finally, we used false discovery rate[66] to correct for multiple comparisons.

**Sexual Dimorphisms in Canonical Structures**. To test the significance of sex on the structure volumes, we fit two models. Model 1 contains fixed effects sex $s$ and time point $\tau_k$ (where $k$ goes from 1 to 9 for each of our experimental time points), as well as interaction terms. Model 1 also had a random intercept for each individual mouse $\beta_i$. Based on the general equation (1), the formula for Model 1 is given below:

$$y_{ij} = \alpha_1 + \alpha_2 s_i + \sum_{k=1}^{9} \alpha_{k+2} \tau_{kij} + \sum_{k=1}^{9} s_i \alpha_{k+11} \tau_{kij} + \beta_{0i} + \varepsilon_{ij} \tag{4}$$

Model 2 was identical to Model 1 but with no effect of sex and no interaction terms. The formula is:

$$y_{ij} = \alpha_1 + \sum_{k=1}^{9} \alpha_{k+2} \tau_{kij} + \beta_{0i} + \varepsilon_{ij} \tag{5}$$

The significance of sex can be computed from the likelihood ratio of the two models. Model 1 also provided standard error estimates, which were used to shade the appropriate regions in figures. To estimate the timing of when sexual dimorphisms emerged, we applied a Satterthwaite approximation[65] to estimate statistical degrees of freedom (df) and computed p-values for the estimates in Model 1 associated with sex differences. The earliest significant time point for sex differences in absolute volumes of canonical structures in Fig. 3 are p10 for MeA ($t = 2.7$, df = 213, $P < 10^{-2}$), p10 for BNST ($t = 4.0$, df = 213, $P < 10^{-4}$), and p10 for MPON ($t = 2.4$, df = 217, $P = 0.02$). In relative volumes, the corresponding earliest significant time point are p7 for MeA ($t = 2.6$, df = 17, $P = 0.02$), p5 for BNST ($t = 3.1$, df = 18, $P = 0.01$), and p5 for MPON ($t = 3.3$, df = 4, $P = 0.03$). We cross-sectionally ran an F-Test for equality of variances and did not find a significant difference in variance between the sexes at any time point using either relative or absolute volumes (variances reported in Supplementary Table 1).

**Sexual Dimorphisms in Voxels**. The effect of sex on voxel determinants was analyzed in a similar way by first fitting two models for every voxel in the brain. Model 1 predicted the relative log determinants at that voxel (determinants after correcting for different whole-brain sizes) using fixed effects of sex, age ($t$), and their interaction and random intercept for each mouse. Growth was modeled as a linear function of age.

$$y_{ij} = \alpha_1 + \alpha_2 s_i + \alpha_3 t_{ij} + \alpha_4 s_i t_{ij} + \beta_{0i} + \varepsilon_{ij} \tag{6}$$

Model 2 was identical to Model 1 but with no effect of sex and no interaction between sex and age.

$$y_{ij} = \alpha_1 + \alpha_3 t_{ij} + \beta_{0i} + \varepsilon_{0j} \tag{7}$$

Likelihood-ratio statistic was computed for every voxel comparing the fit between Model 1 and Model 2, and this statistic was used to compute the significance of sex. We experimented with different growth models—growth modeled as a linear and quadratic function of age and random effect of growth for each individual mouse

$\beta_1 t_{ij}$—and observed similar regions of the brain exhibiting sexual dimorphism. Supplementary Discussion details additional models we tested for consistency: spacing time points equally (Supplementary Fig. 15), removing the p65 time point (Supplementary Fig. 16), and modeling absolute Jacobian determinants (Supplementary Fig. 17).

**Age-Centered models**. To visualize sexual dimorphisms at a particular age $t'$, we used an age-centered model. Similar to the model in (6), the age-centered model references all ages to $t'$, so that the time-independent fixed sex effect ($a_2$) represents the age-specific difference.

$$y_{ij} = \alpha_1 + \alpha_2 s_i + \alpha_3 \left(t_{ij} - t'\right) + \alpha_4 s_i \left(t_{ij} - t'\right) + \beta_{1i} + \varepsilon_{ij} \tag{8}$$

For each voxel, we then extracted the coefficients associated with sex at this age of interest and assigned significance values to the sex effect using the Satterthwaite approximation[65]. False discovery rate was used to correct statistics for multiple comparisons.

**Sexual dimorphism clusters and gene expression**. To examine developmental patterns amongst sexually dimorphic areas, $k$-means clustering was used to find groups of voxels that show the same pattern of sexual dimorphism across time. Sexual dimorphism was defined by computing effect size—Cohen's $d$ (9)—of the relative Jacobian determinant (positive being bigger in males and negative being bigger in females). Voxels that showed significance of sex at a false discovery rate of 10% from the linear mixed-effects modeling analysis were included. Supplementary Fig. 8 shows that 4 clusters was appropriate for this data.

$$d = \left(\mu_1 - \mu_0\right) \sqrt{\frac{n_1 + n_0 - 2}{n_1 \sigma_1^2 + n_0 \sigma_0^2}} \tag{9}$$

where $n_i, \mu_i, \sigma_i^2$ are the number, mean, variance of values in $i$ subjects and $i = 0$ corresponds to females and $i = 1$ corresponds to males

Growth rate was estimated by fitting the relative determinant at every voxel for every individual with natural spline functions of age, then differentiating the fitted function with respect to age. At every voxel, the order of the fitted natural spline was determined by finding which order minimized the Akaike Information Criterion[65]. Detailed in Supplementary Methods, we used the Allen Brain Institute's gene expression dataset[26] to identify genes spatially enriched in our clusters[68]. Preferential spatial expression of a gene was measured using a fold-change measure: mean expression signal in an ROI (region of interest) divided by mean expression signal in the whole brain. Fold-change greater than 1 indicates that a gene is preferentially expressed in the ROI and fold-change less than one indicates that a gene is preferentially expressed outside the ROI. We estimated fold-change for every gene and tested whether genes on sex chromosomes were more likely to have higher fold-changes using the Kolmogorov–Smirnov test.

**Cortical thickness**. The Pydpiper pipeline was used to segment the cortex[24] for all nine age-consensus averages, and these cortical segmentations were then transformed to every subject image using appropriate transformations from Level 1 of the two-level registration (Detailed in Supplementary Methods). The end result of these procedures is a cortical segmentation for every subject and at every time point. Laplace's equation was then solved for all subject images and at every time point, with the inner and outer cortical surface having different potentials, thereby defining the boundary conditions. A property of Laplace's equation is that streamlines are always perpendicular to equipotential surfaces. Taking advantage of this property, for each point on the cortical surface, the thickness was defined as the length of the streamline connecting the inner and outer cortical surface. Using the transformations from both levels in the two-level registration, we mapped any point on the cortical surface of the consensus average mouse to that same point at any age and for any subject.

**Post-development neuroanatomy prediction**. We obtained structure volumes for all 182 bilateral structures in our atlas, for every subject and at every time point. When predicting the volume of a structure from subject $i$ at time $t$, we excluded all data that belonged to subject $i$ at time $t$ to form the training set. We modeled each structure independently using weighted linear mixed-effects models and trained these models on the training set (Supplementary Methods). Then, we used the trained models to predict the excluded data of subject $i$ at time $t$ and compared the predicted data for any structure $j$ ($\hat{y}_j$) to the observed data for the same structure ($y_j$).

$$\text{RMSD} = \sqrt{\frac{1}{182} \sum_{j=1}^{182} \left(\hat{y}_j - y_j\right)^2} \tag{10}$$

$$\text{RMSPD} = \sqrt{\frac{1}{182} \sum_{j=1}^{182} \left(\frac{\hat{y}_j - y_j}{y_j}\right)^2} \tag{11}$$

Model accuracy was evaluated using RMSD and RMSPD—the latter of which is less biased against small structures. RMSPD also corrects for whole-brain volume as normalization to brain volume ($V$) results in the substitutions $\hat{y}_j \rightarrow \hat{y}_j/V$, $y_j \rightarrow y_j/V$, which leave Equation (11) unchanged.

We also trained two additional models to check for consistency. The first model predicted structures after co-varying for total brain volume. The second model used a random forest to predict structure volumes from other structures at earlier times. All models showed similar results.

Individualization of neuroanatomy was demonstrated by withholding information about the predicted subject prior to model training. We computed RMSD as a function of the accessed time point information $x$; that is, when $x = 7$, the model must make its prediction on subject data from time points p7 and earlier (p5 and p3), and does not have access to data from time points after p7. To test if models made specific predictions, we computed the RMSD between the prediction for subject $i$ and the observation for subject $i$ (predicted vs self), and compared it to the RMSD between the prediction for subject $i$ and observations for other subjects $\neq i$ (predicted vs other). Repeating this for all subjects, we compared the RMSD values from predicted vs self and RMSD values from predicted vs other using Kolmogorov–Smirnov test.

As the accessed time point $x$ increases, model predictions become better and RMSD decreases. To test for sex differences in the timing of individualization of neuroanatomy, we fit a linear mixed-effects model to the RMSD versus the accessed timepoints $x$. The model had fixed effects of accessed time point, sex, their interactions and random effect of mouse. For both sexes, RMSD significantly decreased with time (the degrees of freedom calculated using Satterthwaite approximation[65]). We subsetted a time window over which individualization occurs: the lower bound being the first time point where at least one sex had a significant RMSD decrease ($P<0.05$) and the upper bound being the first time point where both sexes had a very significant RMSD decrease ($P<0.01$). For each individual, we Z-transformed the RMSD values (subtracted their mean value over time and normalized to the standard deviation). Then, we computed the average Z-RMSD values for males and females in the time window. The difference between these two values is related to sexual dimorphisms in the timing of neuroanatomy individualization between the sexes. To estimate the significance of this difference, we performed a permutation test by shuffling the sex labels of our data, recomputing the average Z-RMSD for males and females, and computing the difference. P-values for the permutation test were defined by finding what fraction of the 10,000 permuted differences exceeded or were equal to the true difference between average Z-RMSD male and female values.

**Code availability**. Code for image registration (https://github.com/Mouse-Imaging-Centre/pydpiper) and statistical analysis (https://github.com/Mouse-Imaging-Centre/RMINC) is freely available online.

**Data availability**. Data are available upon contacting the corresponding author.

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

## Acknowledgements

This work was supported by funding from the Brain Canada Foundation, Canadian Institute of Health Research, and Ontario Brain Institute. L.R.Q. was supported by a Restracomp Fellowship funded by the Hospital for Sick Children. D.J.F. was supported by an Ontario Graduate Scholarship and a Doctoral Postgraduate Scholarship from the Natural Sciences and Engineering Research Council of Canada. K.U.S. and D.H.T. were supported by NIH grant R01NS038461. We thank Sharon Portnoy for MRI sequence development assistance; Christina Corre and Ariane Metcalfe for colony management and technical assistance; Matthijs van Eede and Benjamin Darwin for help with computation pipelines; Chris Hammill for help with RMINC; and Lindsay Cahill for helping to edit the manuscript.

## Author contributions

L.R.Q. designed and performed experiments, analyzed data, wrote manuscript. D.J.F. performed experiments, analyzed data, created figures, wrote manuscript. K.U.S. and D. H.T. provided information on neonatal imaging. J.D. designed and built holders and scanning array. B.J.N. provided guidance with MRI pulse sequence development and image reconstruction. J.A.F. provided details on developmental milestone behavioral testing. M.R.P. and J.P.L. designed experiments and supervised project. All authors edited manuscript.

## Additional information

**Competing interests:** The authors declare no competing interests.

