## [Peer Review File · Nature Communications]

Reviewers' comments:

Reviewer #1 (Remarks to the Author):

This paper describes a study of sexual dimorphisms in the developmental trajectories of mouse brain structure as studied with longitudinal acquisitions of Manganese-Enhanced MRI. Comparisons are made at the level of atlas-based brain regions and at a voxelwise level, and the regions of interest are shown to be associated with expression of genes relevant to sexual development.

The paper is a generally well-written description of a well-designed study that includes several elegant, novel, and sophisticated approaches. There are a few conceptual issues, some clarifications that should be made, and a few issues with respect to wording, which are addressed below.

Conceptual issues:

1. The sentences in the first paragraph beginning "Males have a predisposition for disorders that have early onset during childhood.... [4]" are not without some controversy and not without counter-examples (e.g., schizophrenia). This should be clarified.

2. Importantly, much is made about the potential for this work to contribute to knowledge about sex differences in humans and in human disorders. However, it must be acknowledged that the considerable cross-species differences in the endocrine events of puberty may limit the interpretations with regard to human development. First is the obvious example that rodents are born much more mature than humans. And the endocrinology is considerably different (e.g., adrenarche in humans is a primarily human event). These and related issues should be discussed as limitations.

3. Similarly, the mouse brain, particularly the cortex, differs from that of humans in critical respects, and in regions critical for development and for disease. e.g. the dorsolateral prefrontal cortex, while uniquely developed in humans, is virtually nonexistent in the mouse. This limitation, too, should be discussed.

Additional points:

4. Section 3.1: "Weight at puberty was significantly affected [by scanning]." Even though the authors found no interaction between scanning and sex, possible implications should be addressed and discussed since adipose tissue can contribute significantly to the circulating pool of estrogens.

5. The pN terminology used to describe postnatal age should be described in the last paragraph of the introduction, immediately following the descriptions of the 9 time points, before using the term p65, for example, in the results section.

6. The relationship of these time points in the mouse to development in the human should be made clear.

7. Grammatical errors, such as "These pattern mirrors..." (3rd paragraph in Discussion) should be corrected. Also, e.g., in the heading of 3.4, "reflect" should be "reflects"

8. When discussing the longitudinal registration, references to "linear registration" should technically be "affine registration".

9. How were statistical corrections carried out for the multiple analyses, multiple time points, and multiple regions/clusters examined?

10. In section 5.4.2 (Methods), description of the likelihood ratio should include an appropriate reference to Wilks' Theorem.

11. The caption for Supplemental Figure 6 indicates that models were approximated with 6th order natural splines, while the methods section seems to refer to 3rd order splines were used. Was one used for absolute measures and the other for relative measures? The authors should clarify.

12. The description of the multi-level ANTS-based normalization technique is relatively clear, but the description should specify whether the concatenated ANTS transforms were performed in a single interpolation step. On a potentially-related note, the use of the final time point image as a reference is somewhat counter to recent recommendations (Reuter, 2011 and others) for avoiding asymmetry-induced bias by using central time point images as reference images. The authors should address the issues of interpolation and whether their approach might introduce any bias.

13. Although the asserted sexually dimorphic differences in time courses of the emergence of relative regional size differences are intriguing and potentially represent a valuable contribution to the field, the statistical strength of these findings, the main conclusion, is difficult to judge without further clarification about the uncertainty measures shown in the developmental plots in Figure 6. Do the plots in part D represent confidence intervals, standard deviation, standard error, or something else? The authors should clarify this information and indicate statistical significance of the differences between males and females at various time points, thereby allowing the reader to make informed judgements about the strength of these key results.

Reviewer #2 (Remarks to the Author):

Qiu et al. Nature communications

"Mouse MRI shows brain areas larger in males emerge earlier than those larger in females"

The paper by Qiu and colleagues describes the use of manganese-enhanced MRI (MEMRI) to map both canonical and novel regions of the mouse brain, which show sexually dimorphic development and relate these to spatial measures of gene expression, thus providing clues to possible mechanisms. Based on the data presented the authors make the following major claims:

- (1) MEMRI is a useful method to capture developmental neuroanatomical differences without adverse effect
- (2) MEMRI detects sex differences in the volume of three brain regions (BNST, MeA, MPON), previously reported to be sexually dimorphic, providing validation of the sensitivity of the method
- (3) Brain regions that are larger in males tend to predominate early in development, in the neonatal and prepubertal period, whilst regions that are larger in females emerge post-puberty
- (4) This can be refined to reveal 4 distinct trajectories based on k-means clustering showing distinct spatiotemporal differences in volumes of various brain regions between males and females
- (5) These data overlap with prior studies of sexual dimorphism (e.g. using 4-core genotype mice)
- (6) Brain regions that show sexual dimorphisms in volume are enriched in expression of genes known to be involved in the sexualisation of the brain, for example *Esr1*, *Esr2*, but also novel genes such as *Slc6a4* and *Tph2*, based on spatial normalization of the MRI data to the Allen Brain atlas.

These claims are novel, at least in terms of mouse studies, since the majority of such studies in mice have only examined sex differences in adults, or from adolescence and not included earlier time points during development. The data presented herein are therefore novel for this field.

In terms of human MRI studies of sexual dimorphism in anatomy, the concept that some aspects

of neuroanatomy, such as cortical thickness are larger in infant males as compared to females at 2 years of age (Li et al., *J Neuroscience*, 2014; doi: 10.1523/JNEUROSCI.3976-13.2014) has already been suggested. Other studies also provide evidence for early gender differences in volume, cortical thickness and surface area (e.g. Wierenga et al., *Neuroimage* 2014; doi: 10.1016/j.neuroimage.2013.11.010). I would suggest a more exhaustive search of the extant literature to acknowledge that the idea that sexual dimorphisms might arise early based on human studies of infants, which are few, but do exist as show above is warranted to justify the novel aspect of these new mouse data. Despite these points, the claims are however in general, adequately discussed in the context of prior literature.

In particular, where these mouse data do bring novelty to the human field is in the identification of many more brain regions that show sexual dimorphism and clustering these into distinct trajectories. One difficulty here is how best to translate these findings to humans, since as the authors acknowledge "MRI allows investigation of the whole brain, but lacks the ability to detect smaller sexually dimorphic nuclei". This conceptual and technical barrier should be considered in the discussion.

This paper will be of interest to the field and may influence thinking, since it provides clear hypothesis to test in human data about sexual dimorphisms in neuroanatomy and their trajectory, both in health and disease. There are also clearly identified regions for probing of mechanisms.

The claims presented in the paper are to my mind, convincing but I see additional areas, in which these claims could be reinforced. For example, whilst one can often find a significant sex difference in many parameters of brain structure, function and thus behaviour, little attention is often paid to the biological relevance of such p-values using measures of effect size, such as Cohen's d. For example in Figure 4 and 5, the data are shown as FDR-corrected q-value, but this could also be represented as effect size, particularly for some of the clusters which look very small and not biologically relevant, for example the motor cortex.

Additional studies could also be performed to increase the validity and, potentially, novelty of the claims.

First, the primary advantage of mouse studies is to allow investigation of potential mechanisms. This is somewhat lacking in this study, beyond identification of localization with gene expression changes. Whilst this is an elegant approach, additional mechanistic insights, or validation of the MRI-gene expression alignment would, in my opinion, be helpful. Clearly, in-depth histology is beyond the remit of this paper, but it does provide scope for others in the field to address this.

Second, behavioural analysis of the manganese injected pups in terms of neurological scoring for developmental milestones and assessment of some simple behavioural tests in adulthood (e.g. open field, social interactions and so on), would confirm the absence of any potential toxicity or sexual dimorphisms arising as a function of the injection, particularly since there will be a small amount of physical stress associated with the injection. Handling and injection of the lactating dams could also have some influence on their maternal care – was this assessed?

Third, in relation to behaviour, what is the functional relevance of these sexual dimorphisms? Demonstrating that these developmental differences may read out, or be predictive of, known sexually dimorphic behaviour would be extremely interesting as this would allow predictions about aberrant behaviour and whether the underlying neuroanatomy in females reflects more that of males in mouse models carrying mutant genes associated with autism – for example in humans see the work of Lai et al., *Brain* 2013 doi: 10.1093/brain/awt216; and Ecker et al., *JAMA Psychiatry*, 2017 doi: 10.1001/jamapsychiatry.2016.3990.

Fourth, there are structural datasets of human infant and even neonatal MRI emerging – for example, the developing human connectome. Is it therefore possible to validate some of the

emerging hypotheses from these mouse data in such publicly available datasets?

Fifth, the authors are leading experts in the field of mouse brain anatomy. It would therefore be interesting to example how cortical thickness also varies as a function of sex across developmental time periods, particularly in light of aforementioned existing human data on sex differences in cortical thickness in infants (e.g. Li et al., J Neuroscience, 2014; doi: 10.1523/JNEUROSCI.3976-13.2014).

The manuscript is clearly written, although some of the statistical analyses and image registration methods may be hard to grasp for non-specialists and could be made more accessible.

Sufficient methodological data exists to test reproduction of the study. The authors may also wish to consider making the dataset publicly available for use by the mouse imaging community.

In my opinion the statistical analysis is sound, with the exception of my earlier point regarding effect sizes.

There are no ethical concerns arising.

Reviewer #3 (Remarks to the Author):

Summary of the key results

This manuscript describes the use of manganese-enhanced MRI to generate a longitudinal map of developing sexual dimorphisms in the C57BL/6 mouse brain. The authors observed that neuroanatomical structures that are larger in male mice in adulthood develop early, whereas structures that are larger in adult females develop around and after puberty. K-means clustering was used to differentiate between different patterns of sexual dimorphic development over time, suggesting 4 different modes. Lastly, the authors present an analysis of genes that demonstrated enriched expression within the sexually dimorphic clusters.

Validity:

The manuscript describes experiments that are well designed and appropriate to address the conclusions.

Originality and interest:

These findings are of particular interest to investigators working in structural imaging, sex differences, or development. I believe that they are of interest to the broader scientific community as well.

Data and methodology:

The quality of the magnetic resonance imaging is outstanding, especially considering the technical difficulties in imaging neonatal mice. The approach used to evaluate structural changes in the brain over time are appropriate for the analysis. The data are well presented, both in the paper and the supplementary data. The supplementary videos were excellent - very compelling. The methods appear to be readily reproducible, with the exception of the methodology used to calculate "spatial gene expression patterns."

Appropriate use of statistics and treatment of uncertainties:

The use of linear mixed-effect model was appropriate for this analysis.

Conclusions:

The manuscript's conclusions are sound and reached logically from the results.

Suggested improvements:

The methodology for the analysis of the colocalization of the sexually dimorphic clusters and gene expression maps from the Allen Brain Atlas is not clear and should be presented in greater detail. I think it would be informative to report the mean volumes of the BNST, MeA, MPON, and PAG at each time point as illustrative of the development of these structures over time (perhaps a table?).

The graphs on the right of figure 3 are interesting, but unreadable. The text is invisible. This is true of several of the figures. The optimum font size suggested by the journal is 8pt. Please use this as a guideline and use a larger font size where practical.

References:

Appropriate use of references.

Clarity and context:

This manuscript was remarkable easy to read. I did not find any grammatical errors or typos. The manuscript itself was clear and straightforward. Very well written.

Allan MacKenzie-Graham

Reviewer #4 (Remarks to the Author):

Longitudinal MEMRI of male and female mice spanning ages P3-P65 confirmed a number of previously established neuroanatomical sexual dimorphisms and also revealed new dimorphisms. Of particular interest is the finding that male-biased regions (male>female) appear relatively early in postnatal life (before weaning), whereas female-biased regions tend to appear relatively later, around the time of puberty. Cluster analysis identified 4 clusters of brain regions based on developmental trajectory: 1) regions larger in males throughout development; 2) regions initially male-biased that transitioned to female-bias; 3) regions initially not sexually dimorphic that became female-biased; and 4) regions larger in females throughout development. This report demonstrates the utility of MEMRI in studying developmental trajectories of brain regions in mice and contains intriguing, novel findings particularly in female-biased regions, highlighting puberty as a significant window of development for the female brain. However, the paper falls short in a number of ways that limit the conclusions that can be drawn and this reviewer's enthusiasm.

Major comments

1. There is a large gap, more than one month, between the last two ages at which scans were obtained, P36 and P65. Although mice may be post-pubertal by P36 on some somatic and endocrinological measures, a significant degree of brain and behavioral maturation occurs during the adolescent period between P36 and P65. This gap raises some concern about missing information during an important developmental window, and necessarily means that the temporal resolution for changes in brain region volume is much poorer for the pubertal/adolescent period than for the neonatal/prepubertal period in this study. The authors should comment on how this disparity in temporal resolution could alter findings, models, or cluster analysis. For example, what should we make of the curvilinear nature of relative volumes between P36 and P65 in the absence of data during that time (e.g. Fig 6D)?

2. The authors draw parallels between earlier emergence of male-biased brain regions and male-biased neurodevelopmental disorders and later emergence of female-biased brain regions and psychiatric disorders. This begs the question of what brain region size really means (if anything) in the context of these disorders. This question is not directly addressed; rather the authors frame

the issue as periods of relative change in brain region size being periods of vulnerability (discussion p 9-10). This framing does make sense, but it also seems to ignore the fact that the findings are largely centered around sex differences in brain region volume over time, which does not provide much insight into relative change. Growth rates, which would be indicators of relative change, are shown only for clusters in Fig 6. Authors should be more explicit in how they view the relationship between the developmental trajectory of sexual dimorphisms based on brain region size and the developmental trajectory of disorders that disproportionately affect males or females.

3. I found figure captions to be generally lacking in important details. Authors should examine them carefully and add information, either to the captions or appropriate results section, so that the reader can clearly understand what's in the figures.

a. Fig 1: The box plots need to be explained in better detail, specifying what the line within each box represents (the median?), and what the whiskers and data points represent. Fig 1A is a photo of custom holders for neonatal mice; the mouse pictured does not look neonatal—has hair, looks relatively large, e.g.

b. Fig 3: Please state specifically that the red and green dots/lines in the relative volume are data for individual mice. The relative volume measure is somewhat confusing. Figure caption states that "relative volume corrects for whole-brain size differences between subjects and is expressed as a percent difference from the average volume of the brain structure". If the black line at 0 is the average volume of the brain structure, then how can both males and females be below the average MPON or PAG volume at P65? It's just not clear what the relative volume measure is and how to interpret these graphs.

c. Fig 7. Does the reference to gene expression at P56 reflect the age at which Allen Brain Institute data are from? Are gene expression data overlaid onto P65 images from the current study?

d. Figure captions for all supplementary figures need to be expanded to clearly describe what's being shown in the figures. Supplementary figures should be specifically referred to in the results section at the point where they are needed. I could find no specific references except to the supplemental movies. Also the top of p 8 refers to a full list of genes in the appendix—I found no appendix.

4. The gene expression data are not particularly useful, as presented. Results refer to spatial gene expression data from the Allen Brain Institute "which has genome-wide gene expression maps in the adult male mouse brain" (p 7, bottom). Yet expression patterns for some genes are shown for P4, P14, P28, and P56; presumably these are from the ABI developmental data base. Are expression patterns for all ages based solely on males? If so, then these data are not especially informative, as what one would really like to know is whether expression patterns differ in males and females at particular time points. What are we to make of the gene expression data based on males as they relate to female-biased brain regions? At the very least, gene expression results need to be discussed in light of this limitation. Also, the expression of some genes at some time points appears to be strikingly unilateral—is that really the case, and if so, what does that mean (e.g., Supp figs 4 and 5, Gabrg on P4, P14, P28)?

Other comments

5. The concentrations of E2 and testosterone shown in Fig 1 are puzzling and raise questions about the assays. For example, are testosterone levels in P66 males really 0.005 ng/ml? In most assays, that would be undetectable. It also seems odd that circulating E2 levels are somewhat higher in non-scanned males than in non-scanned females. More details on the hormone assays should be provided: are these RIAs? What is the minimum detectable hormone level? What are the inter- and intra-assay coefficients of variation?

6. Please clarify the distinction between the medial amygdala in cluster 1 and the amygdala in cluster 2. Does the cluster 2 amygdala refer to basolateral, central, cortical amygdala? Something else?

7. Last sentence of middle paragraph on p 10 states that "...this sex difference is dependent on neonatal circulating estradiol during the critical period...". This is not exactly right; it's not circulating estradiol that's different in males and females—it's circulating testosterone. Testosterone is aromatized to estradiol locally in the brain of males, which does not get translated into sex differences in circulating estradiol.

8. Methods, p 8: please clarify whether the two pups from each litter that were selected for scanning were one male and one female, or not.

Reviewer #5 (Remarks to the Author):

Qui, Fernandes and colleagues have imaged the developing C57BL/6J mouse brain from postnatal day 3 at 9 time points to young adulthood (P65). They replicated size differences in three subcortical areas known to have male sex differences from many existing studies. Then they categorized relative sex differences in other parts of the brain making generalizations about when sex differences appear in various clusters of neural structures. Technically, the study is competently done. The topic is also of interest - certainly there is more to learn about sex differences in the rodent brain. The problem is size does not tell us what we need to know at this point.

Structural and functional MRI have been a great advance for the study of humans where there are inherent limitations in being able to investigate the cellular and molecular underpinnings of sex differences due to issues of brain preservation and availability of postmortem tissue. Animal models are useful because they allow examination of these levels, and currently much is known about many cellular phenomena underlying sex differences including their developmental time course and many of their connections. This makes the approach in the present study, though elegant, not particularly useful.

Furthermore, the cellular underpinnings are the essence of the overall sex difference in brain size in many species (never explicitly reported here). Examining sex differences relative to overall brain size obviates what makes the size different. Each neural area is a composite of numbers of neurons, types of glia, the dendritic tree, incoming axons etc., each contributing to the size of the neural area which in turn contribute to the size of the brain. Any quantification of, for example, the number of synapses is found by multiplying the density of synapses (synapses/volume) by the volume of the structure. Absolute volume is the only meaningful entity. The researchers examining humans has gotten caught up with relative volume, partly because they do not have the opportunity to get to the cellular level and it is an easier political message for the general public. Neither of these are excuses for the animal literature. Neuropsychiatric disorders are due to problems at the cellular and molecular levels, some of which may manifest themselves in size, but size is not the problem per se.

Other issues with the study:

- It is an overgeneralization that females have a predisposition for disorders with a late onset. Male adolescents are more prone to addiction and schizophrenia during adolescence.
- Does the generalization presented in title hold if absolute size differences are examined? There is a growing literature indicating greater pruning in female rodents during adolescence in both the cortex and hippocampus. This does not lead to greater size and is another indication of how misleading relative size is.
- Exposure to any anesthetic during the early postnatal period in altricial rodents leads to extensive cell death in the brain (see work by JW Olney) and there is evidence that the effect is larger in males (JL Nunez). This is a confound in the present study. The authors did look at body weight and pubertal onset compared to controls, but the anesthetic effects are neural rather than in the body.

-The authors often cite work examining humans but the current study is on mice. For example, in the last paragraph on p.5, they state that males often have bigger brains than females and they cite a study on humans. Does this generalize to rodents and to the strain of mouse they are examining? They also make some generalizations about sex difference in pain that are not supported by the literature. Mogil has found that sex differences in pain sensitivity varies with the species and even among strains of a species.

Reviewer #1 (Remarks to the Author):

This paper describes a study of sexual dimorphisms in the developmental trajectories of mouse brain structure as studied with longitudinal acquisitions of Manganese-Enhanced MRI. Comparisons are made at the level of atlas-based brain regions and at a voxelwise level, and the regions of interest are shown to be associated with expression of genes relevant to sexual development.

The paper is a generally well-written description of a well-designed study that includes several elegant, novel, and sophisticated approaches. There are a few conceptual issues, some clarifications that should be made, and a few issues with respect to wording, which are addressed below.

Conceptual issues:

1. The sentences in the first paragraph beginning “Males have a predisposition for disorders that have early onset during childhood.... [4]” are not without some controversy and not without counter-examples (e.g., schizophrenia). This should be clarified.

This sentence has been changed to clarify that not all male-biased disorders emerge during childhood. Mention of schizophrenia as well as addiction, both of which affect males more in adolescence, has also been added to the discussion.

Page 1

processes including pain [1], learning and memory [2] and language [3]. Notably, there are robust sex differences in the prevalence, age of onset, and course of various psychiatric disorders. Males **tend to** have a predisposition for disorders that have **earlier onset, many of which emerge during childhood, including** autism spectrum disorders, attention deficit disorders and Tourette syndrome. Females **tend to** have a predisposition for disorders that have later onset, during adolescence and early adulthood, which include major depressive disorders, anxiety disorders and eating disorders [4].

Page 9

areas in females predominate the brain in later, post-pubertal life. These patterns **s** mirror what is known about sex differences in age of onset for many psychiatric disorders: males are more likely to be diagnosed with disorders that are developmental in nature, and have an onset during childhood, while females are disproportionately diagnosed with disorders that have an emotional nature and emerge in adolescence and young adulthood [4]. **Although, it should be noted that males are more prone to addiction and schizophrenia in adolescence, prior to when onset occurs in females [4] [29].**

2. Importantly, much is made about the potential for this work to contribute to knowledge about sex differences in humans and in human disorders. However, it must be acknowledged that the considerable cross-species differences in the endocrine events of puberty may limit the interpretations with regard to human development. First is the obvious example that rodents are born much more mature than humans. And the endocrinology is considerably different (e.g., adrenarche in humans is a primarily human event). These and related issues should be discussed as limitations.

A section on limitations of this research regarding cross-species differences in physiology, endocrinology and development has been included in the discussion.

The development of the brain at birth between rodents and humans is indeed at different stages of progress; however, the rodent brain at birth is about the equivalent to the third trimester human brain, so the rodent brain is born less mature than the human brain (Semple et al 2013). We have added this distinction in brain development at birth, and clarified that our results from mouse neonatal time points (p03-p10) shed insight on the human prenatal brain. We have also acknowledged the existence of adrenarche in humans, and the implications of adrenarche timing on the brain.

Semple, B. D., Blomgren, K., Gimlin, K., Ferriero, D. M., & Noble-Haeusslein, L. J. (2013). Brain development in rodents and humans: Identifying benchmarks of maturation and vulnerability to injury across species. Progress in neurobiology, 106, 1-16.

3. Similarly, the mouse brain, particularly the cortex, differs from that of humans in critical respects, and in regions critical for development and for disease. e.g. the dorsolateral prefrontal cortex, while uniquely developed in humans, is virtually nonexistent in the mouse. This limitation, too, should be discussed.

Indeed, there are several differences between the mouse and human cortex with the existence of specific cortical areas that are present only in the human and not in the mouse. This has been noted and is discussed in the limitations paragraph of the discussion.

Page 11 & 12

There are several limitations concerning cross-species differences in development, endocrinology and brain anatomy to acknowledge. The rodent brain is less mature than the human brain at birth [21]; thus our neonatal findings shed light on the human prenatal brain. Investigating this period of development is important though, as sexual differentiation of the human brain begins in the latter half of pregnancy [58]. Furthermore, there is evidence that some processes that underlie certain psychiatric disorders in postnatal life occur during prenatal development, such as autism [59]. Unlike mice, humans and some upper-level primates undergo adrenarche prior to puberty. Studies that examine the impact of adrenarche on the brain are both limited and show conflicting results, although there is some evidence suggesting that dysregulation of adrenarche timing is related to future mental health symptoms, and may affect males and females differently [60]. We cannot overcome this in mouse studies, but it is worth considering this endocrine event which may have additional activation effects in humans. The human cortex is highly folded and has a more developed prefrontal cortex that contains areas such as the dorsolateral prefrontal cortex (DLPFC) which is virtually nonexistent in the mouse. Prefrontal cortical areas are important due to their roles in complex behaviours and symptoms of many psychiatric disorders. This inter-species difference in cortical structure and function is difficult to reconcile, although it has been demonstrated that certain rat cortical areas contain features that resemble the primate DLPFC [61]. However, the cortex does not operate in isolation and is connected to subcortical structures, whose interconnections and functions have been highly conserved across mammalian species.

Additional points:

4. Section 3.1: “Weight at puberty was significantly affected [by scanning].” Even though the authors found no interaction between scanning and sex, possible implications should be addressed and discussed since adipose tissue can contribute significantly to the circulating pool of estrogens.

Thank you for this comment. The ability of adipose tissue to contribute to the circulating pool of estrogens has been included in this section, along with a discussion related to potential functional implications on timing of puberty across scanned and non-scanned mice.

Page 5

several measurements were collected from scanned mice and their non-scanned littermates. Weight at puberty (Figure 1D) was significantly affected in scanned mice; however, we did not find a significant effect of scanning on any other measurements collected (Figure 1C,E,F), nor did we find evidence for an interaction between scanning and sex. Weight could impact circulating steroid levels as adipose tissue has aromatase activity [22]. Our scanned mice weighed less at puberty, so it is possible that they had lower levels of circulating estradiol. However, there were no differences in pubertal timing, which suggests that there were no functional effects of potentially differing levels of estradiol between groups of mice. Neonatal anaesthesia exposure can cause cell death in the brain,

5. The pN terminology used to describe postnatal age should be described in the last paragraph of the introduction, immediately following the descriptions of the 9 time points, before using the term p65, for example, in the results section.

Noted. “p” as an acronym for postnatal day/timepoint has been added to the last paragraph of the introduction.

Page 4

Here, we use MEMRI to investigate the development of structural sex differences in the mouse brain beginning from early neonatal life. Male and female C57BL/6J mice were scanned longitudinally with MEMRI across 9 postnatal time points (p), at days 3, 5, 7, 10, 17, 23, 29, 36, and 65. First, known sex differences in the BNST, MeA, and MPON were investigated to affirm that

6. The relationship of these time points in the mouse to development in the human should be made clear.

Explicit comparisons of our scanning time points and their corresponding human developmental periods have been included in Section 3.1 of the Results.

Page 5

The methodology and image analysis techniques employed here are an extension of previous work [20], expanding both the throughput and the imaging window to include adult brain development, and applied to both sexes. **Imaging over such a comprehensive time window allows us to study neuroanatomical change that occurs across the human equivalent of prenatal life (p3-10), birth (p10), childhood (p10-p29), puberty and adolescence (p23-p36) and adulthood (p65) [21].**

7. Grammatical errors, such as “These pattern mirrors...” (3rd paragraph in Discussion) should be corrected. Also, e.g., in the heading of 3.4, “reflect” should be “reflects”

Noted. These grammatical errors have been corrected.

Page 9

Relatively larger areas in males predominate the brain in early, pre-pubertal life and relatively larger areas in females predominate the brain in later, post-pubertal life. These patterns mirror what is known about sex differences in age of onset for many psychiatric disorders: males are more likely to

8. When discussing the longitudinal registration, references to “linear registration” should technically be “affine registration”.

Noted. All instances of “linear registration” have been changed to “affine registration” and ‘non-linear registration’ had been changed to ‘non-affine registration’.

Page 15

sponding points in the other images. Thus, the differences between the images are captured by this transformation. Our procedure for image registration is composed of **an affine** registration, followed by a series of non-affine registrations. The **affine** registration applies global translation, rotation, scaling, and shearing to align images. Information regarding global deformations (i.e. the overall brain sizes) are stored in these transformation models. The non-affine registration creates a vector field that maps every point in one image to another and provides information about localized

9. How were statistical corrections carried out for the multiple analyses, multiple time points, and multiple regions/clusters examined?

For each analysis we carried out, the methods section details whether and how statistics were corrected for multiple comparisons. Multiple correction was not done for the analysis of scanned vs. non-scanned mouse data (i.e. body weight, hormones, etc). This would remove the only significant difference between non-scanned and scanned mice, further demonstrating that neonatal scanning does not have adverse effects. Multiple correction was not done when analysing Sexual Dimorphisms in Canonical Structures as these were assumed to be sexually dimorphic a priori. All voxel-significance-statistics were corrected using false discovery rate. Significance testing was never conducted cross-sectionally and therefore corrections were not carried out for multiple timepoints. Significance testing was never conducted on clustered data and therefore no corrections were applied. Significance statistics were not carried out on individual genes regarding their spatial expression data and therefore, no corrections were done. Kolmogorov Smirnov tests were applied independently and therefore no corrections were done.

Page 14

We did not correct data comparing scanned and non-scanned mice for multiple corrections, as the only significant differences were for weight at puperty. Upon correcting for mutiple comparisions using FDR [66], these differences disappear further illustrating the similarity between scanned and non-scanned mice.

10. In section 5.4.2 (Methods), description of the likelihood ratio should include an appropriate reference to Wilks' Theorem.

The reference has been added.

Page 16

$$D = -2 \ln \frac{L_p}{L_f} \quad (3)$$

According to Wilks' theorem [65], Equation (3) follows the χ^2 distributions with degrees-of-

11. The caption for Supplemental Figure 6 indicates that models were approximated with 6th order natural splines, while the methods section seems to refer to 3rd order splines were used. Was one used for absolute measures and the other for relative measures? The authors should clarify.

We thank the reviewer for this comment as we were not particularly rigorous with model selection in our original manuscript. In these revisions, we used bayes' factors to select the best model for absolute volumes and found 6th order splines to be best for about 80% of the structures in the brain. We then analyzed the voxelwise data using 6th order splines (Supplementary Figure S17). For measuring growth rate, a spline function was fit to every voxel for every individual. The order of the natural splines at every voxel was selected by minimizing Akaike information criterion. Once the order was selected, every individual was fit with a unique spline and we calculated growth rate using finite differences. Both model selection procedures have been added to their respective sections (Methods 5.3.6 and Supplementary S1.1.6).

Page 18

Growth rate was estimated by fitting the relative determinant at every voxel for every individual with natural spline functions of age, then differentiating the fitted function with respect to age. At every voxel, the order of the fitted natural spline was determined by finding which order minimized the Akaike Information Criterion [65]. Detailed in Supplementary Section S1.1.4, we used the Allen Brain Institute's gene expression dataset [26] to identify genes spatially enriched in our clusters [68].

Page S5

S1.1.6 Optimizing Growth Models for Absolute Determinants

Absolute volumes required more complex growth curves than relative volumes. To model this curve, we used a similar procedure to Section S1.1.5. We found the Bayes Factor associated with Model (12) for values of spline order N (fixed effect of growth) ranging from 1 to 8. For 80% of structures, Bayes Factor was maximized by splines of order $N \leq 6$. The data was then fit with the model defined by Equation (13) with $N = 6$ and the optimized M (spline order associated with random effect growth) is determined by finding the model with the minimum BIC. We found that 95% of structures were best fit by $M \leq 2$. Thus, we chose order-6 natural splines for fixed effects of age and order-2 natural splines for random effects of age to fit absolute Jacobian determinants. We computed the likelihood-ratio statistic comparing this optimized model to a similar one without sex and sex-age interactions to ascertain significance of sex on absolute determinants.

12. The description of the multi-level ANTS-based normalization technique is relatively clear, but the description should specify whether the concatenated ANTS transforms were performed in a single interpolation step. On a potentially-related note, the use of the final time point image as a reference is somewhat counter to recent recommendations (Reuter, 2011 and others) for avoiding asymmetry-induced bias by using central time point images as reference images. The authors should address the issues of interpolation and whether their approach might introduce any bias.

Text has been changed to explicitly mention that concatenated transformations were performed in a single interpolation step.

Page 15

average registered to p5 average, p5 average to p7 average, etc). The final step in the registration is to concatenate the transforms from both levels so all images can be mapped to the p65 consensus average brain **in a single interpolation step**. For example, to align the image of a p29 subject brain to the p65 average brain, the following transformations are concatenated: p29 subject to p29 average,

Registration bias is an important point and we thank the reviewer for bringing this to our attention. Reuter et al. (2011) specifically recommend against using a specific time point as a consensus space in cases when within-subject variability is less than between-subject variability. In our case, within-subject variability is greater than between-subject variability, as within-subject variability encompasses neuroanatomy development over time. It is for this reason we first register between subjects (at the same age), and then register across time. Registration over time is where biases can enter our analysis in two ways. First is interpolation bias, where the final timepoint is treated differently from the other timepoints as it is the only one that is not interpolated. Second is label bias, where the atlas is registered to features of the final time point and volumetric measurements for all time is only made in reference to this final time point.

We first reproduced Figure 4 using p17 as the reference timepoint instead of p65 and found the statistics maps to correlate strongly (Supplementary Figure S9). P17 was chosen as the reference as it is the median timepoint and this follows more closely to the recommendations of Reuter et al. (2011). We also generated arbitrary statistics for each timepoint natively and compared them to the same statistics generated in the p65 reference space, and again found high correlation (Supplementary Figure S10). This indicates that interpolation bias does not play an important role in our analysis.

Figure 4: Sexually dimorphic neuroanatomy. Two linear mixed-effects models were fit to the data: Model 1 had sex as a predictor and Model 2 did not. Every voxel was assessed on whether Model 1 had a significantly better fit than Model 2. The resulting q-values (thresholded to $q < 0.1$) were overlaid on p65 average brain cross-sections, identifying several regions in the brain where volumes are significantly dependent on sex. These sexually dimorphic regions include parts of the cerebellum (Cb), medulla (My), midbrain (Mb), Pons (P), PAG, thalamus (Th), hippocampus (Hip), amygdala (Amy), sensory cortex (SCx), hypothalamus (Hy), cingulate cortex (Ccx), caudoputamen (Cp), BNST, motor cortex (Mcx), and olfactory bulbs (Ob).

Figure S9: Sexually dimorphic areas in the mouse brain calculated after choosing p17 age-consensus average as the registration consensus average. Voxelwise statistics was computed similar to that in Figure 4 except that p17 age-consensus average was chosen as the registration consensus instead of p65. The resultant statistics map was in p17 age-consensus space and was transformed to p65 age-consensus space for comparison to Figure 4. A high correlation was observed between these two maps ($r = 0.992$) indicating that choosing p65 as the consensus versus p17 (the median time point) incurs little bias in identifying sexually dimorphic regions.

Figure S10: Statistics maps generated without longitudinal registration are similar to those generated with longitudinal registration. Random statistical maps were generated for each time point by permuting sex labels and computing effect size comparing males and females. For every permutation, the effect sizes were calculated for Level 1 determinants (agnostic to longitudinal data and in age-consensus space) and Level 2 determinants (dependent on longitudinal registration and in p65 age-consensus space). The effect sizes from Level 2 determinants were transformed to the age-consensus space corresponding the effect sizes from Level 1 determinants. Correlations are computed between the transformed Level 2 effect size map and Level 1 effect size map and correlation was observed to be high for all time points. The dotted line indicate correlation when the sex labels are not permuted and correspond to true volumetric effect sizes between males and females.

To test label biases, we compared volume estimates for every structure using three sets of labels (Illustrated in Supplementary Figure S13). The first are consensus labels, which are labels placed on the p65 average brain. We used these labels throughout our study. The second are resampled labels, where consensus labels were transformed to each timepoint using the Level 2 transforms in our registration. Volume estimations using these labels can be done in the native space of each timepoint. The final set of labels were those generated with the PydPiper pipeline. Starting with the p65 atlas, we created multiple p65 intermediate atlases by transforming the atlas into the native space for all p65 subjects. These intermediate atlases were then registered to the p36 average and, for each voxel, we vote across all atlases to generate the p36 atlas. The p36 atlas was then used as a starting point for the p29 atlas. In this way, bias is minimized (not eliminated) as each time point's atlas is directly influenced by the nearest older time point's atlas by multiple intermediate atlases and not just the p65 time point atlas. We found that the three different sets of labels do generate different measures of structure volume, therefore there is an important label bias. However, we also found that this bias does not affect individuals and sexes differently (Supplementary Figure S14). We thank the reviewer for pointing out this potential source of bias in our analysis. This discussion about registration bias can be found in Supplementary Section S1.1.3.

Figure S13: Three different sets of atlases used to check for registration bias in structures. Consensus atlas is the MRI atlas registered to the registration consensus average (which is also the p65 age-consensus average). Since all images are registered to this consensus average, we use this atlas alone to quantify structure volumes in our main study. The two additional sets of atlases created test for different types of biases. The resampled atlases are created by transforming the consensus atlas to every single age in a single interpolation step with transformations obtained from Level 2 of our registrations. This atlas allows us to check for resampling bias as with this atlas volumetric information need not be transformed to p65 space prior to quantification of structure volumes. For a given time point, the voted atlases are created aligning the age-consensus average of this time point to the atlas overlaid on every subject of the next immediate-older time point. A voxel voting procedure is taken across all subject atlases to create the voted atlas on the time point's age-consensus average. This method greatly reduces the bias in choosing single starting adult atlas and transforming it to younger time points. There are multiple intermediate atlases and each time point is only responsible for creating an atlas for the immediate-younger time point.

Figure S14: Registration bias exists with atlases but do not discriminate individuals or sex. A) The volume of the Lobule 1-2 white matter estimated using the three sets of atlases: consensus, resampled, and voted (see Supplementary Figure S13). Trendline and standard error (shaded region) were obtained by fitting a linear mixed effects model. We see that registration bias does play a role in volume estimation of small structures like the Lobule 1-2 white matter as below p07, voted atlases say this structure does not exist and the other atlases do not agree. B) We computed z-score, removing the overall mean and standardizing variability across the three sets of atlases for each age. We test whether registration bias of structure volumes applies equally across all individuals using two linear mixed effects model: the first model had z-score volumes as a response variable; fixed effect of time point, sex, and atlas (Consensus, Resampled, or Voted), as well at all interactions; and random effect of individual and individual-atlas interaction, and the second was the same but lacked the random effect of individual-atlas interaction. A log likelihood test showed that registration bias does not discriminate individuals ($\chi^2_5 = 0.33, P > 0.99$). To test if registration bias was the same between the sexes, we performed a similar analysis, except the second model lacked all interactions between sex and atlas. We found that registration bias does not discriminate sex ($\chi^2_{12} = 0.08, P > 0.99$). All other structures showed little effects of registration bias affecting each individual differently (uncorrected all pvalues > 0.93) or affecting each sex differently (uncorrected all pvalues > 0.999). This supports our conclusion that while registration bias may exist in structure volume measurements, this bias applies equally to all individuals and sexes.

13. Although the asserted sexually dimorphic differences in time courses of the emergence of relative regional size differences are intriguing and potentially represent a valuable contribution to the field, the statistical strength of these findings, the main conclusion, is difficult to judge without further clarification about the uncertainty measures shown in the developmental plots in Figure 6. Do the plots in part D represent confidence intervals, standard deviation, standard error, or something else? The authors should clarify this information and indicate statistical significance of the differences between males and females at various time points, thereby allowing the reader to make informed judgements about the strength of these key results.

We thank the reviewer for bringing this to our attention. We over-relied on the default confidence interval in our plotting software (ggplot2) and this confidence interval was calculated without accounting for the fact that data was longitudinal. In response we have placed standard error (either as bars or shaded regions) throughout the manuscript and they were all estimated using linear-mixed effects models, which does account for the longitudinal nature of our analysis. We have put standard error shaded regions and now use Cohen's d values instead of log male-female ratios for Figure 6. Statistical significance of sexual dimorphisms are impossible to ascertain with clustered data as the data was clustered by sex differences - thus p-values from traditional tests would inflate false-positives. We hope that by providing effect-sizes values, we can let the reader judge the strength of the results as the reviewer suggests.

Figure 6: Coordinated growth of sexually dimorphic functional networks. Sexually-dimorphic voxels with similar effect sizes through time were clustered into 4 groups using k -means. A) Results of the clustering analysis on sexually dimorphic voxels. Effect sizes (positive is bigger in males, and negative is bigger in females) of B) Relative volumes and C) Growth rate for the different clusters. D) Average volume and growth rate in each cluster for each individual. Cluster 1 corresponds to regions larger in males and this dimorphism emerges early in development. Regions involved in the vomeronasal system, which processes pheromonal information, are found in this cluster. Cluster 2 are also regions larger in males but the onset more delayed. However, this cluster in early life shows strong bias in growth rate towards males. Parts of this cluster include the olfactory bulb and pallidum. Cluster 3 voxels trajectory switches from being larger in males in early life, to larger in females post-puberty. Parts of the sensory cortex and PAG belong to this cluster. Cluster 4 contains regions that are not sexually dimorphic in early life but become larger in females over the course of development. This cluster includes association related areas such as parts of the central thalamus and temporal association cortex; and motor related areas such as parts of the hindbrain, cerebellum, caudoputamen, and motor cortex. Both Cluster 3 and 4 show growth rate bias towards females peaking around puberty.

Reviewer #2 (Remarks to the Author):

Qiu et al. Nature communications

“Mouse MRI shows brain areas larger in males emerge earlier than those larger in females”

The paper by Qiu and colleagues describes the use of manganese-enhanced MRI (MEMRI) to map both canonical and novel regions of the mouse brain, which show sexually dimorphic development and relate these to spatial measures of gene expression, thus providing clues to possible mechanisms. Based on the data presented the authors make the following major claims:

(1) MEMRI is a useful method to capture developmental neuroanatomical differences without adverse effect

(2) MEMRI detects sex differences in the volume of three brain regions (BNST, MeA, MPON), previously reported to be sexually dimorphic, providing validation of the sensitivity of the method

(3) Brain regions that are larger in males tend to predominate early in development, in the neonatal and prepubertal period, whilst regions that are larger in females emerge post-puberty

(4) This can be refined to reveal 4 distinct trajectories based on k-means clustering showing distinct spatiotemporal differences in volumes of various brain regions between males and females

(5) These data overlap with prior studies of sexual dimorphism (e.g. using 4-core genotype mice)

(6) Brain regions that show sexual dimorphisms in volume are enriched in expression of genes known to be involved in the sexualisation of the brain, for example *Esr1*, *Esr2*, but also novel genes such as *Slc6a4* and *Tph2*, based on spatial normalization of the MRI data to the Allen Brain atlas.

These claims are novel, at least in terms of mouse studies, since the majority of such studies in mice have only examined sex differences in adults, or from adolescence and not included earlier time points during development. The data presented herein are therefore novel for this field.

In terms of human MRI studies of sexual dimorphism in anatomy, the concept that some aspects of neuroanatomy, such as cortical thickness are larger in infant males as compared to females at 2 years of age (Li et al., J Neuroscience, 2014; doi: 10.1523/JNEUROSCI.3976-13.2014) has already been suggested. Other studies also provide evidence for early gender differences in volume, cortical thickness and surface area (e.g. Wierenga et al., Neuroimage 2014; doi:

10.1016/j.neuroimage.2013.11.010). I would suggest a more exhaustive search of the extant literature to acknowledge that the idea that sexual dimorphisms might arise early based on human studies of infants, which are few, but do exist as show above is warranted to justify the novel aspect of these new mouse data. Despite these points, the claims are however in general, adequately discussed in the context of prior literature.

Thank you for this comment. Acknowledgement of male-biased neuroanatomy emerging in infancy, including the Li et al J Neuroscience 2014 paper, has been included in the introduction

Page 3 & 4

either cross-sectional, or examine sex differences on a relatively short developmental timescale. A recent meta-analysis examining sex differences in human brain structure across life determined that very few, if any, studies investigate sex differences during infancy and early childhood (0-6 years of age) [12]. Of those that do, however, there is an indication that some aspects of neuroanatomy, such as cortical gyrification, are larger in males in infancy compared to females [17], pointing to the existence of early life sex differences, and further emphasizing the need to examine sex differences within neonatal development. Because there are opportunities for activational sex differences to

In particular, where these mouse data do bring novelty to the human field is in the identification of many more brain regions that show sexual dimorphism and clustering these into distinct trajectories. One difficulty here is how best to translate these findings to humans, since as the authors acknowledge “MRI allows investigation of the whole brain, but lacks the ability to detect smaller sexually dimorphic nuclei”. This conceptual and technical barrier should be considered in the discussion.

Although current human MRI methods may lack the ability to detect these smaller nuclei, they still do exist in the human brain. Our mouse MRI results, then, inform about the human brain and how it may change across development. This point has been added to the Introduction.

Page 3

others. In humans, MRI allows for whole-brain investigation, but lacks the ability to detect smaller sexually dimorphic nuclei, even though they do exist. Furthermore, the long lifespan of humans

This paper will be of interest to the field and may influence thinking, since it provides clear hypothesis to test in human data about sexual dimorphisms in neuroanatomy and their trajectory, both in health and disease. There are also clearly identified regions for probing of mechanisms.

The claims presented in the paper are to my mind, convincing but I see additional areas, in which these claims could be reinforced. For example, whilst one can often

find a significant sex difference in many parameters of brain structure, function and thus behaviour, little attention is often paid to the biological relevance of such p-values using measures of effect size, such as Cohen's d. For example in Figure 4 and 5, the data are shown as FDR-corrected q-value, but this could also be represented as effect size, particularly for some of the clusters which look very small and not biologically relevant, for example the motor cortex.

We agree with the reviewer that effect sizes are a more-intuitive measure than q-values and allows the reader to make informed decisions about our findings. We abandoned our clustering analysis using log male/female ratio and instead clustered by effect size (Figure 6). This does change the clusters somewhat but the overall story regarding time courses is the same. We do not think it is appropriate to use effect size measures for Figures 4 and 5. In the case of Figure 5, highlighted regions are areas of the brain where sex significantly influences brain volume. This could be because of persistent sexual dimorphisms, which would be captured by effect sizes like Cohen's d, or changing sexual dimorphisms over time, which would not be captured by effect sizes at a particular time. In Figure 5, the advantage of t-statistics is that they are outputs of a linear mixed-effects model and thus take into account information across time to smooth data, which Cohen's d do not. Nevertheless, agreeing with the reviewer's suggestions, we made Figure 6 and Supplementary figures S2-S4 contain effect sizes for easier interpretability.

Figure 6: Coordinated growth of sexually dimorphic functional networks. Sexually-dimorphic voxels with similar effect sizes through time were clustered into 4 groups using *k*-means. A) Results of the clustering analysis on sexually dimorphic voxels. Effect sizes (positive is bigger in males, and negative is bigger in females) of B) Relative volumes and C) Growth rate for the different clusters. D) Average volume and growth rate in each cluster for each individual. Cluster 1 corresponds to regions larger in males and this dimorphism emerges early in development. Regions involved in the vomeronasal system, which processes pheromonal information, are found in this cluster. Cluster 2 are also regions larger in males but the onset more delayed. However, this cluster in early life shows strong bias in growth rate towards males. Parts of this cluster include the olfactory bulb and pallidum. Cluster 3 voxels trajectory switches from being larger in males in early life, to larger in females post-puberty. Parts of the sensory cortex and PAG belong to this cluster. Cluster 4 contains regions that are not sexually dimorphic in early life but become larger in females over the course of development. This cluster includes association related areas such as parts of the central thalamus and temporal association cortex; and motor related areas such as parts of the hindbrain, cerebellum, caudoputamen, and motor cortex. Both Cluster 3 and 4 show growth rate bias towards females peaking around puberty.

Additional studies could also be performed to increase the validity and, potentially, novelty of the claims.

First, the primary advantage of mouse studies is to allow investigation of potential mechanisms. This is somewhat lacking in this study, beyond identification of localization with gene expression changes. Whilst this is an elegant approach,

additional mechanistic insights, or validation of the MRI-gene expression alignment would, in my opinion, be helpful. Clearly, in-depth histology is beyond the remit of this paper, but it does provide scope for others in the field to address this.

In-depth histology would be a useful complement to our findings, and this has been mentioned in the discussion; however, as the reviewer mentioned, it is beyond the scope of this paper resubmission.

Page 11

how they drive sexual dimorphisms is warranted. Additionally, more direct investigations — such as in-depth histology — of the cellular underpinnings of our mesoscopic neuroanatomical changes would be useful. Our results provide indications of candidate genes and areas to explore in further detail.

Second, behavioural analysis of the manganese injected pups in terms of neurological scoring for developmental milestones and assessment of some simple behavioural tests in adulthood (e.g. open field, social interactions and so on), would confirm the absence of any potential toxicity or sexual dimorphisms arising as a function of the injection, particularly since there will be a small amount of physical stress associated with the injection. Handling and injection of the lactating dams could also have some influence on their maternal care – was this assessed?

We appreciate this feedback, and in response to it have conducted new experiments to assess markers of neonatal neurodevelopment amongst scanned and non-scanned mice.

We had three groups. The first was a neonatally scanned cohort: these mice were scanned at postnatal days 3, 5, 7, 10 and 17, and thus were exposed to isoflurane during the scans and were exposed to maternal MnCl₂ at postnatal days 2, 4, 6 and 9, as well as an intraperitoneal injection of MnCl₂ at postnatal day 16. The second group was their non-scanned littermates. These mice were exposed to maternal MnCl₂ at postnatal days 2, 4, 6 and 9. The third group consisted of non-scanned control mice. These mice were not exposed to any isoflurane or MnCl₂.

We produced 6 new cages of C57BL/6 mice, each culled to a litter size of 6, with the exception of one cage that had 5 pups. Four cages were randomly selected for scanning. 1-2 mice were chosen from each cage to be scanned, while the remainder of the mice were non-scanned littermates, resulting in 7 scanned mice and 17 non-scanned littermates. There were 11 non-scanned control mice.

With the guidance of Dr Jane Foster, all of these mice underwent behavioural testing to assess developmental milestones. Righting reflex was performed on postnatal days 2, 3 and 4. Eye opening was assessed everyday from postnatal day 10 to 17. Open field was conducted on postnatal day 16. Following completion of in-depth neonatal behaviour testing, we kept the mice in order to assess adult open field performance.

Comprehensive maternal behaviour tests were not conducted; however, there were no apparent differences in maternal behaviour across groups. In particular, similar levels of nursing activity and pup retrieval following scanning and developmental milestone testing was observed by the experimenter.

For righting reflex, the time it took for pups to right themselves was recorded; for eye opening, mice were scored based on the number of eyes open (0, 1 or 2); and for open field, time spent in the centre, as well as total distance travelled was measured.

We did not find a significant effect of group (scan, non-scan littermate or non-scan control), sex, or an effect of group-sex interaction on any of the neonatal behavioural test metrics. We did find a group effect of centre time in the adult open field results, where scanned mice spent less time in the centre than non-scanned littermates and control mice; however, ambulatory distance was the same across all groups. Furthermore, there were no apparent sex differences across groups for any of the adult metrics.

The results have been mentioned in Results 3.1. Further details on these new neonatal experiments have been added to the manuscript under section Supplementary Methods S1.1.1 as well as Supplementary Figure S1 displaying the results of this testing.

Page 5

estradiol between groups of mice. Neonatal anaesthesia exposure can cause cell death in the brain, and can affect males more severely [23]; thus, we assessed several neonatal neurodevelopmental outcomes (righting reflex, eye opening and open field) on additional groups of mice (Supplementary Methods S1.1.1). We found no significant effects of group or a group-sex interaction on any of the neonatal metrics collected (Supplementary Fig. S1). We conclude that the effect of MEMRI scanning on neurodevelopment was small and not sexually biased.

Page S1

S1.1.1 Behavioural assessment

To assess neonatal brain development and behavioural outcomes of mice undergoing scanning, developmental milestone and behavioural testing was conducted on additional groups of mice. Testing was conducted on three groups of mice. The first group consisted of scanned mice: these mice were scanned at postnatal days 3, 5, 7, 10 and 17, and thus were exposed to isoflurane during the scans and to maternal MnCl₂ at postnatal days 2, 4, 6 and 9, as well as an intraperitoneal injection of MnCl₂ at postnatal day 16. The second group consisted of their non-scanned littermates. These mice were exposed to maternal MnCl₂ at postnatal days 2, 4, 6 and 9. The third group consisted of non-scanned control mice, who were not exposed to any isoflurane or MnCl₂. MnCl₂ administration and scanning procedures are previously described in sections above. Six cages of C57BL/6 mice were used, each culled to a litter size of 6, with the exception of one cage that had 5 pups. Four cages were randomly selected for scanning. 1-2 mice were used from each cage to be scanned, while the remainder of the mice were non-scanned littermates, resulting in 7 scanned mice (4 male, 3 female) and 17 non-scanned littermates (7 male, 10 female). There were 11 non-scanned control mice (4 male, 7 female).

Behavioural testing was done prior to MnCl_2 administration or scanning if both fell upon the same day. Righting reflex was conducted on postnatal days 4, 5, and 6. Prior to testing, the dam was removed from the home cage. Each pup was tested individually. The pup was placed on its back, and the time required for the pup to flip over and place all four paws on the ground surface was measured. If no righting took place within 30 seconds, the pup was picked up and righted manually. After all pups in one cage were tested, the dam was returned. Eye opening was observed daily from postnatal day 10 until day 17. Each mouse was given a score depending on number of eyes open: 0 for no eyes open, 1 for one eye open and 2 for both eyes open. Open field was conducted on postnatal day 16. Behaviour was recorded in an arena that was 44 cm^2 , with 16 beams on each axis (1 inch apart) (Med Associates Inc, Fairfax VT). Each arena was enclosed inside a sound attenuating chamber. The light intensity used was 200 LUX in the centre, and 150 LUX in the periphery. Data was collected with Activity Monitor 7 (Med Associates Inc, Fairfax VT). Six mice were tested at a time. Length of neonatal open field testing was 30 minutes. Time spent in the centre of the open field, as well as total ambulatory distance were measured. Following testing, mice were returned to their home cage. Open field was performed for a second time at postnatal day 65. The length of testing was 10 minutes. For an unrelated reason, one of our cages was terminated prematurely by veterinarian technicians, thereby decreasing our number of female scanned mice by one, and our female non-scanned littermates by two.

Righting reflex time and eye opening score were compared across scanned (S), non-scanned littermates (L) and non-scanned control (C) animals by running three linear mixed effects models, and then comparing across models to assess whether group and/or sex had a significant effect. The first linear mixed effects model had fixed effects of group, postnatal day, sex, and their interactions, with a random effect of individual mouse. The second model had fixed effects of postnatal day, sex, and their interaction, and a random effect of individual mouse. These two models were compared with a likelihood ratio test to assess whether group affected righting reflex time or eye opening score. A third model was run with fixed effects of group, postnatal day, group-postnatal day interaction, sex and a sex-postnatal day interaction, with a random effect of individual mouse. This third model was compared to the first model with a likelihood ratio test to assess whether there was a group by sex interaction. For open field, time spent in the centre of the open field and total ambulatory distance were analysed using linear models.

Figure S1: Time to right, eye opening, as well as time spent in centre and total ambulatory distance travelled in the open field was assessed neonatally across scanned mice (S), their non-scanned littermates (L) and non-scanned controls (C). A) There was no effect of group ($\chi^2_8 = 14.54, P = 0.07$), nor was there a group-sex interaction ($\chi^2_4 = 6.59, P = 0.16$) on time it took for pups to right themselves across postnatal days 4, 5 and 6. B) There was also no effect of group ($\chi^2_{32} = 20.61, P = 0.94$) or a group-sex interaction effect ($\chi^2_{16} = 9.30, P = 0.90$) on when eyes opened across postnatal days 10 to 17. For both A and B, linear mixed-effects models were used to create trendlines and bars representing standard error. C) There was no effect of group ($F_{2,29} = 0.47, P = 0.63$) or a group-sex interaction ($F_{2,29} = 0.64, P = 0.54$) on time spent in the centre of the open field at postnatal day 16, nor was there an effect of group on total ambulatory distance travelled ($F_{2,29} = 1.16, P = 0.33$) or a group-sex interaction ($F_{2,29} = 0.17, P = 0.85$). Thus, no neonatal behavioural metrics collected were significantly impacted by scanning, and the results were the same across both males and females. These mice were kept for further testing in the open field as adults (postnatal day 65). Although there was no group-sex interaction on centre time ($F_{2,26} = 0.91, P = 0.41$), there was a significant effect of group ($F_{2,26} = 6.81, P = 0.004$) as scanned mice spent less time in the centre of the open field compared to non-scanned controls (post hoc Tukey test $P_{adj} = 0.003$). Total ambulatory distance, however, did not show any significant differences across group ($F_{2,26} = 1.37, P = 0.27$), or by sex within each group ($F_{2,26} = 1.34, P = 0.28$). For both C and D, mean and standard error (bars) were calculated using linear models.

Third, in relation to behaviour, what is the functional relevance of these sexual dimorphisms? Demonstrating that these developmental differences may read out, or be predictive of, known sexually dimorphic behaviour would be extremely interesting as this would allow predictions about aberrant behaviour and whether the underlying neuroanatomy in females reflects more that of males in mouse models carrying mutant genes associated with autism – for example in humans see the work of Lai et al., *Brain* 2013 doi: 10.1093/brain/awt216; and Ecker et al., *JAMA Psychiatry*, 2017 doi: 10.1001/jamapsychiatry.2016.3990.

Understanding the functional implications of these sexual dimorphisms would indeed be highly interesting. We did not assess such behaviours in this study, but have now added a section in the Discussion section that references these aforementioned studies, as well as others, that point to possible structure-function connections. Examination of sex differences in behaviour as related to sexual dimorphisms in the brain would be an excellent project to pursue in the future, particularly because there exists literature indicating that canonical sex differences in anatomy, such as the sexually dimorphic nucleus in the medial preoptic nucleus, are related to sex differences in behaviour, and, when manipulated by neonatal hormone administration or gonadectomy, size and behaviour change in a similar fashion (review by McCarthy et al, 2009). We hope that our findings can inform future studies that seek to investigate novel sexually dimorphic structure-function relationships of the brain.

Page 10 & 11

Our study, although thorough in neuroanatomical characterization of sex differences across development, does not directly address the functional relevance of these dimorphisms. However, there are robust examples of neuroanatomical sex differences which directly relate to behavioural sex differences, that also change in a corresponding fashion upon hormonal manipulation; for example, the MPON and male- and female-typical sexual behaviour [52]. Furthermore, recent research points to the utility of information about typically developing male and female brain anatomy in predicting the presence of psychiatric disorders that show sex differences, such as autism [53] [54]. Our findings can inform future work that seeks to further elucidate sexually dimorphic structure-function relationships of the brain.

Reference

McCarthy, Margaret M., Christopher L. Wright, and Jaclyn M. Schwarz. "New tricks by an old dogma: mechanisms of the Organizational/Activational Hypothesis of steroid-mediated sexual differentiation of brain and behavior." *Hormones and behavior* 55.5 (2009): 655-665.

Fourth, there are structural datasets of human infant and even neonatal MRI emerging – for example, the developing human connectome. Is it therefore possible to validate some of the emerging hypotheses from these mouse data in such publicly available datasets?

Addressing this suggestion directly is beyond the scope of this resubmission. However, based on this suggestion we decided to explore novel analyses regarding brain prediction that could be relevant to human imaging data sets.

Our in-vivo longitudinal imaging throughout mouse neurodevelopment, in conjunction with the control of genetic and environmental factors afforded by mouse studies, renders it possible to explore exciting analyses in brain prediction. We have added new sections in our revised manuscript (Figure 8, Results Section 3.5, Methods 5.3.8, Supplementary S1.1.5) that investigates how well the neuroanatomy of the young mouse brain predicts neuroanatomy of the adolescent and adult mouse brain. By developing a computational model to predict mouse brain neuroanatomy, we found that only the first 10-17 days of neuroanatomical data are enough to make sensitive and specific predictions of individualised neuroanatomy at p36 and p65. As expected, if we provide the model with only p3 data and ask it to predict p36 or p65, the model will fail to make specific predictions because young brains are so similar to each other. However, the more data we provide from later in development, the better the model is at predicting mature individualised neuroanatomy. It is through a combination of many timepoints and more recent timepoints that individual differences in young neuroanatomy emerge, which the model can use to make predictions about mature neuroanatomy. Our prediction accuracies were not influenced by the sex of the predicted subject (i.e. males and females are equally well predicted), however, we found that male neuroanatomy individualised (became easier to predict) significantly earlier in development than female neuroanatomy.

Neuroanatomical phenotyping tends to be used to study how groups of organisms are similar. However, we hope that this analysis of individuality can generate progress into what makes organisms unique from the rest of the members of its group. The control of genetic and environmental factors limits the variability in mice making it easier for machine-learning tools to make individualised predictions. "Easier" in this context means using simple models and few subjects. While humans have far more variability compared to mice, with the advent of ever-bigger datasets collected on human neurodevelopment, it will be possible in the near future to use more advanced machine-learning tools to predict some individuality in these data sets. It might even be possible to predict the development of neuroanatomical pathologies associated with neurological disorders prior to disorder onset. We hope that our analysis of mouse individuality spurs research in this direction.

Figure 8: Prediction and individualisation of neuroanatomical structure volumes. A) Predicted and Observed p36 structure volumes for all subjects and three representative structures: Primary Motor Cortex, Striatum, and the Barrel Cortex. When predicting any subject at p36, model was trained on all data excluding the predicted subject at p36 and all subjects at p65. Despite this exclusion, the model uses neuroanatomical information from earlier time points to predict structure volumes' variation from the average (horizontal line). B) Matrix plotting the Root-Mean-Square-Difference (RMSD) between the observed structure volumes (rows) and the predicted structure volumes (columns) for each subject at p36. Prediction used all data prior to and including p29. Each column (Subject X prediction) was centered so the diagonal RMSD (observation vs prediction for Subject X) is 0. Blue or red off-diagonal cells (observation Subject Y vs prediction Subject X) indicate RMSD greater or less than diagonal entry. Density plot shows diagonal RMSD elements (values as points and median as vertical line) tend to be less than off-diagonal RMSD (grey distribution), indicating high prediction specificity (One-sample Kolmogorov-Smirnov test: $D^+ = 0.79894, P < 10^{-15}, n = 28$). C) Models were trained on data closer to the time point of prediction (p36). For example, models corresponding to 'p10' were trained on p10 data and all earlier data (p03,p05,p07). As more data is included, model accuracy and specificity improved with significant specificity at p10 and above (One-sample Kolmogorov-Smirnov test: $D^+ = 0.34392, P = 0.001, n = 28$). D) Prediction Accuracy (RMSD) at p36 for each subject tends to improve over the course of neurodevelopment. When x-axis = X , subject data from ages $\leq X$ were used in the prediction. Trendlines denote patterns seen by sex and there is no significant difference in prediction accuracy between the sexes at any time (with shaded regions denoting standard error). Improvement in prediction accuracy (* $P < 0.05$, ** $P < 0.01$, *** $P < 0.001$) indicates neuroanatomy individualisation, which occurs significantly earlier in males (permutation test, $P = 0.025$).

3.5 Individualisation of neuroanatomy emerges earlier in males

Genetic and environmental variability is limited with the usage of standardized laboratory mice;

however, neuroanatomy is still remarkably individualised by adulthood. Most studies typically treat this individualisation as variability that is either accounted for as residuals or random effects in a statistical model in order to study other factors influencing variability, such as sex. We instead hypothesized that some of the variability in structure volumes of mature brains are inherently individualised. We sought to identify when this neuroanatomical individuality emerged across development, and whether it differed across sexes.

We used a set of linear mixed-effect regression models to predict structure volume at a specified time from the structure's volume at earlier times. Similar to validation methods followed by Tavor et al. [27], we used a leave-one-out approach to evaluate our model: the model for predicting volume of a structure s from a particular subject i at a time t was not trained on data containing information about any structures of subject i at time t . Figure 8A demonstrates the model's sensitivity by plotting the predicted and observed volume of three representative structures for every individual at p36. We quantitatively assessed the specificity of the model (Figure 8B) and found that predicted structure volumes for subject i generally matched observations for subject i closer than observations for other subjects. This is despite the fact that when trying to predict any subject at a time point, the model was trained on everything but the data from the subject at that time point. Yet, the prediction made is closer to the unseen subject data than the seen data from other subjects.

The model accurately captures neuroanatomical individualisation in the mature mouse brain. To investigate when this individuality emerged, we withheld more information regarding the subject to be predicted. When only considering data from p3 to predict p36 structure volumes, model specificity and accuracy is quite poor (56% probability predicted volumes match predicted subject better than other subjects; 0.13 Root-Mean-Square-Difference (RMSD)). However, when considering data from p10 and younger, specificity and accuracy improves (70%;0.12) and is quite high for data from p17 and younger (85%; 0.096mm³) (Figure 8C). Thus, only the first 10-17 days of brain development is sufficient to predict individualisation of mature mouse brain anatomy.

We plotted how the prediction accuracy (RMSD) at p36 changed for all subjects as we included more data closer to p36 (Figure 8D). As expected, accuracy increased as more data was included. Accuracy of predicting male neuroanatomy was not significantly different from predicting female neuroanatomy at any time point. However, male accuracy improved earlier than female accuracy (permutation test, $P=0.025$), needing only data from the first 7 days of life, while females required data from the first 17 days of life to improve accuracy significantly. Thus, male neuroanatomy individualises earlier than female neuroanatomy. This was also true ($P=0.034$) when we computed Root-Mean-Square-Percent-Difference (RMSPD) which is less biased against smaller brain structures (Supplementary Fig. S6). We found a similar pattern when predicting p29 ($P=0.025$) and p65 ($P=0.057$; Supplementary Fig. S7) time points, but it was no longer significant for p65. Furthermore, we also improved our model by using a random forest ($P=0.030$) and introducing a covariate for whole-brain volume ($P=0.036$) and found similar results.

5.3.8 Post-Development Neuroanatomy Prediction

We obtained structure volumes for all 182 bilateral structures in our atlas, for every subject and at every time point. When predicting the volume of a structure from subject i at time t , we excluded all data that belonged to subject i at time t to form the training set. We modelled each structure independently using weighted linear mixed-effects models and trained these models on the training set (Supplementary Section S1.1.5). Then, we used the trained models to predict the excluded data of subject i at time t and compared the predicted data for any structure j (\hat{y}_j) to the observed data for the same structure (y_j).

$$\text{RMSD} = \sqrt{\frac{1}{182} \sum_{j=1}^{182} (\hat{y}_j - y_j)^2} \quad (10)$$

$$\text{RMSPD} = \sqrt{\frac{1}{182} \sum_{j=1}^{182} \left(\frac{\hat{y}_j - y_j}{y_j} \right)^2} \quad (11)$$

Model accuracy was evaluated using root-mean-square-difference (RMSD) and root-mean-square-percent-difference (RMSPD) — the latter of which is less biased against small structures. RMSPD also corrects for whole-brain volume as normalisation to brain volume (V) results in the substitutions $\hat{y}_j \rightarrow \hat{y}_j/V, y_j \rightarrow y_j/V$, which leave Equation (11) unchanged.

We also trained two additional models to check for consistency. The first predicted structures after covarying for total brain volume. The second model used a random forest to predict structure

volumes from other structures at earlier times. All models showed similar results.

Individualisation of neuroanatomy was demonstrated by withholding information about the predicted subject prior to model training. We computed RMSD as a function of the accessed time point information x ; that is, when $x = 7$, the model must make its prediction on subject data from time points p7 and earlier (p5 and p3), and does not have access to data from time points after p7. To test if models made specific predictions, we computed the RMSD between the prediction for subject i and the observation for subject i (predicted vs self), and compared it to the RMSD between the prediction for subject i and observations for other subjects $\neq i$ (predicted vs other). Repeating this for all subject, we compared the RMSD values from predicted vs self and RMSD values from predicted vs other using Kolmogorov-Smirnov test.

As the accessed time point x increases, model predictions become better and RMSD decreases. To test for sex differences in the timing of individualisation of neuroanatomy, we fit a linear mixed-effects model to the RMSD versus the accessed timepoints x . The model had fixed effects of accessed time point, sex, their interactions and random effect of mouse. For both sexes, RMSD significantly decreased with time (the degrees-of-freedom calculated using Satterthwaite approximation [65]). We subsetted a time window over which individualisation occurs: the lower bound being the first time point where at least one sex had a significant RMSD decrease ($P < 0.05$) and the upper bound being the first time point where both sexes had a very significant RMSD decrease ($P < 0.01$). For each individual, we Z -transformed the RMSD values (subtracted their mean value over time and normalized to the standard deviation). Then, we computed the average Z -RMSD values for males and females in the time window. The difference between these two values is related to sexual dimorphisms in the timing of neuroanatomy individualisation between the sexes. To estimate the significance of this difference, we performed a permutation test by shuffling the sex labels of our data, recomputing the average Z -RMSD for males and females, and computing the difference. P -Values for the permutation test were defined by finding what fraction of the 10000 permuted differences exceeded or were equal to the true difference between average Z -RMSD male and female values.

S1.1.5 Neuroanatomy Prediction

We wanted to predict the absolute volumes of structures. To model this growth we used natural spline functions. N th-order natural splines are characterized by N basis functions of age t , where the k th basis function is represented by $f_k(t)$. For each structure, the structure volume y_{ij} in the training data was fit with the following model:

$$y_{ij} = \alpha_1 + \alpha_2 s_i + \sum_{k=1}^N \alpha_{k+2} f_k(t_{ij}) + \sum_{k=1}^N s_i \alpha_{k+N+2} f_k(t_{ij}) + \beta_{0i} + \varepsilon_{ij} \quad (12)$$

While increasing the order N of the natural splines allows one to model more complex growth curves, it can also overfit data leading to inaccurate predictions for data outside the training set. We computed Bayes Factors, using the `BayesFactor` package [6] to decide which order N of natural splines to use. Bayes Factors compute the evidence any model (12) has versus an intercept only model ($y_{ij} = \alpha_1 + \varepsilon_{ij}$) and assumes a reasonable set of priors on the predictors. We used the `BayesFactor` package's default Jeffreys prior for our analysis. By finding the Bayes Factor associated with Model (12) for values of spline order N ranging from one (linear growth) to the number of time points in the data minus one. The spline order N chosen for the structure's model is the one with the highest Bayes Factor associated with it.

Once we determined the order N of the natural splines modelling fixed growth effects, we wanted to similarly model random growth effects with M th-order natural splines and optimize M . The training data was fit with the model below:

$$y_{ij} = \alpha_1 + \alpha_2 s_i + \sum_{k=1}^N \alpha_{k+2} f_k(t_{ij}) + \sum_{k=1}^N s_i \alpha_{k+N+2} f_k(t_{ij}) + \beta_{0i} + \sum_{k=1}^M \beta_{ki} f_k(t_{ij}) + \varepsilon_{ij} \quad (13)$$

After fitting multiple models with different values for M , we chose the model with the lowest Bayesian information criterion [7] to predict structure volumes.

Lastly, we also placed a Gaussian weighting on data in the training set depending on the age. This step was motivated by the fact that when predicting volumes at a particular age, the time point closest to that age is the most informative. However, only considering the closest time point alone may be less informative than taking some information from the other time points. To balance the two extremes, we placed a gaussian weighting on the data. The gaussian weighting was centered on the time t which we want to predict and its spread (σ^2) can be optimized. A high σ^2 implies data over all time is weighted equally by the model, and a low σ^2 implies data closest to the prediction time t is weighed higher than data further away. The variance parameter was optimized using leave-one-out cross validation.

The model described above was used primarily in our study. However, we also explored two improvements to our model to check for consistency. The first was adding a covariate for total brain volume $V_{i,j}$ at the time point prior to the one being predicted ($j \rightarrow j - 1$), which was done to control for whole-brain volume effects in subjects. The model with this covariate is given below, and fixed N splines and random M splines are optimized as detailed above:

$$y_{ij} = \alpha_1 + \alpha_2 s_i + \sum_{k=1}^N \alpha_{k+2} f_k(t_{ij}) + \sum_{k=1}^N s_i \alpha_{k+N+2} f_k(t_{ij}) + \alpha_{k+2N+2} V_{i,(j-1)} + \beta_{0i} + \sum_{k=1}^M \beta_{ki} f_k(t_{ij}) + \varepsilon_{ij} \quad (14)$$

The second improvement was to use a random forest. Thus far, structures were modelled independently of each other, i.e. a structure's volume at a certain time t was predicted from the same structure's volume at earlier times. Using the random forest machine learning method, we can predict a structure's volume from other structures at an earlier time. To do so, we first fit the primary model described above to the training data and obtained the residuals of this model at the age t we wanted to predict. We then identified the volume of all 182 structures in the brain at the immediate earlier time and used them to model these residuals using a random forest from the `randomForest` package [8]. The random forest contained 500 trees and randomly sampled 5 structures at each tree split. The final predicted value was the sum of predicted values from both the initial model and the random forest.

Fifth, the authors are leading experts in the field of mouse brain anatomy. It would therefore be interesting to example how cortical thickness also varies as a function of sex across developmental time periods, particularly in light of aforementioned existing human data on sex differences in cortical thickness in infants (e.g. Li et al., J Neuroscience, 2014; doi: 10.1523/JNEUROSCI.3976-13.2014).

We thank the author for this suggestion and the complements. We decided to conduct a consistent clustering analysis over the neuroanatomical measures of relative volumes (Supplementary Figure S2), absolute volumes (Supplementary Figure S3), and cortical thickness (Supplementary Figure S4). For each measure, we selected the top 5% largest voxels or vertices in males and top 5% largest voxels or vertices in females at p65. This is to have a consistent ROI for the different measures and is different from the approach taken for relative volumes in the main paper (Figure 7). In the main paper, the approach was to take all voxels that had $FDR < 0.1$ and this approach could not be applied here as 15% of voxels from the relative jacobian analysis survived the FDR threshold as opposed to 0.1% of vertices from the cortical thickness. Therefore, we used this top-5% male, top-5% female ROI approach. Similar to the relative volume analysis, all neuroanatomical measures showed a consistent pattern where larger voxels/vertices in males tended to occur early and larger voxels/vertices in females tended to occur late. This discussion can be found in Results 3.4, methods in 5.3.7, and additional methods and results in S1.2.2.

Figure S2: Top 5% of largest voxels (relative to whole brain) in males and females clustered by their effect sizes over time. Cluster 1 and 2 correspond to regions larger in males in adulthood and these sexual dimorphisms emerge early. Cluster 3 corresponds to regions larger in females and emerges around puberty.

Figure S3: Top 5% of largest voxels in males and females clustered by their effect sizes over time. Like the case with relative volumes, cluster 1 and 2 correspond to regions larger in males in adulthood and these sexual dimorphisms emerge early. Cluster 3 corresponds to regions larger in females and emerges around puberty.

Figure S4: Top 5% of largest vertices in males and females clustered by their effect sizes over time. Like with relative and absolute volumes, cluster 1 correspond to regions larger in adult males and these dimorphisms occur early in development, while cluster 3 corresponds to regions larger in females and these dimorphisms occur around puberty. Similarly, cluster 2 in both the thickness analysis and volume analysis correspond to regions larger in males whose dimorphisms emerge after the regions in cluster 1. However, while volume analysis had both cluster 1 and cluster 2 dimorphisms emerging in the first 10 days of life, cortical thickness analysis shows regions in cluster 2's dimorphisms emerging around male puberty.

The manuscript is clearly written, although some of the statistical analyses and image registration methods may be hard to grasp for non-specialists and could be made more accessible.

We have addressed this by giving additional method details in the supplementary section targeted towards non-specialists. Please refer to Supplementary Section S1.1.2.1 and S1.1.2.2 and Supplementary Figures S11 and S12.

S1.1.2 Registration

S1.1.2.1 Affine and Non-affine Registrations

To illustrate the registration procedure, we will register the p03 average to the p05 average (top row Figure S11). The former will be called the source image and the latter the target image. The core registration procedure is composed of two stages: affine registration performed using the `mnireg` tools[1] and non-affine registration performed using the ANTs toolkit[2]. Overlaying the two images (middle image, first row), it is clear the source image needs to be distorted to fit the target image. After performing the affine registration, we use the resultant transformation to transform the source image and overlay it with the target image (middle image, second row). It is clear that the alignment has gotten better but is still unsatisfactory in some brain regions (third row). After performing the non-affine registration, the source image is transformed using the previous affine registration and this non-affine registration and this procedure results in satisfactory alignment between the two images (fourth and fifth row).

S1.1.2.2 Jacobian Determinants

Jacobian determinants are used to quantify the volumetric changes caused by deformations. In Figure S12, we illustrate the concept of absolute Jacobian determinants. Gridlines in the target image cerebellum are warped upon transformation to the source image. The Jacobian determinants of this transformation are called the absolute Jacobian determinants. It is computed for every voxel and the resulting voxel map is overlaid on the target image. This illustrates how absolute Jacobian determinants capture the extent to which regions in the source image are smaller or larger than the corresponding regions in the target. A similar procedure is applied to find relative Jacobian determinants. Gridlines in the target image cerebellum are warped upon transformation to the affine-transformed source image. The determinant of this transformations are called relative Jacobian determinants and its voxel map is overlaid on the target image. Relative Jacobian determinants capture volumetric changes after scaling brains to the same size.

Figure S11: Registration of a source image (p03 average) to a target (p05 average). The native images (after rigid alignment) are on the top row and their overlay is in the middle column. Poor alignment can be found in structures like the cerebellum where there is rapid neonatal growth. Affine registration scales and shears the source image to better align with target image. The affine transformation (generated from the affine registration) is applied to the source image and is shown in the second row. The overlay shows a good match between the affine-transformed source and the target images. However, zooming into the cerebellum of the affine-transformed source and target image (third row), shows that affine registration does not produce proper alignment of the cerebellum. This is illustrated by applying a red contour to the cerebellum of the target image and overlaying this contour to the source image. The non-affine registration corrects this discrepancy (fourth row) and produces the best alignment between source and target images (fifth row).

Figure S12: Visualizing deformations caused by transformation of target image using grids and determinants. Illustrated in the left figure, upon transformation of the target image to the source image (this transformation is the inverse of the transformation in Figure S11), gridlines in the target image become warped. In the top row, the gridlines warp from transformation to the source image; and in the bottom row, the gridlines warp from transformation to the affine-transformed source. Volumetric changes caused by the transformation can be qualitatively assessed by observing how the volume of a square region (region defined by the open space between gridlines) changes after transformation. From this, it is clear from the convergence of gridlines in the cerebellum, that much of the cerebellum decreases in size after transformation, implying that the cerebellum is smaller in the source image than the target image. Volumetric changes can also be quantified by calculating the determinants (right figure). If a region in the source image is smaller than the corresponding regions in the target image (i.e. gridlines converge), the region has determinants between 0 and 1. Conversely, regions larger in the source image (i.e. gridlines diverge) have determinants larger than 1. Absolute determinants (top row) characterize volumetric changes upon transformation from target to source images and measure the true volumetric differences between target and source images. Relative determinants (bottom row) characterize volumetric changes upon transformation from target to affine-transformed source images and measures the volumetric differences between target and source images upon removal a scaling factor (this scaling factor makes source and target images the same size as seen in Figure S11: second row). The advantage of absolute determinants is that they can be used to calculate the volumes of regions in canonical units like mm^3 . Relative determinants, on the other hand, calculate volumes relative to total brain volume instead. However, relative determinants remove whole-brain size variability (which is the largest source of variability among mice [11]) to expose more subtle variations in neuroanatomy.

Sufficient methodological data exists to test reproduction of the study. The authors may also wish to consider making the dataset publicly available for use by the mouse imaging community.

We do plan on making our datasets public. Currently our databases only contain ex vivo data, but the backend for uploading in vivo data is coming soon. We have also added statements on data and code availability.

Page 20

5.4 Data availability

Data is available upon contacting the corresponding author. We plan on publicly releasing the data on Brain-CODE (<http://braininstitute.ca/research-data-sharing/brain-code>) at some point in the future.

5.5 Code availability

Code for image registration (<https://github.com/Mouse-Imaging-Centre/pydpiper>) and statistical analysis (<https://github.com/Mouse-Imaging-Centre/RMINC>) is freely available online.

In my opinion the statistical analysis is sound, with the exception of my earlier point regarding effect sizes.

There are no ethical concerns arising.

Reviewer #3 (Remarks to the Author):

Summary of the key results

This manuscript describes the use of manganese-enhanced MRI to generate a longitudinal map of developing sexual dimorphisms in the C57BL/6 mouse brain. The authors observed that neuroanatomical structures that are larger in male mice in adulthood develop early, whereas structures that are larger in adult females develop around and after puberty. K-means clustering was used to differentiate between different patterns of sexual dimorphic development over time, suggesting 4 different modes. Lastly, the authors present an analysis of genes that demonstrated enriched expression within the sexually dimorphic clusters.

Validity:

The manuscript describes experiments that are well designed and appropriate to address the conclusions.

Originality and interest:

These findings are of particular interest to investigators working in structural imaging, sex differences, or development. I believe that they are of interest to the broader scientific community as well.

Data and methodology:

The quality of the magnetic resonance imaging is outstanding, especially considering the technical difficulties in imaging neonatal mice. The approach used to evaluate structural changes in the brain over time are appropriate for the analysis. The data are well presented, both in the paper and the supplementary data. The supplementary videos were excellent - very compelling. The methods appear to be readily reproducible, with the exception of the methodology used to calculate "spatial gene expression patterns."

Appropriate use of statistics and treatment of uncertainties:

The use of linear mixed-effect model was appropriate for this analysis.

Conclusions:

The manuscript's conclusions are sound and reached logically from the results.

Suggested improvements:

The methodology for the analysis of the colocalization of the sexually dimorphic clusters and gene expression maps from the Allen Brain Atlas is not clear and should be presented in greater detail.

Thank you for this suggestion. Additional details have been provided about spatial gene expression analysis is provided in Methods 5.3.6 and Supplementary Section 1.1.4.

Page 18 & 19

Growth rate was estimated by fitting the relative determinant at every voxel for every individual with natural spline functions of age, then differentiating the fitted function with respect to age. At every voxel, the order of the fitted natural spline was determined by finding which order minimized the Akaike Information Criterion [65]. Detailed in Supplementary Section S1.1.4, we used the Allen Brain Institute's gene expression dataset [26] to identify genes spatially enriched in our clusters [68]. Preferential spatial expression of a gene was measured using a fold-change measure: mean expression signal in an ROI (region of interest) divided by mean expression signal in the whole brain. Fold-change greater than 1 indicated gene is preferentially expressed in the ROI and fold-change less than one indicates gene is preferentially expressed outside the ROI. We estimated fold-change for every gene and tested whether genes on sex chromosomes were more likely to have higher fold-changes using the Kolmogorov-Smirnov test.

Page S3

S1.1.4 Gene Expression

The underlying gene expression changes associated with sexually dimorphic neuroanatomy (Figure 6) remains unknown. We wanted to identify candidate genes that might be associated with these dimorphisms. To do so, we used the genome-wide gene spatial expression data available from the Allen Brain Institute [4] and compared them to our neuroanatomical results. There are, however, four caveats associated with using gene expression data for our purpose.

The first caveat is that genome-wide gene expression data was only collected in males at p56. While some genes have developmental gene expression at p4, p14, and p28[5], most genes do not and there is no gene expression data for females. The second caveat is that most genes had their expression data come from only one mouse. The third caveat is that the majority of gene expression data was collected using ISH (*In situ* hybridization) on sagittal slices spanning only one hemisphere. Only a small subset of genes had ISH conducted on coronal slices spanning the whole brain. Lastly, despite extensive quality-control steps taken by the Allen Brain Institute, several regions in the brain were missing gene expression data. To compensate for this, whenever any gene had multiple replicates, we chose the replicate with the least amount of missing data. Furthermore, we excluded experiments where expression data spanned less than 20% of the brain. While these caveats do limit the conclusions made from this analysis, spatial gene expression data is still useful to identify candidate genes associated with sexual dimorphisms for further exploration.

I think it would be informative to report the mean volumes of the BNST, MeA, MPON, and PAG at each time point as illustrative of the development of these structures over time (perhaps a table?).

Mean volumes of these structures are now reported in a Supplementary Table S1.

Table S1: Mean Volume of Sexually Dimorphic Structures in Males and Females

Absolute Volume (mm ³)								
Age	Bed Nucleus of The Stria Terminalis		Medial Preoptic Nucleus		Medial Amygdala		Periaqueductal Grey	
	M	F	M	F	M	F	M	F
3	0.589	0.583	0.0856	0.0844	0.506	0.490	2.630	2.580
5	0.719	0.724	0.103	0.102	0.573	0.578	2.950	2.980
7	0.857	0.850	0.117	0.115	0.660	0.653	3.310	3.310
10	1.060	1.010	0.152	0.147	0.800	0.764	3.960	3.860
17	1.160	1.110	0.184	0.178	0.973	0.933	4.070	3.990
23	1.110	1.060	0.149	0.146	0.976	0.938	3.820	3.770
29	1.100	1.050	0.150	0.146	0.966	0.917	3.740	3.670
36	1.130	1.060	0.158	0.152	1.000	0.930	3.760	3.690
65	1.200	1.120	0.168	0.158	1.060	0.983	4.010	3.970

Relative Volume (% Brain)								
Age	Bed Nucleus of The Stria Terminalis		Medial Preoptic Nucleus		Medial Amygdala		Periaqueductal Grey	
	M	F	M	F	M	F	M	F
3	0.2810	0.2850	0.0394	0.0400	0.2420	0.2410	0.9740	0.9840
5	0.2800	0.2750	0.0397	0.0382	0.2440	0.2400	0.9830	0.9700
7	0.2810	0.2740	0.0400	0.0389	0.2450	0.2380	0.9750	0.9580
10	0.2780	0.2710	0.0396	0.0390	0.2440	0.2360	0.9590	0.9510
17	0.2760	0.2650	0.0399	0.0388	0.2440	0.2350	0.9700	0.9580
23	0.2740	0.2630	0.0393	0.0386	0.2450	0.2350	0.9620	0.9490
29	0.2700	0.2620	0.0383	0.0378	0.2430	0.2350	0.9580	0.9560
36	0.2690	0.2630	0.0380	0.0380	0.2430	0.2340	0.9400	0.9590
65	0.2770	0.2640	0.0386	0.0373	0.2430	0.2310	0.9240	0.9340

The graphs on the right of figure 3 are interesting, but unreadable. The text is invisible. This is true of several of the figures. The optimum font size suggested by the journal is 8pt. Please use this as a guideline and use a larger font size where practical.

Font size has been increased for figures throughout the document.

References:

Appropriate use of references.

Clarity and context:

This manuscript was remarkable easy to read. I did not find any grammatical errors or typos. The manuscript itself was clear and straightforward. Very well written.

Reviewer #4 (Remarks to the Author):

Longitudinal MEMRI of male and female mice spanning ages P3-P65 confirmed a number of previously established neuroanatomical sexual dimorphisms and also revealed new dimorphisms. Of particular interest is the finding that male-biased regions (male>female) appear relatively early in postnatal life (before weaning), whereas female-biased regions tend to appear relatively later, around the time of puberty. Cluster analysis identified 4 clusters of brain regions based on developmental trajectory: 1) regions larger in males throughout development; 2) regions initially male-biased that transitioned to female-bias; 3) regions initially not sexually dimorphic that became female-biased; and 4) regions larger in females throughout development. This report demonstrates the utility of MEMRI in studying developmental trajectories of brain regions in mice and contains intriguing, novel findings particularly in female-biased regions, highlighting puberty as a significant window of development for the female brain. However, the paper falls short in a number of ways that limit the conclusions that can be drawn and this reviewer's enthusiasm.

We would like to thank the reviewer for their honest criticism of our work. In response, we have made extensive modifications to our manuscript to fulfill three main purposes. Our first goal was to make our manuscript more transparent by tidying our text and graphs. This includes having better trendlines, standard error estimates, figure captions, cross-referencing, and explanation of spatial gene expression analysis, among many other changes throughout the text in direct response to the reviewer's criticisms.

Our second goal was to provide more rigour to our novel findings. As the reviewer commented, our results contain 'intriguing, novel findings', however 'It's just not clear what the relative volume measure is'. We provide more detail in Supplementary Sections S1.1.2.1, S1.1.2.2 and Supplementary Figures S11, S12 regarding just what relative volumes measure regarding neuroanatomy. We also abandoned measures like log male/female ratio, in favour of more commonly used effect sizes (Figure 6). This changed our clusters slightly but the essential findings in our manuscripts are still the same. Finally, we explored neuroanatomy using more intuitive measures like absolute volumes and cortical thickness to show that our novel findings are consistent across the various measures of neuroanatomy (Supplementary Figures S2-S4). We hope that these changes make our findings more interpretable to the reader.

Our last goal was to rekindle the reviewer's enthusiasm with our novel analysis of brain prediction. Our in-vivo longitudinal imaging throughout mouse neurodevelopment, in conjunction with the control of genetic and environmental factors afforded by mouse studies, renders it possible to explore exciting analysis in brain prediction. We have added new sections in our revised manuscript (Figure 8, Results Section 3.5, Methods 5.3.8, Supplementary S1.1.5) that investigates how well the neuroanatomy of the young mouse brain predicts neuroanatomy of the adolescent and adult mouse brain. By developing a computational model to predict mouse brain neuroanatomy, we found that only the first 10-17

days of neuroanatomical data are enough to make sensitive and specific predictions of individualised neuroanatomy at p36 and p65. As expected, if we provide the model with only p3 data and ask it to predict p36 or p65, the model will fail to make specific predictions because young brains are so similar to each other. However, the more data we provide from later in development, the better the model is at predicting mature individualised neuroanatomy. It is through a combination of many timepoints and more recent timepoints that individual differences in young neuroanatomy emerge, which the model can use to make predictions about mature neuroanatomy. Our prediction accuracies were not influenced by the sex of the predicted subject (i.e. males and females are equally well predicted), however, we found that male neuroanatomy individualised (became easier to predict) significantly earlier in development than female neuroanatomy.

Neuroanatomical phenotyping tends to be used to study how groups of organisms are similar. However, we hope that this analysis of individuality can generate progress into what makes organisms unique from the rest of the members of its group. The control of genetic and environmental factors limits the variability in mice making it easier for machine-learning tools to make individualised predictions. "Easier" in this context means using simple models and few subjects. While humans have far more variability compared to mice, with the advent of ever-bigger datasets collected on human neurodevelopment, it will be possible in the near future to use more advanced machine-learning tools to predict some individuality in these data sets. It might even be possible to predict the development of neuroanatomical pathologies associated with neurological disorders prior to disorder onset. We hope that our analysis of mouse individuality spurs research in this direction.

Major comments

1. There is a large gap, more than one month, between the last two ages at which scans were obtained, P36 and P65. Although mice may be post-pubertal by P36 on some somatic and endocrinological measures, a significant degree of brain and behavioral maturation occurs during the adolescent period between P36 and P65. This gap raises some concern about missing information during an important developmental window, and necessarily means that the temporal resolution for changes in brain region volume is much poorer for the pubertal/adolescent period than for the neonatal/prepubertal period in this study. The authors should comment on how this disparity in temporal resolution could alter findings, models, or cluster analysis. For example, what should we make of the curvilinear nature of relative volumes between P36 and P65 in the absence of data during that time (e.g. Fig 6D)?

The reviewer makes an important point about the way we sampled temporal information---with a greater emphasis on neonatal/prepubertal period compared to pubertal/adolescent period---and how this could alter our findings. There are two main ways this happens. First, registration efficiency drops to unacceptable levels in the pubertal/adolescent time period thereby confounding all volumetric analysis. Second, the temporal regularization employed by the statistical analysis does not perform adequately with this sparse sampling. We will address both these possibilities in this response.

It is important to note that in addition to the scientific interest we have regarding mouse brain neonatal neuroanatomy, the dense sampling of neonatal timepoints allows us to maintain optimal registration efficiency (Szulc et al 2015). Due to the high growth rate and growth rate variability (See Figure 6), the brain topology changes rapidly in early development. This challenges registration algorithms like ours that try to perform topology-preserving transformations to map all subjects to the same space. If the brain changes too rapidly, automated registration of the whole brain becomes impossible. Sampling densely ensures that registration can always make topology-preserving transformations. However, sampling too densely would over-expose mice to manganese and handling, thereby confounding our results. Szulc et al (2015) measured registration efficiencies of neonatal brains using the kappa statistic. Given two segmentations for the same brain, kappa statistic measures what fraction of the average volume of both segmentations is the overlap of both segmentations. Kappa statistic of 1 implies perfect overlap between segmentations (best registration) and statistic of 0 implies no overlap (worst registration) with statistic values above 0.75 being the threshold for good registration. Szulc et al (2015) showed that the optimal spacing to preserve registration efficiency for the cerebellum, which has the most rapid alterations in topology during neonatal development, is about 2-3 days. This is the temporal resolution we chose in neonatal life. We performed a similar analysis using kappa statistics on our data and found kappa of 0.87 for the registration of the p3 to p5 brain, indicating a good registration. Post-puberty, even though there is still a significant amount of neurodevelopment still happening, most of the brain changes do not involve rapid alterations in topology making registrations, even across distant timepoints robust. For example, the kappa statistic for registration between p36 and p65 is 0.96 indicating excellent registration. Despite the large time gap between p36 and p65, the registration is of comparable quality to p5 and p3. We thus conclude that registration is as effective at capturing changes in the p3 to p5 changes as it is at capturing changes from p36 to p65. We opted not to include discussion on registration efficiencies in our manuscript as it has been extensively explored in literature and requires technical understanding beyond the scope of this paper.

We now turn our attention to the effect of time sampling on statistics. As the reviewer eluded to, statistical smoothing of our temporal data is inherently dependent on the sampling of the data. Our lax attitude towards this fact resulted in our inappropriate implementation of the default trendline function in ggplot2 (the R package used for plotting), which uses Local Polynomial Regression Fitting. This, in turn, created plots with nonsensical trajectories between p36 and p65, where there was no data available. The reviewer correctly criticized this and we have made changes to Figure 6D in response. We determined a more sensible trendline and standard error by using linear mixed-effects models with time point as the predictor instead of age. Unlike age, timepoint is evenly spaced but with the trade-off being it costs more degrees of freedom to model. This sensible trendline and standard error regions are used throughout the manuscript. We also reproduced the key Figure 4 findings after using timepoints instead of ages (Supplementary Figure S15) and excluding p65 from the analysis (Supplementary Figure S16). Thus, we conclude that our statistical estimates are not influenced by time point spacing. The k-means clustering analysis in Section 3.4 implicitly assumes timepoints instead of ages during the clustering process and is therefore not dependent on the timespan between timepoints.

Figure 6: Coordinated growth of sexually dimorphic functional networks. Sexually-dimorphic voxels with similar effect sizes through time were clustered into 4 groups using k -means. A) Results of the clustering analysis on sexually dimorphic voxels. Effect sizes (positive is bigger in males, and negative is bigger in females) of B) Relative volumes and C) Growth rate for the different clusters. D) Average volume and growth rate in each cluster for each individual. Cluster 1 corresponds to regions larger in males and this dimorphism emerges early in development. Regions involved in the vomeronasal system, which processes pheromonal information, are found in this cluster. Cluster 2 are also regions larger in males but the onset more delayed. However, this cluster in early life shows strong bias in growth rate towards males. Parts of this cluster include the olfactory bulb and pallidum. Cluster 3 voxels trajectory switches from being larger in males in early life, to larger in females post-puberty. Parts of the sensory cortex and PAG belong to this cluster. Cluster 4 contains regions that are not sexually dimorphic in early life but become larger in females over the course of development. This cluster includes association related areas such as parts of the central thalamus and temporal association cortex; and motor related areas such as parts of the hindbrain, cerebellum, caudoputamen, and motor cortex. Both Cluster 3 and 4 show growth rate bias towards females peaking around puberty.

Figure S15: Sexually dimorphic voxels when using time point instead of age as a predictor. Our time point are not evenly spaced in development with more time points concentrated in early life. We assessed whether this has an effect of the sexually dimorphic regions identified by using time point as a predictor, which, unlike age, would be evenly spaced. We found more sexually dimorphic voxels compared to Figure 4, but many of the same regions are implicated in both figures.

Figure S16: Sexually dimorphic voxels in the mouse brain after removal of all data corresponding to p65. The figure was generated in a similar manner to Figure 4, except all data from p65 was removed prior to statistical analysis. This was done as p65 is almost a month after the previous p36 time point and we wanted to assess whether this type of sampling would have any effect. Since the results are similar to Figure 4, we conclude that this does not for our study.

Reference

Szulc, Kamila U., et al. "4D MEMRI atlas of neonatal FVB/N mouse brain development." *Neuroimage* 118 (2015): 49-62.

2. The authors draw parallels between earlier emergence of male-biased brain regions and male-biased neurodevelopmental disorders and later emergence of female-biased brain regions and psychiatric disorders. (1) This begs the question of what brain region size really means (if anything) in the context of these disorders. This question is not directly addressed; rather the authors frame the issue as periods of relative change in brain region size being periods of vulnerability (discussion p 9-10). This framing does make sense, but it also seems to ignore the fact that the findings are largely centered around sex differences in brain region volume over time, which does not provide much insight into relative change. Growth rates, which would be indicators of relative change, are shown only for clusters in Fig 6. (2) Authors should be more explicit in how they view the relationship between the developmental trajectory of sexual dimorphisms based on brain region size and the developmental trajectory of disorders that disproportionately affect males or females.

(1) Convergence between any particular sex difference in anatomy and a sex difference in behaviour or psychiatric disorder symptom cannot be guaranteed. But interesting recent research have utilized normally developing sex differences in anatomy in order to predict or identify females who have autism, for example (Ecker et al 2017, Lai et al, 2013). This indicates that in the case of some psychiatric disorders that show sexual dimorphisms, they are related to sex differences in brain anatomy or an atypical movement away from characterized sex differences in the brain. Our data strengthens the understanding of normally developing sex differences with the added temporal context we provide.

(2) We did not explicitly test for behavioural outcomes across development that may be related to the sexually dimorphic changes in neuroanatomy that we observe in our study, thus rendering it difficult to be specific about this relationship. However, neuroanatomical change can be driven by a multitude of reasons: changes in the numbers and/or morphology of cells, types of cells, vasculature, and more. Given that our results with relative volumes correlate highly with absolute volumetric data, and that absolute volumetric data is driven by any number of these cellular events, we can conclude that changes in brain shape across time indicate changes in cellular processes across time, and that these events can happen more across different times in males and females. The processes that underlie both normal development and mental illness rely on some sort of change in the brain, and periods where there is increased levels of change can act as periods of higher vulnerability. Thus, the differences in timing of relative change in the brain across males and females serve as sex-specific opportunities where predispositions to specific stimuli/insults/processes can shift the likelihood of a particular behaviour or outcome to one sex or the other (McCarthy 2016).

References

Ecker, C., Andrews, D. S., Gudbrandsen, C. M., Marquand, A. F., Ginestet, C. E., Daly, E. M., ... & Bullmore, E. T. (2017). Association between the probability of autism spectrum disorder and normative sex-related phenotypic diversity in brain structure. JAMA psychiatry, 74(4), 329-338.

Lai, M. C., Lombardo, M. V., Suckling, J., Ruigrok, A. N., Chakrabarti, B., Ecker, C., ... & MRC AIMS Consortium. (2013). *Biological sex affects the neurobiology of autism*. *Brain*, 136(9), 2799-2815.

McCarthy, M. M. (2016). *Sex differences in the developing brain as a source of inherent risk*. *Dialogues in clinical neuroscience*, 18(4), 361.

Page 9

MRI-detectable change in brain structure represents underlying cellular or molecular processes. Our results show that periods of relative neuroanatomical change differ across development for males and females, and thus may reflect a difference in cellular or molecular processes between males and females across developmental times. Periods of relative change in the brain can be considered, then, both as windows of brain development and windows of vulnerability when development goes awry [30]. The differences in timing of relative change in the brain across males and females serve as sex-specific opportunities where predispositions to certain stimuli, insults or processes can shift the likelihood of a particular behaviour or psychiatric outcome to one sex or the other.

3. I found figure captions to be generally lacking in important details. Authors should examine them carefully and add information, either to the captions or appropriate results section, so that the reader can clearly understand what's in the figures.

Noted. Figure captions have been revised to be more relevant.

a. Fig 1: The box plots need to be explained in better detail, specifying what the line within each box represents (the median?), and what the whiskers and data points represent. Fig 1A is a photo of custom holders for neonatal mice; the mouse pictured does not look neonatal—has hair, looks relatively large, e.g.

We replaced these plots and now consistently plot the mean and use whiskers and shaded regions to represent standard error estimated using linear models. This image (Fig 1A) has been replaced with a new image of a younger mouse that is more representative of a neonatal mouse - it is a postnatal day 3 mouse, which has no hair, and is much smaller than the mouse in the previous image. We thank the reviewer for pointing this out.

Figure 1: Scanning apparatus, and scanned mice (S) vs. non-scanned littermates (L) data. (A) Custom 3D printed holders for neonatal mice. (B) Up to seven mice at a time were scanned using a saddle coil array. (C) Body weight was not significantly different between scanned and unscanned mice ($\chi^2_2 = 0.13, P = 0.94$). There was also no interaction effect of scanning and sex ($\chi^2_1 = 0.097, P = 0.75$). Trendlines and bars for standard error were calculated using centered linear mixed-effect models. (D) Repeated scanning did not have a significant effect on puberty onset ($F_{3,64} = 24.67, t_{64} = 1.56, P = 0.12$) but did have a significant effect on weight at puberty ($F_{3,64} = 10.67, t_{64} = -2.43, P = 0.02$). However, neither measure had a sex-scanning interaction ($t_{64} = -0.045, P = 0.96$ and $t_{64} = 0.451, P = 0.65$). (E) Scanning did not have a significant effect on the levels of sex hormones estradiol ($F_{3,48} = 1.11, t_{48} = 0.16, P = 0.87$), testosterone ($F_{3,48} = 4.17, t_{48} = 0.32, P = 0.15$), LH ($F_{3,48} = 0.68, t_{48} = 1.03, P = 0.31$), FSH ($F_{3,48} = 14.19, t_{48} = 0.28, P = 0.78$). (F) Organ weight of testes ($F_{1,18} = 1.05, t_{18} = -1.0, P = 0.32$), ovaries ($F_{1,28} = 0.03, t_{28} = 0.17, P = 0.87$), and uteri ($F_{1,28} = 0.17, t_{28} = -0.41, P = 0.69$) were not affected by repeated scanning. Means and bars for standard error for B, C, and D were estimated using linear models.

b. Fig 3: Please state specifically that the red and green dots/lines in the relative volume are data for individual mice. The relative volume measure is somewhat confusing. Figure caption states that “relative volume corrects for whole-brain size differences between subjects and is expressed as a percent difference from the average volume of the brain structure”. If the black line at 0 is the average volume of the brain structure, then how can both males and females be below the average MPON or PAG volume at P65? It’s just not clear what the relative volume measure is and how to interpret these graphs.

This was an error on our part. Relative volumes actually represent volume fraction (volume of structure divided by volume of brain). This has been corrected everywhere.

Figure 3: MEMRI captures sex differences in brain structure sizes. A) Sagittal and coronal slices of the average p65 brain showing segmentations of mouse brain structures: Bed Nucleus of the Stria Terminalis (BNST), Medial Preoptic Nucleus (MPON), Medial nucleus of the Amygdala (MeA), and Periaqueductal Grey (PAG). B-E) the absolute and relative volumes of these structures over time. **Points in these plots represent measurement at a time point for individual mice and lines connect the measurements for the same mouse over time. Shaded regions represent standard error estimated using linear mixed-effects models.** Relative volume corrects for whole-brain size differences between subjects and is expressed as a percent difference from the average volume of the brain structure. Using linear mixed effects models, we recapitulate known canonical sex differences in absolute volumes of the MeA ($\chi^2_5 = 66, P < 10^{-10}$), BNST ($\chi^2_9 = 77, P < 10^{-12}$), and MPON ($\chi^2_9 = 28, P < 10^{-3}$). Sex differences in these structures emerge pre-puberty, at around p10. Using relative volumes to correct for whole brain size, we see that sex differences in these structures are preserved but differences emerge very early in development around p5. PAG relative volume also shows a significant effect of interaction between sex and age ($\chi^2_8 = 21, P < 10^{-2}$), which is not found in the absolute volumes ($\chi^2_8 = 10, P = 0.2$). Taken together, these results indicate that relative volumes obtained by MEMRI are a sensitive marker for detecting canonical and novel sexual dimorphisms in neuroanatomy.

B) Bed Nucleus of The Stria Terminalis

C) Medial Preoptic Nucleus

D) Medial Amygdala

E) Periaqueductal Grey

c. Fig 7. Does the reference to gene expression at P56 reflect the age at which Allen Brain Institute data are from? Are gene expression data overlaid onto P65 images from the current study?

Yes. Expression data was overlaid onto our P65 average. Since all images are registered to the p65 average, we can relate our neuroanatomical statistics to gene expression data from the Allen Brain Institute.

d. Figure captions for all supplementary figures need to be expanded to clearly describe what's being shown in the figures. Supplementary figures should be specifically referred to in the results section at the point where they are needed. I could find no specific references except to the supplemental movies. Also the top of p 8 refers to a full list of genes in the appendix—I found no appendix.

The supplementary sections have been expanded and we have added the appendix. Apologies and thank you for bringing this to our attention.

4. The gene expression data are not particularly useful, as presented. Results refer to spatial gene expression data from the Allen Brain Institute “which has genome-wide gene expression maps in the adult male mouse brain” (p 7, bottom). Yet expression patterns for some genes are shown for P4, P14, P28, and P56; presumably these are from the ABI developmental data base. Are expression patterns for all ages based solely on males? If so, then these data are not especially informative, as what one would really like to know is whether expression patterns differ in males and females at particular time points. What are we to make of the gene expression data based on males as they relate to female-biased brain regions? At the very least, gene expression results need to be discussed in light of this limitation. Also, the expression of some genes at some time points appears to be strikingly unilateral—is that really the case, and if so, what does that mean (e.g., Supp figs 4 and 5, Gabrq on P4, P14, P28)?

As the reviewer pointed out, there are several limitations in the use of spatial gene expression analysis in neuroanatomical differences between males and females. The first caveat is that genome-wide gene expression data was only collected in males at p56. While some genes have developmental gene expression at p4, p14, and p28, most genes do not and there is no gene expression data for females. The second caveat is that most genes had their expression data come from only one mouse. The third caveat is that the majority of gene expression data was collected using ISH (In situ hybridization) on sagittal slices spanning only one hemisphere. Only a small subset of genes had ISH conducted on coronal slices spanning the whole brain. Lastly, despite extensive quality-control steps taken by the Allen Brain Institute (ABI), several regions in the brain were missing gene expression data. To compensate for this, whenever any gene had multiple replicates, we chose the replicate with the least amount of missing data. Furthermore, we excluded experiments where expression data spanned less than 20% of the brain. All of these caveats have been explicitly noted in Supplementary Section S1.1.4. While these caveats do limit the conclusions made from this analysis, we will proceed to explain how spatial gene expression

data is still useful to identify candidate genes associated with sexual dimorphisms for further exploration.

Page S3

S1.1.4 Gene Expression

The underlying gene expression changes associated with sexually dimorphic neuroanatomy (Figure 6) remains unknown. We wanted to identify candidate genes that might be associated with these dimorphisms. To do so, we used the genome-wide gene spatial expression data available from the Allen Brain Institute [4] and compared them to our neuroanatomical results. There are, however, four caveats associated with using gene expression data for our purpose.

The first caveat is that genome-wide gene expression data was only collected in males at p56. While some genes have developmental gene expression at p4, p14, and p28[5], most genes do not and there is no gene expression data for females. The second caveat is that most genes had their expression data come from only one mouse. The third caveat is that the majority of gene expression data was collected using ISH (*In situ* hybridization) on sagittal slices spanning only one hemisphere. Only a small subset of genes had ISH conducted on coronal slices spanning the whole brain. Lastly, despite extensive quality-control steps taken by the Allen Brain Institute, several regions in the brain were missing gene expression data. To compensate for this, whenever any gene had multiple replicates, we chose the replicate with the least amount of missing data. Furthermore, we excluded experiments where expression data spanned less than 20% of the brain. While these caveats do limit the conclusions made from this analysis, spatial gene expression data is still useful to identify candidate genes associated with sexual dimorphisms for further exploration.

In the paper we mention that the spatial gene expression analysis used in this manuscript has previously been done in literature in the context of single-gene mutation mouse models of autism. In several mouse models of autism with single gene mutations, significant neuroanatomical differences between mutants and wildtype controls often occur in regions where the mutated gene would be expressed in a wildtype mouse (Fernandes et al 2017). Despite not having information about gene spatial expression in mutant mice, it is still possible to make correlative statements about the gene expression changes driving the neuroanatomical phenotype. In the same way, despite not having gene spatial expression in female mice, it is equally possible to make correlative statements about the gene expression changes driving the neuroanatomical sexual dimorphisms.

The reviewer is correct in the assertion that ‘what one would really like to know is whether expression patterns differ in males and females at particular time points’. While extensive research has been done in males mouse brains looking at both the spatial expression of genes and their time course, the research on female is sorely lacking in comparison. Our main goal in this analysis is to motivate the community to address this shortcoming. Even with just the gene spatial expression data limited to male mice, we found strong associations between spatial expression patterns in the brain and sexually dimorphic neuroanatomy. The fact that we found a significant spatial bias for genes expressed on sex chromosomes (Figure 7) in sexually dimorphic neuroanatomy indicates these affected brain regions are associated with sex-linked gene expression. Research typically focuses on gene expression differences in a few key sexually dimorphic structures, or in the whole brain around the

embryonic stage (Dewing et al 2003). We hope that these results will motivate acquisition of spatial gene expression data for female mouse brains.

The spatial gene expression analysis that the ABI did for male mice would be difficult to replicate in female mice. To aid the community, we identified candidate genes that might be important for study. Many of the candidate genes identified, such as *Esr1* and *Esr2*, not only have a preferential expression bias in sexually-dimorphic neuroanatomy, they are also functionally associated with sex processes, and therefore represent strong candidates for spatial gene expression analysis in females. Furthermore, genes with the most spatial expression bias--*Slc6a4* and *Tph2*--are known to be associated with anxiety behaviours and the former's expression changes drastically in development (Figure 7). Given that we have found differences in the timing of sex dimorphisms, it is also worth studying the time-course of *Slc6a4* expression in females and contrast it with the already known male-expression.

In conclusion, we agree with the reviewer that what is truly informative in gene expression analysis is expression patterns differing in males and females at particular time points. However, with our spatial gene expression and neuroanatomy results, we have provided candidate genes - with spatial expression bias in regions associated with sexually dimorphic neuroanatomy - as a starting point for the community to further explore this.

Page 11

Overlaying gene expression maps onto our images provides insight into the underlying causes that drive our MRI results. Genes *Esr1* and *Esr2*, which encode for estrogen receptors, as well as *Slc6a4*, which encodes the serotonin transporter, were amongst the most preferentially expressed genes from our analysis. *Esr1* and *Esr2* are known to be involved in sexual differentiation of the brain and behaviour [14], while variants of *Slc6a4* have been implicated in disorders that show sex bias in type and in age of onset [55]. Expression of these genes also changes across development and are sensitive to hormones [56]. Such comprehensive gene expression datasets currently only exist for males and not females [26]. However, since these genes were preferentially expressed in sexually dimorphic areas, further investigation of how these candidate genes are expressed in females, and how they drive sexual dimorphisms is warranted. Additionally, more direct investigations — such as in-depth histology — of the cellular underpinnings of our mesoscopic neuroanatomical changes would be useful. Our results provide indications of candidate genes and areas to explore in further detail.

References

Dewing, Phoebe, et al. "Sexually dimorphic gene expression in mouse brain precedes gonadal differentiation." *Molecular Brain Research* 118.1-2 (2003): 82-90.

Fernandes, D. J. et al. *Spatial gene expression analysis of neuroanatomical differences in mouse models.* *NeuroImage* 163, 220–230 (2017).

Other comments

5. The concentrations of E2 and testosterone shown in Fig 1 are puzzling and raise questions about the assays. For example, are testosterone levels in P66 males really 0.005 ng/ml? In most assays, that would be undetectable. It also seems odd that circulating E2 levels are somewhat higher in non-scanned males than in non-scanned females. More details on the hormone assays should be provided: are these RIAs? What is the minimum detectable hormone level? What are the inter- and intra-assay coefficients of variation?

Apologies for this mistake and thank you for noticing this.

- 1) These levels are incorrect. We discovered that we had accidentally divided the amounts of hormone by the volume of the plasma sample, when the amounts were already in units of ng/ml or pg/ml. This has now been corrected in the figures. More specific details on the assay have also been included in the methods section.*
- 2) We did not read the data file properly into the statistical analysis software as it was in a format we were not used to. Upon including the missing data we found that E2 levels in non-scanned males are lower than females.*

Page 13 & 14

scale accurate to 0.1 mg. Blood samples were sent to The Endocrine Technologies Support Core at the Oregon National Primate Research Center (Beaverton, OR). Barring samples of insufficient size, all samples were analysed for estradiol, testosterone, follicle-stimulating hormone (FSH) and luteinizing hormone (LH). Estradiol and testosterone levels were measured with extraction-chromatography RIA, with an intra-assay CV of 14.7% and 3.3%, respectively. Assay sensitivity was 5 pg/ml for estradiol, and 0.2 ng/ml for testosterone. LH and FSH were analysed with RIA, with an intra-assay CV of 9.9% and 3.0 %, respectively. All experiments were approved by The Centre for Phenogenomics Animal Care Committee.

6. Please clarify the distinction between the medial amygdala in cluster 1 and the amygdala in cluster 2. Does the cluster 2 amygdala refer to basolateral, central, cortical amygdala? Something else?

Apologies for the confusion. The “amygdala” (as in cluster 2) refers to all nuclei of the amygdala except for the medial amygdala, which was segmented out as a separate structure. This difference has been made explicit in the text.

Page 6

the cerebellum, midbrain, pons, medulla, PAG, thalamus, hypothalamus, hippocampus, amygdala (defined in our atlas as all amygdalar nuclei except the MeA), caudoputamen, BNST and olfactory bulbs (OB).

7. Last sentence of middle paragraph on p 10 states that “...this sex difference is dependent on neonatal circulating estradiol during the critical period...”. This is not exactly right; it’s not circulating estradiol that’s different in males and females—it’s

circulating testosterone. Testosterone is aromatized to estradiol locally in the brain of males, which does not get translated into sex differences in circulating estradiol

Thank you for this comment. This sentence has been corrected to indicate that there are differences in circulating testosterone, and not estradiol.

Page 8 & 9

In neonatal life, males have high levels of circulating testosterone, which becomes aromatized to estradiol; this estradiol plays a crucial role in the sexual differentiation of the brain and modulates many cellular processes [14]. A vulnerability for neurodevelopmental disorders is conferred to

8. Methods, p 8: please clarify whether the two pups from each litter that were selected for scanning were one male and one female, or not.

The two pups from each litter that were selected for scanning consisted of one male and one female.

Page 13

Two pups from each cage (one male and one female) were used for longitudinal scanning. The pups were randomly selected by the experimenter with the only restriction being that they were of dissimilar sex—however, no formal randomization procedure was used. Experimenter was not blind to sex. Both scanned and non-scanned littermates were weighed either on the day of scanning, or

Reviewer #5 (Remarks to the Author):

Qui, Fernandes and colleagues have imaged the developing C57BL/6J mouse brain from postnatal day 3 at 9 time points to young adulthood (P65). They replicated size differences in three subcortical areas known to have male sex differences from many existing studies. Then they categorized relative sex differences in other parts of the brain making generalizations about when sex differences appear in various clusters of neural structures. Technically, the study is competently done. The topic is also of interest - certainly there is more to learn about sex differences in the rodent brain. The problem is size does not tell us what we need to know at this point.

Structural and functional MRI have been a great advance for the study of humans where there are inherent limitations in being able to investigate the cellular and molecular underpinnings of sex differences due to issues of brain preservation and availability of postmortem tissue. Animal models are useful because they allow examination of these levels, and currently much is known about many cellular phenomena underlying sex differences including their developmental time course and many of their connections. This makes the approach in the present study, though elegant, not particularly useful.

Thank you for this comment. The advantage of animal models is indeed that they allow us to elucidate cellular and molecular mechanisms and processes related to sex differences, of which we could not possibly do in humans. Thus, there is a tremendous wealth of cellular and molecular information garnered from animal studies. However, there are two significant benefits to studying sex differences in the brain with a technique like manganese-enhanced MRI (MEMRI). First, MEMRI is uniquely able to provide whole-brain coverage in the in-vivo mouse, and can thus be used to repeatedly investigate changes across the whole brain in the same individual across multiple time points. Histology and other mechanisms can only be done at a singular time point, and cannot provide whole-brain coverage as extensive as can be done with MRI. Secondly, MRI in mice provides a crucial link to human studies by providing a bridge between microscopic cellular and molecular data from animals to anatomical shape and size information in the human brain. With humans, macroscopic changes in the brain can be readily assessed with MRI, but more microscopic changes are difficult to investigate due to lack of postmortem tissue and preservation, as previously mentioned. The advantage of using rodent animal models is that one can perform in-depth histological and molecular investigations of the brain, in addition to macroscopic anatomical investigations which are more readily translatable to humans. Thus, this technique offers a critical bridge between these two types of information about the brain.

Furthermore, the cellular underpinnings are the essence of the overall sex difference in brain size in many species (never explicitly reported here). Examining sex differences relative to overall brain size obviates what makes the size different. Each neural area is a composite of numbers of neurons, types of glia, the dendritic tree, incoming axons etc., each contributing to the size of the neural area which in turn contribute to the size of the brain. Any quantification of, for example, the number of synapses is found by multiplying the density of synapses (synapses/volume) by the

volume of the structure. Absolute volume is the only meaningful entity. The researchers examining humans has gotten caught up with relative volume, partly because they do not have the opportunity to get to the cellular level and it is an easier political message for the general public. Neither of these are excuses for the animal literature. Neuropsychiatric disorders are due to problems at the cellular and molecular levels, some of which may manifest themselves in size, but size is not the problem per se.

We thank the reviewer for their detailed explanation regarding the shortcomings of relative volumes in the goal of inferring cellular underpinnings of sex differences. There are two main criticisms put forth by the reviewer: relative volumes are less meaningful than absolute volumes, and neuroanatomy does not leverage the strength of mice as models for investigating cellular underpinnings. We will proceed to address both these criticisms.

Absolute volumes are built from local microscale properties such as neurons, glia, and the dendritic tree. It is the sum of these microscale properties that determines the absolute volume of a region and ultimately contributes to the size of the brain. Relative volumes, on the other hand, normalise to whole brain size and, therefore, don't have any trivial local influences. The reviewer posits, therefore, that absolute volumes are a more meaningful mesoscale quantity than relative volumes. This criticism is valid; however upon closer examination of neuroanatomical data, subtle nuances exist that strengthen the case for the usage of relative volumes.

In order to evaluate whether absolute volumes or relative volumes are more useful to study sex differences in the brain, we employed techniques in machine learning to develop a classifier that predicts sex from neuroanatomical structure volumes at a particular age. First, we excluded one mouse's neuroanatomy data and ran a LASSO feature selection to train a classifier on the remaining data to predict sex from neuroanatomy. Once trained, the classifier was then provided the excluded mouse's data to test whether it could successfully predict the excluded mouse's sex. We repeated this process for every mouse and every age, training a unique classifier each time. Importantly, each classifier was always assessed for accuracy on data it had not seen during training. We trained one set of classifiers to predict sex from absolute volumes and another set to predict sex from relative volumes. Unpublished Figure 1 shows the accuracy of the classifier and we see that for most ages, classifiers trained using relative volumes predicted sex better than those trained with absolute volumes. More importantly, relative-based classifiers vastly out-performed their absolute-based counterparts at the early ages between p3-17. While absolute volumes may have a favorable interpretation, an unbiased machine learning procedure favors studying sex differences in terms of relative volumes. Absolute volumes may actually be biased against sex differences at the early ages between p3-17.

To understand why the classifier performs better with relative volumes over absolute volumes, it is instructive to look at Unpublished Figure 2. We plotted the Coefficient of Variation (CV) over time for the Bed Nucleus of the Stria Terminalis (BNST) measured using absolute and relative volumes. CV is the standard deviation divided by the mean and is therefore unitless. In the BNST, as well as other brain regions, we see a clear pattern where

the young brain has high CV in their absolute volumes but remarkably stable CV in relative volumes, therefore explaining why classifiers based on relative volumes outperformed their absolute counterparts at young ages. This consistent low CV is what makes relative volumes so useful in studying sexual dimorphisms. The young brain has quite a lot of variability in growth between subjects that relative volumes effectively correct for. Once corrected, the subtle neuroanatomy that distinguishes males and females becomes clearer to both the LASSO-based classifiers, and to our statistical models. This also explains why the emergence of sexual dimorphisms in canonical sexually dimorphic structures (like the BNST) is so delayed in absolute volumes compared to relative volumes and literature studies regarding cell counts (Figure 3 in our manuscript). In summary, relative volumes are useful in studying sex differences over time as they have relatively consistent coefficients of variations, unlike absolute volumes whose coefficient of variation is quite high in the young brain.

Unpublished Figure 1: Relative volumes predict sex better than absolute volumes especially before weaning (p21).

Unpublished Figure 2: Relative volumes have low coefficient of variation over the neurodevelopment time period.

The reviewer noted that absolute volumes have local determinants, while relative volumes do not. While it is certainly true that relative volumes may have nonlocal determinants, we find that nonlocal effects are either exceedingly small in a mouse or rare in a proper image registration procedure. To illustrate relative volumes, we have placed two new figures in the Supplementary (Supplementary Figure S11 and S12). We showed the relative and absolute determinants for a registration of the p3 average and the p5 average. In both relative and absolute determinants, there is a great deal of agreement as to which structures have a high degree of change from p5 to p3 (cerebellum, cortex, olfactory bulb). Therefore, local determinants that affect absolute determinants would affect relative volumes as well.

Figure S11: Registration of a source image (p03 average) to a target (p05 average). The native images (after rigid alignment) are on the top row and their overlay is in the middle column. Poor alignment can be found in structures like the cerebellum where there is rapid neonatal growth. Affine registration scales and shears the source image to better align with target image. The affine transformation (generated from the affine registration) is applied to the source image and is shown in the second row. The overlay shows a good match between the affine-transformed source and the target images. However, zooming into the cerebellum of the affine-transformed source and target image (third row), shows that affine registration does not produce proper alignment of the cerebellum. This is illustrated by applying a red contour to the cerebellum of the target image and overlaying this contour to the source image. The non-affine registration corrects this discrepancy (fourth row) and produces the best alignment between source and target images (fifth row).

Figure S12: Visualizing deformations caused by transformation of target image using grids and determinants. Illustrated in the left figure, upon transformation of the target image to the source image (this transformation is the inverse of the transformation in Figure S11), gridlines in the target image become warped. In the top row, the gridlines warp from transformation to the source image; and in the bottom row, the gridlines warp from transformation to the affine-transformed source. Volumetric changes caused by the transformation can be qualitatively assessed by observing how the volume of a square region (region defined by the open space between gridlines) changes after transformation. From this, it is clear from the convergence of gridlines in the cerebellum, that much of the cerebellum decreases in size after transformation, implying that the cerebellum is smaller in the source image than the target image. Volumetric changes can also be quantified by calculating the determinants (right figure). If a region in the source image is smaller than the corresponding regions in the target image (i.e. gridlines converge), the region has determinants between 0 and 1. Conversely, regions larger in the source image (i.e. gridlines diverge) have determinants larger than 1. Absolute determinants (top row) characterize volumetric changes upon transformation from target to source images and measure the true volumetric differences between target and source images. Relative determinants (bottom row) characterize volumetric changes upon transformation from target to affine-transformed source images and measures the volumetric differences between target and source images upon removal a scaling factor (this scaling factor makes source and target images the same size as seen in Figure S11: second row). The advantage of absolute determinants is that they can be used to calculate the volumes of regions in canonical units like mm^3 . Relative determinants, on the other hand, calculate volumes relative to total brain volume instead. However, relative determinants remove whole-brain size variability (which is the largest source of variability among mice [11]) to expose more subtle variations in neuroanatomy.

In summary, relative volumes remove the high degree of neuroanatomical variation in the young brain allowing better modelling of neuroanatomical phenotypes associated with sex. Relative volumes also have similar local determinants as absolute volumes. Finally, as seen throughout our manuscript, relative volumes scale away the overall growth patterns that occur with neurodevelopment, making it easier to model phenotypes and conduct statistics. In the revised manuscript, we computed similar statistics for absolute determinants and cortical thickness to show that the pattern we discovered regarding early-male, late-female changes holds across the different measures of neuroanatomy.

The second main criticism of the reviewer is the fact that mouse models are useful for the study of cellular mechanisms and we did not use them in this manner. We were inspired by the tremendous insight that cellular mouse research has provided us about the underpinnings of sexual dimorphisms in the brain. While we are not able add to it directly, we have identified novel spatio-temporal patterns in the whole brain through the course of development. We hope that this, in turn, could inspire novel research into the cellular

mechanisms of sexual dimorphisms in structures and time windows that are not as highly explored, such as the such periaqueductal grey around puberty.

Our in-vivo longitudinal imaging throughout mouse neurodevelopment, in conjunction with the control of genetic and environmental factors afforded by mouse studies, renders it possible to explore exciting analysis in brain prediction. We have added a new section in our revised manuscript (Figure 8, Results Section 3.5, Methods 5.3.8, Supplementary S1.1.5) that investigates how well the neuroanatomy of the young mouse brain predicts neuroanatomy of the adolescent and adult mouse brain. By developing a computational model to predict mouse brain neuroanatomy, we found that only the first 10-17 days of neuroanatomical data are enough to make sensitive and specific predictions of individualised neuroanatomy at p36 and p65. As expected, if we provide the model with only p3 data and ask it to predict p36 or p65, the model will fail to make specific predictions because young brains are so similar to each other. However, the more data we provide from later in development, the better the model is at predicting mature individualised neuroanatomy. It is through a combination of many timepoints and more recent timepoints that individual differences in young neuroanatomy emerge, which the model can use to make predictions about mature neuroanatomy. Our prediction accuracies were not influenced by the sex of the predicted subject (i.e. males and females are equally well predicted), however, we found that male neuroanatomy individualised (became easier to predict) significantly earlier in development than female neuroanatomy.

Neuroanatomical phenotyping tends to be used to study how groups of organisms are similar. However, we hope that this analysis of individuality can generate progress into what makes organisms unique from the rest of the members of its group. The control of genetic and environmental factors limits the variability in mice making it easier for machine-learning tools to make individualised predictions. “Easier” in this context means using simple models and few subjects. While humans have far more variability compared to mice, with the advent of ever-bigger datasets collected on human neurodevelopment, it will be possible in the near future to use more advanced machine-learning tools to predict some individuality in these data sets. It might even be possible to predict the development of neuroanatomical pathologies associated with neurological disorders prior to disorder onset. We hope that our analysis of mouse individuality spurs research in this direction.

Figure 8: Prediction and individualisation of neuroanatomical structure volumes. A) Predicted and Observed p36 structure volumes for all subjects and three representative structures: Primary Motor Cortex, Striatum, and the Barrel Cortex. When predicting any subject at p36, model was trained on all data excluding the predicted subject at p36 and all subjects at p65. Despite this exclusion, the model uses neuroanatomical information from earlier time points to predict structure volumes' variation from the average (horizontal line). B) Matrix plotting the Root-Mean-Square-Difference (RMSD) between the observed structure volumes (rows) and the predicted structure volumes (columns) for each subject at p36. Prediction used all data prior to and including p29. Each column (Subject X prediction) was centered so the diagonal RMSD (observation vs prediction for Subject X) is 0. Blue or red off-diagonal cells (observation Subject Y vs prediction Subject X) indicate RMSD greater or less than diagonal entry. Density plot shows diagonal RMSD elements (values as points and median as vertical line) tend to be less than off-diagonal RMSD (grey distribution), indicating high prediction specificity (One-sample Kolmogorov-Smirnov test: $D^+ = 0.79894, P < 10^{-15}, n = 28$). C) Models were trained on data closer to the time point of prediction (p36). For example, models corresponding to 'p10' were trained on p10 data and all earlier data (p03,p05,p07). As more data is included, model accuracy and specificity improved with significant specificity at p10 and above (One-sample Kolmogorov-Smirnov test: $D^+ = 0.34392, P = 0.001, n = 28$). D) Prediction Accuracy (RMSD) at p36 for each subject tends to improve over the course of neurodevelopment. When x-axis = X , subject data from ages $\leq X$ were used in the prediction. Trendlines denote patterns seen by sex and there is no significant difference in prediction accuracy between the sexes at any time (with shaded regions denoting standard error). Improvement in prediction accuracy (* $P < 0.05$, ** $P < 0.01$, *** $P < 0.001$) indicates neuroanatomy individualisation, which occurs significantly earlier in males (permutation test, $P = 0.025$).

3.5 Individualisation of neuroanatomy emerges earlier in males

Genetic and environmental variability is limited with the usage of standardized laboratory mice;

however, neuroanatomy is still remarkably individualised by adulthood. Most studies typically treat this individualisation as variability that is either accounted for as residuals or random effects in a statistical model in order to study other factors influencing variability, such as sex. We instead hypothesized that some of the variability in structure volumes of mature brains are inherently individualised. We sought to identify when this neuroanatomical individuality emerged across development, and whether it differed across sexes.

We used a set of linear mixed-effect regression models to predict structure volume at a specified time from the structure's volume at earlier times. Similar to validation methods followed by Tavor et al. [27], we used a leave-one-out approach to evaluate our model: the model for predicting volume of a structure s from a particular subject i at a time t was not trained on data containing information about any structures of subject i at time t . Figure 8A demonstrates the model's sensitivity by plotting the predicted and observed volume of three representative structures for every individual at p36. We quantitatively assessed the specificity of the model (Figure 8B) and found that predicted structure volumes for subject i generally matched observations for subject i closer than observations for other subjects. This is despite the fact that when trying to predict any subject at a time point, the model was trained on everything but the data from the subject at that time point. Yet, the prediction made is closer to the unseen subject data than the seen data from other subjects.

The model accurately captures neuroanatomical individualisation in the mature mouse brain. To investigate when this individuality emerged, we withheld more information regarding the subject to be predicted. When only considering data from p3 to predict p36 structure volumes, model specificity and accuracy is quite poor (56% probability predicted volumes match predicted subject better than other subjects; 0.13 Root-Mean-Square-Difference (RMSD)). However, when considering data from p10 and younger, specificity and accuracy improves (70%;0.12) and is quite high for data from p17 and younger (85%; 0.096mm³) (Figure 8C). Thus, only the first 10-17 days of brain development is sufficient to predict individualisation of mature mouse brain anatomy.

We plotted how the prediction accuracy (RMSD) at p36 changed for all subjects as we included more data closer to p36 (Figure 8D). As expected, accuracy increased as more data was included. Accuracy of predicting male neuroanatomy was not significantly different from predicting female neuroanatomy at any time point. However, male accuracy improved earlier than female accuracy (permutation test, $P=0.025$), needing only data from the first 7 days of life, while females required data from the first 17 days of life to improve accuracy significantly. Thus, male neuroanatomy individualises earlier than female neuroanatomy. This was also true ($P=0.034$) when we computed Root-Mean-Square-Percent-Difference (RMSPD) which is less biased against smaller brain structures (Supplementary Fig. S6). We found a similar pattern when predicting p29 ($P=0.025$) and p65 ($P=0.057$; Supplementary Fig. S7) time points, but it was no longer significant for p65. Furthermore, we also improved our model by using a random forest ($P=0.030$) and introducing a covariate for whole-brain volume ($P=0.036$) and found similar results.

5.3.8 Post-Development Neuroanatomy Prediction

We obtained structure volumes for all 182 bilateral structures in our atlas, for every subject and at every time point. When predicting the volume of a structure from subject i at time t , we excluded all data that belonged to subject i at time t to form the training set. We modelled each structure independently using weighted linear mixed-effects models and trained these models on the training set (Supplementary Section S1.1.5). Then, we used the trained models to predict the excluded data of subject i at time t and compared the predicted data for any structure j (\hat{y}_j) to the observed data for the same structure (y_j).

$$\text{RMSD} = \sqrt{\frac{1}{182} \sum_{j=1}^{182} (\hat{y}_j - y_j)^2} \quad (10)$$

$$\text{RMSPD} = \sqrt{\frac{1}{182} \sum_{j=1}^{182} \left(\frac{\hat{y}_j - y_j}{y_j} \right)^2} \quad (11)$$

Model accuracy was evaluated using root-mean-square-difference (RMSD) and root-mean-square-percent-difference (RMSPD) — the latter of which is less biased against small structures. RMSPD also corrects for whole-brain volume as normalisation to brain volume (V) results in the substitutions $\hat{y}_j \rightarrow \hat{y}_j/V, y_j \rightarrow y_j/V$, which leave Equation (11) unchanged.

We also trained two additional models to check for consistency. The first predicted structures after covarying for total brain volume. The second model used a random forest to predict structure

volumes from other structures at earlier times. All models showed similar results.

Individualisation of neuroanatomy was demonstrated by withholding information about the predicted subject prior to model training. We computed RMSD as a function of the accessed time point information x ; that is, when $x = 7$, the model must make its prediction on subject data from time points p7 and earlier (p5 and p3), and does not have access to data from time points after p7. To test if models made specific predictions, we computed the RMSD between the prediction for subject i and the observation for subject i (predicted vs self), and compared it to the RMSD between the prediction for subject i and observations for other subjects $\neq i$ (predicted vs other). Repeating this for all subject, we compared the RMSD values from predicted vs self and RMSD values from predicted vs other using Kolmogorov-Smirnov test.

As the accessed time point x increases, model predictions become better and RMSD decreases. To test for sex differences in the timing of individualisation of neuroanatomy, we fit a linear mixed-effects model to the RMSD versus the accessed timepoints x . The model had fixed effects of accessed time point, sex, their interactions and random effect of mouse. For both sexes, RMSD significantly decreased with time (the degrees-of-freedom calculated using Satterthwaite approximation [65]). We subsetted a time window over which individualisation occurs: the lower bound being the first time point where at least one sex had a significant RMSD decrease ($P < 0.05$) and the upper bound being the first time point where both sexes had a very significant RMSD decrease ($P < 0.01$). For each individual, we Z -transformed the RMSD values (subtracted their mean value over time and normalized to the standard deviation). Then, we computed the average Z -RMSD values for males and females in the time window. The difference between these two values is related to sexual dimorphisms in the timing of neuroanatomy individualisation between the sexes. To estimate the significance of this difference, we performed a permutation test by shuffling the sex labels of our data, recomputing the average Z -RMSD for males and females, and computing the difference. P -Values for the permutation test were defined by finding what fraction of the 10000 permuted differences exceeded or were equal to the true difference between average Z -RMSD male and female values.

S1.1.5 Neuroanatomy Prediction

We wanted to predict the absolute volumes of structures. To model this growth we used natural spline functions. N th-order natural splines are characterized by N basis functions of age t , where the k th basis function is represented by $f_k(t)$. For each structure, the structure volume y_{ij} in the training data was fit with the following model:

$$y_{ij} = \alpha_1 + \alpha_2 s_i + \sum_{k=1}^N \alpha_{k+2} f_k(t_{ij}) + \sum_{k=1}^N s_i \alpha_{k+N+2} f_k(t_{ij}) + \beta_{0i} + \varepsilon_{ij} \quad (12)$$

While increasing the order N of the natural splines allows one to model more complex growth curves, it can also overfit data leading to inaccurate predictions for data outside the training set. We computed Bayes Factors, using the `BayesFactor` package [6] to decide which order N of natural splines to use. Bayes Factors compute the evidence any model (12) has versus an intercept only model ($y_{ij} = \alpha_1 + \varepsilon_{ij}$) and assumes a reasonable set of priors on the predictors. We used the `BayesFactor` package's default Jeffreys prior for our analysis. By finding the Bayes Factor associated with Model (12) for values of spline order N ranging from one (linear growth) to the number of time points in the data minus one. The spline order N chosen for the structure's model is the one with the highest Bayes Factor associated with it.

Once we determined the order N of the natural splines modelling fixed growth effects, we wanted to similarly model random growth effects with M th-order natural splines and optimize M . The training data was fit with the model below:

$$y_{ij} = \alpha_1 + \alpha_2 s_i + \sum_{k=1}^N \alpha_{k+2} f_k(t_{ij}) + \sum_{k=1}^N s_i \alpha_{k+N+2} f_k(t_{ij}) + \beta_{0i} + \sum_{k=1}^M \beta_{ki} f_k(t_{ij}) + \varepsilon_{ij} \quad (13)$$

After fitting multiple models with different values for M , we chose the model with the lowest Bayesian information criterion [7] to predict structure volumes.

Lastly, we also placed a Gaussian weighting on data in the training set depending on the age. This step was motivated by the fact that when predicting volumes at a particular age, the time point closest to that age is the most informative. However, only considering the closest time point alone may be less informative than taking some information from the other time points. To balance the two extremes, we placed a gaussian weighting on the data. The gaussian weighting was centered on the time t which we want to predict and its spread (σ^2) can be optimized. A high σ^2 implies data over all time is weighted equally by the model, and a low σ^2 implies data closest to the prediction time t is weighed higher than data further away. The variance parameter was optimized using leave-one-out cross validation.

The model described above was used primarily in our study. However, we also explored two improvements to our model to check for consistency. The first was adding a covariate for total brain volume $V_{i,j}$ at the time point prior to the one being predicted ($j \rightarrow j - 1$), which was done to control for whole-brain volume effects in subjects. The model with this covariate is given below, and fixed N splines and random M splines are optimized as detailed above:

$$y_{ij} = \alpha_1 + \alpha_2 s_i + \sum_{k=1}^N \alpha_{k+2} f_k(t_{ij}) + \sum_{k=1}^N s_i \alpha_{k+N+2} f_k(t_{ij}) + \alpha_{k+2N+2} V_{i,(j-1)} + \beta_{0i} + \sum_{k=1}^M \beta_{ki} f_k(t_{ij}) + \varepsilon_{ij} \quad (14)$$

The second improvement was to use a random forest. Thus far, structures were modelled independently of each other, i.e. a structure's volume at a certain time t was predicted from the same structure's volume at earlier times. Using the random forest machine learning method, we can predict a structure's volume from other structures at an earlier time. To do so, we first fit the primary model described above to the training data and obtained the residuals of this model at the age t we wanted to predict. We then identified the volume of all 182 structures in the brain at the immediate earlier time and used them to model these residuals using a random forest from the `randomForest` package [8]. The random forest contained 500 trees and randomly sampled 5 structures at each tree split. The final predicted value was the sum of predicted values from both the initial model and the random forest.

Other issues with the study:

-It is an overgeneralization that females have a predisposition for disorders with a late onset. Male adolescents are more prone to addiction and schizophrenia during adolescence.

Noted. The wording in the introduction and in the discussion has been corrected to convey that not only females are prone to disorders that have a later onset in adolescence. The preponderance of males to being more prone to addiction and schizophrenia in adolescence, prior to when onset occurs in females, has been mentioned.

Page 1

processes including pain [1], learning and memory [2] and language [3]. Notably, there are robust sex differences in the prevalence, age of onset, and course of various psychiatric disorders. Males tend to have a predisposition for disorders that have earlier onset, many of which emerge during childhood, including autism spectrum disorders, attention deficit disorders and Tourette syndrome. Females tend to have a predisposition for disorders that have later onset, during adolescence and early adulthood, which include major depressive disorders, anxiety disorders and eating disorders [4].

Page 9

areas in females predominate the brain in later, post-pubertal life. These patterns mirror what is known about sex differences in age of onset for many psychiatric disorders: males are more likely to be diagnosed with disorders that are developmental in nature, and have an onset during childhood, while females are disproportionately diagnosed with disorders that have an emotional nature and emerge in adolescence and young adulthood [4]. Although, it should be noted that males are more prone to addiction and schizophrenia in adolescence, prior to when onset occurs in females [4] [29].

-Does the generalization presented in title hold if absolute size differences are examined? There is a growing literature indicating greater pruning in female rodents during adolescence in both the cortex and hippocampus. This does not lead to greater size and is another indication of how misleading relative size is.

We repeated our analysis using both cortical thickness and absolute volumes and found similar patterns as with the relative volumes. There were some regions of the cortex that were larger in males by adulthood but emerged around puberty. However, the pattern of larger areas in males tending to occur early and larger areas in female occurring late held true for all the neuroanatomical measurements analysed.

Page 19

5.3.7 Cortical Thickness

The Pydpipe pipeline was used to segment the cortex [24] for all nine age-consensus averages and these cortical segmentation are then transformed to every subject image using appropriate transformations from Level 1 of the two-level registration (Detailed in Supplementary Section S1.1.3). The end result of these procedures is a cortical segmentation for every subject and at every time point. Laplace's equation was then solved for all subject images and at every time point, with the inner and outer cortical surface having different potentials, thereby defining the boundary conditions. A property of Laplace's equation is that streamlines are always perpendicular to equipotential surfaces. Taking advantage of this property, for each point on the cortical surface, the thickness was defined as the length of the streamline connecting the inner and outer cortical surface. Using the transformations from both levels in the two-level registration, we mapped any point on the cortical surface of the consensus average mouse to that same point at any age and for any subject.

Page S8

S1.2.2 Cortical Thickness and volume measurement agree on areas larger areas in females emerging in post-pubertal life

Regions larger in males emerge in early life and regions larger in females emerge in later life. Supplementary Figures S2–S4 show this pattern is consistent across different canonical measures of neuroanatomy. For both relative and absolute volumes, we identified the top 5% of voxels largest in males and top 5% of voxels largest in females at p65. Similarly for absolute cortical thickness, we identified the top 5% of vertices thickest in males and top 5% thickest vertices in females at p65. This was done to ensure a consistent size of voxel or vertex elements for clustering. We then clustered these voxels and vertices by their effect size trajectories over time and identified three clusters. The pattern of larger/thicker areas emerging in early life in males and later life in females held across the different measures of neuroanatomy.

Figure S2: Top 5% of largest voxels (relative to whole brain) in males and females clustered by their effect sizes over time. Cluster 1 and 2 correspond to regions larger in males in adulthood and these sexual dimorphisms emerge early. Cluster 3 corresponds to regions larger in females and emerges around puberty.

Figure S3: Top 5% of largest voxels in males and females clustered by their effect sizes over time. Like the case with relative volumes, cluster 1 and 2 correspond to regions larger in males in adulthood and these sexual dimorphisms emerge early. Cluster 3 corresponds to regions larger in females and emerges around puberty.

Figure S4: Top 5% of largest vertices in males and females clustered by their effect sizes over time. Like with relative and absolute volumes, cluster 1 correspond to regions larger in adult males and these dimorphisms occur early in development, while cluster 3 corresponds to regions larger in females and these dimorphisms occur around puberty. Similarly, cluster 2 in both the thickness analysis and volume analysis correspond to regions larger in males whose dimorphisms emerge after the regions in cluster 1. However, while volume analysis had both cluster 1 and cluster 2 dimorphisms emerging in the first 10 days of life, cortical thickness analysis shows regions in cluster 2's dimorphisms emerging around male puberty.

-Exposure to any anesthetic during the early postnatal period in altricial rodents leads to extensive cell death in the brain (see work by JW Olney) and there is evidence that the effect is larger in males (JL Nunez). This is a confound in the present study. The authors did look at body weight and pubertal onset compared to controls, but the anesthetic effects are neural rather than in the body.

We have added a comment under Results Section 3.1 on the implications of neonatal anesthetic on neurological outcomes, and its potential to have sex-specific effects. Furthermore, we have performed experiments on additional cohorts of mice in order to assess neural development across neonatal scanned mice, their non-scanned littermates and non-scanned control mice. We assessed righting reflex, eye opening and ambulation in the open field test. We found no effect of group (scanned, non-scanned littermate, non-scanned control), nor did we find a group-sex interaction effect on any of the neonatal behavioural metrics measured. We kept these mice until adulthood in order to assess open field at p65. While we did find a significant effect of group on time spent in centre, total ambulatory distance was the same across all groups; furthermore, we detected no group-sex interaction effect. More details on the methods are found in Methods S1.1.1 and Results are shown in Figure S1.

Page 5

estradiol between groups of mice. Neonatal anaesthesia exposure can cause cell death in the brain, and can affect males more severely [23]; thus, we assessed several neonatal neurodevelopmental outcomes (righting reflex, eye opening and open field) on additional groups of mice (Supplementary Methods S1.1.1). We found no significant effects of group or a group-sex interaction on any of the neonatal metrics collected (Supplementary Fig. S1). We conclude that the effect of MEMRI scanning on neurodevelopment was small and not sexually biased.

Methods S1.1.1

S1.1.1 Behavioural assessment

To assess neonatal brain development and behavioural outcomes of mice undergoing scanning, developmental milestone and behavioural testing was conducted on additional groups of mice. Testing was conducted on three groups of mice. The first group consisted of scanned mice: these mice were scanned at postnatal days 3, 5, 7, 10 and 17, and thus were exposed to isoflurane during the scans and to maternal MnCl₂ at postnatal days 2, 4, 6 and 9, as well as an intraperitoneal injection of MnCl₂ at postnatal day 16. The second group consisted of their non-scanned littermates. These mice were exposed to maternal MnCl₂ at postnatal days 2, 4, 6 and 9. The third group consisted of non-scanned control mice, who were not exposed to any isoflurane or MnCl₂. MnCl₂ administration and scanning procedures are previously described in sections above. Six cages of C57BL/6 mice were used, each culled to a litter size of 6, with the exception of one cage that had 5 pups. Four cages were randomly selected for scanning. 1-2 mice were used from each cage to be scanned, while the remainder of the mice were non-scanned littermates, resulting in 7 scanned mice (4 male, 3 female) and 17 non-scanned littermates (7 male, 10 female). There were 11 non-scanned control mice (4 male, 7 female).

Behavioural testing was done prior to $MnCl_2$ administration or scanning if both fell upon the same day. Righting reflex was conducted on postnatal days 4, 5, and 6. Prior to testing, the dam was removed from the home cage. Each pup was tested individually. The pup was placed on its back, and the time required for the pup to flip over and place all four paws on the ground surface was measured. If no righting took place within 30 seconds, the pup was picked up and righted manually. After all pups in one cage were tested, the dam was returned. Eye opening was observed daily from postnatal day 10 until day 17. Each mouse was given a score depending on number of eyes open: 0 for no eyes open, 1 for one eye open and 2 for both eyes open. Open field was conducted on postnatal day 16. Behaviour was recorded in an arena that was 44 cm², with 16 beams on each axis (1 inch apart) (Med Associates Inc, Fairfax VT). Each arena was enclosed inside a sound attenuating chamber. The light intensity used was 200 LUX in the centre, and 150 LUX in the periphery. Data was collected with Activity Monitor 7 (Med Associates Inc, Fairfax VT). Six mice were tested at a time. Length of neonatal open field testing was 30 minutes. Time spent in the centre of the open field, as well as total ambulatory distance were measured. Following testing, mice were returned to their home cage. Open field was performed for a second time at postnatal day 65. The length of testing was 10 minutes. For an unrelated reason, one of our cages was terminated prematurely by veterinarian technicians, thereby decreasing our number of female scanned mice by one, and our female non-scanned littermates by two.

Righting reflex time and eye opening score were compared across scanned (S), non-scanned littermates (L) and non-scanned control (C) animals by running three linear mixed effects models, and then comparing across models to assess whether group and/or sex had a significant effect. The first linear mixed effects model had fixed effects of group, postnatal day, sex, and their interactions, with a random effect of individual mouse. The second model had fixed effects of postnatal day, sex, and their interaction, and a random effect of individual mouse. These two models were compared with a likelihood ratio test to assess whether group affected righting reflex time or eye opening score. A third model was run with fixed effects of group, postnatal day, group-postnatal day interaction, sex and a sex-postnatal day interaction, with a random effect of individual mouse. This third model was compared to the first model with a likelihood ratio test to assess whether there was a group by sex interaction. For open field, time spent in the centre of the open field and total ambulatory distance were analysed using linear models.

Figure S1: Time to right, eye opening, as well as time spent in centre and total ambulatory distance travelled in the open field was assessed neonatally across scanned mice (S), their non-scanned littermates (L) and non-scanned controls (C). A) There was no effect of group ($\chi^2_8 = 14.54, P = 0.07$), nor was there a group-sex interaction ($\chi^2_4 = 6.59, P = 0.16$) on time it took for pups to right themselves across postnatal days 4, 5 and 6. B) There was also no effect of group ($\chi^2_{32} = 20.61, P = 0.94$) or a group-sex interaction effect ($\chi^2_{16} = 9.30, P = 0.90$) on when eyes opened across postnatal days 10 to 17. For both A and B, linear mixed-effects models were used to create trendlines and bars representing standard error. C) There was no effect of group ($F_{2,29} = 0.47, P = 0.63$) or a group-sex interaction ($F_{2,29} = 0.64, P = 0.54$) on time spent in the centre of the open field at postnatal day 16, nor was there an effect of group on total ambulatory distance travelled ($F_{2,29} = 1.16, P = 0.33$) or a group-sex interaction ($F_{2,29} = 0.17, P = 0.85$). Thus, no neonatal behavioural metrics collected were significantly impacted by scanning, and the results were the same across both males and females. These mice were kept for further testing in the open field as adults (postnatal day 65). Although there was no group-sex interaction on centre time ($F_{2,26} = 0.91, P = 0.41$), there was a significant effect of group ($F_{2,26} = 6.81, P = 0.004$) as scanned mice spent less time in the centre of the open field compared to non-scanned controls (post hoc Tukey test $P_{adj} = 0.003$). Total ambulatory distance, however, did not show any significant differences across group ($F_{2,26} = 1.37, P = 0.27$), or by sex within each group ($F_{2,26} = 1.34, P = 0.28$). For both C and D, mean and standard error (bars) were calculated using linear models.

-The authors often cite work examining humans but the current study is on mice. For example, in the last paragraph on p.5, they state that males often have bigger brains than females and they cite a study on humans. Does this generalize to rodents and to the strain of mouse they are examining?

Yes, although the difference is smaller, a whole brain size difference across males and females does generalize to rodents, and to the C57BL/6J strain of mice that we are studying.

Spring, Shoshana, Jason P. Lerch, and R. Mark Henkelman. "Sexual dimorphism revealed in the structure of the mouse brain using three-dimensional magnetic resonance imaging." Neuroimage 35.4 (2007): 1424-1433.

They also make some generalizations about sex difference in pain that are not supported by the literature. Mogil has found that sex differences in pain sensitivity varies with the species and even among strains of a species.

We have removed a sentence in the discussion to avoid making generalizations about sex differences in pain that are too broad.

Thank you for bringing to our attention that sex differences in pain sensitivity can vary with species and strains within in species. We have now mentioned this in our discussion. However, the structures and circuits involved in pain processing and analgesia are highly conserved across species. Thus, we do not believe that species and strain differences in pain processing significantly impacts our interpretation of the structures in the clusters.

evidence that indicates there are sex differences in pain and analgesia [1]. Although sex differences in pain sensitivity can vary with rodent species and even with strain [46], the underlying structures implicated in pain are highly conserved across mammalian species. Interestingly, these structures come from all 4 clusters, belonging to clusters that show relatively larger areas in males in early life, and relatively larger areas in females in later life. This may be a reflection of their sensitivity to hormones during both neonatal and adult life. Indeed, there is evidence suggesting that pain processes are sensitive to, and can be modulated by both neonatal gonadal hormones in males [47], later-life activational hormones in females [48] [49], as well as both [50]. Cluster 4 describes areas of the

REVIEWERS' COMMENTS:

Reviewer #2 (Remarks to the Author):

The authors have made significant efforts to address my comments, including specifically:

(1) Assessment of potential confounds of early-life manganese exposure - which do not seem to be present

(2) Inclusion of additional references and points for discussion in terms of potential clinical relevance of their findings

(3) Inclusion of a new analysis for prediction of (mouse) neuroanatomy in a sex-specific manner which has some translational and cross-disciplinary potential.

I have no further comments and recommend publication.

Reviewer #3 (Remarks to the Author):

No further comments

Reviewer #4 (Remarks to the Author):

The authors have adequately addressed the questions raised in my previous review.

Reviewer #5 (Remarks to the Author):

Qui and colleagues eloquently expound on sex differences and their importance in understanding the origins of many neural disorders. Unfortunately the careful measurement of relative size changes during development, especially relative size, does not illuminate the problem which is on the cellular/molecular level. The essence of the present study is a modest technical point that the technique of manganese-enhanced MRI can be used to observe size changes in the same animals across ages.

Given that I appear to be the lone reviewer making this point, I would like to insist on some small changes to better represent what was done here:

-When the word "size" is used anywhere in the text, it should be modified with "relative" or phrase with similar meaning. This includes the title of the manuscript. Such a modification would be a more accurate representation of what is being examined.

-The table in the supplement comparing the absolute and relative volume of several neural regions (S1.3) should be moved to the main portion of the manuscript. It will allow the reader to compare the merits of both methods directly without further digging into the supplement.

Reviewer #2 (Remarks to the Author):

The authors have made significant efforts to address my comments, including specifically:

- (1) Assessment of potential confounds of early-life manganese exposure - which do not seem to be present
- (2) Inclusion of additional references and points for discussion in terms of potential clinical relevance of their findings
- (3) Inclusion of a new analysis for prediction of (mouse) neuroanatomy in a sex-specific manner which has some translational and cross-disciplinary potential.

I have no further comments and recommend publication.

We wish to thank Reviewer #2 for their insightful comments, in particular comments regarding possible cognitive effects of early-life manganese exposure. The additional experiments conducted to address these comments resulted in a stronger manuscript.

Reviewer #3 (Remarks to the Author):

No further comments

Thank you to Reviewer #3 for their comments.

Reviewer #4 (Remarks to the Author):

The authors have adequately addressed the questions raised in my previous review.

Many thanks to Reviewer #4 for their comments.

Reviewer #5 (Remarks to the Author):

Qui and colleagues eloquently expound on sex differences and their importance in understanding the origins of many neural disorders. Unfortunately the careful measurement of relative size changes during development, especially relative size, does not illuminate the problem which is on the cellular/molecular level. The essence of the present study is a modest technical point that the technique of manganese-enhanced MRI can be used to observe size changes in the same animals across ages.

Given that I appear to be the lone reviewer making this point, I would like to insist on some small changes to better represent what was done here:

-When the word "size" is used anywhere in the text, it should be modified with "relative" or phrase with similar meaning. This includes the title of the manuscript. Such a modification would be a more accurate representation of what is being examined.

Thank you for this comment. We have added the words “relative” or “relatively” in several places where we describe size in the results, discussion and figure captions of the manuscript. Throughout the rest of the manuscript, we have thoroughly checked to ensure that it is clear when we are referring to relative volumes.

189 Cluster 1 describes areas in the brain that were relatively larger in males throughout development,
190 and this dimorphism emerged early in development. Voxels from this cluster reside in the BNST,
191 MeA, MPON, OB, hippocampus and cingulate cortex. This cluster had a higher growth rate in
192 males, which stabilized in later life. Cluster 2 corresponds to regions **relatively** larger in males that
193 emerged later in development. This cluster shows growth rate that is biased towards males in early

305 The clustering analysis shows 4 groups of areas, each cluster characterized by a unique trajectory
306 of sexually dimorphic development over time. Clustering by **relative** volume effect size allows us to
307 provide insight on areas that cluster together as networks of connected structures that may share
308 function [39]. Parts of the BNST, MeA, MPON and OB are featured prominently in Cluster 1. These
309 areas are well known to be sexually dimorphic and larger in males, and these sex differences depend
310 on the presence of neonatal hormones [6] [7] [9] [40] [41] [42]. These areas are part of a functionally

879 Figure 5: Expansion of neuroanatomical structures in males and females over time. Each column
880 follows a coronal cross-section of the developing brain through the nine experimental time points
881 with red regions **relatively** larger in males and blue regions larger in females. Nine age-centered
882 linear-mixed effects models were fit to the data, one for each experimental time point. Each model
888 pallidum. Cluster 3 voxels trajectory switches from being larger in males in early life, to larger in
889 females post-puberty. Parts of the sensory cortex and PAG belong to this cluster. Cluster 4 contains
900 regions that are not sexually dimorphic in early life but become **relatively** larger in females over the
901 course of development. This cluster includes association related areas such as parts of the central
902 thalamus and temporal association cortex; and motor related areas such as parts of the hindbrain,

*The title has also been changed to reflect the usage of relative volumes in our manuscript. It has been changed to: “Mouse MRI shows brain areas **relatively** larger in males emerge before those larger in females.”*

-The table in the supplement comparing the absolute and relative volume of several neural regions (S1.3) should be moved to the main portion of the manuscript. It will allow the reader to compare the merits of both methods directly without further digging into the supplement.

Noted. This table has been moved to the main text to facilitate comparisons across absolute and relative volume results for the reader. It is now cited in our Results section 3.2. Thank you for these suggestions, which make our results more clear.

150 absolute volumes, but did exhibit significant differences in relative volumes. Compared to male-
151 enlarged structures, the PAG became relatively larger in females at a later developmental time,
152 around p29 (**summary structure data in Table 1 and S1**). Correction for whole-brain size using
153 relative volume measurements from MEMRI data reveal time courses of well-established and novel
154 sexual dimorphisms, highlighting the strength of whole-brain MRI.